JCB | Journal of Cell Biology

**REPORT**

# Kinetochore phosphatases suppress autonomous Polo-like kinase 1 activity to control the mitotic checkpoint

Marilia H. Cordeiro*, Richard J. Smith*, and Adrian T. Saurin 

**Local phosphatase regulation is needed at kinetochores to silence the mitotic checkpoint (a.k.a. spindle assembly checkpoint [SAC]). A key event in this regard is the dephosphorylation of MELT repeats on KNL1, which removes SAC proteins from the kinetochore, including the BUB complex. We show here that PP1 and PP2A-B56 phosphatases are primarily required to remove Polo-like kinase 1 (PLK1) from the BUB complex, which can otherwise maintain MELT phosphorylation in an autocatalytic manner. This appears to be their principal role in the SAC because both phosphatases become redundant if PLK1 is inhibited or BUB–PLK1 interaction is prevented. Surprisingly, MELT dephosphorylation can occur normally under these conditions even when the levels or activities of PP1 and PP2A are strongly inhibited at kinetochores. Therefore, these data imply that kinetochore phosphatase regulation is critical for the SAC, but primarily to restrain and extinguish autonomous PLK1 activity. This is likely a conserved feature of the metazoan SAC, since the relevant PLK1 and PP2A-B56 binding motifs have coevolved in the same region on MADBUB homologues.**

## Introduction

The mitotic checkpoint, also known as the spindle assembly checkpoint (SAC), prevents mitotic exit until chromosomes have attached to microtubules via the kinetochore (Corbett, 2017; Saurin, 2018). MPS1 kinase initiates SAC signaling by localizing to unattached kinetochores and phosphorylating the SAC scaffold KNL1 on repeat motifs known as "MELT repeats" (for the amino acid consensus Met-Glu-Leu-Thr; London et al., 2012; Shepperd et al., 2012; Yamagishi et al., 2012). Once phosphorylated, these MELT motifs recruit the heterotetrameric BUB1:BUB3:BUB3:BUBR1 complex (hereafter BUB complex) to kinetochores (Overlack et al., 2015; Primorac et al., 2013; Vleugel et al., 2013; Zhang et al., 2014), which, directly or indirectly, recruits all other proteins needed to activate the SAC and block mitotic exit (Corbett, 2017; Saurin, 2018). Once kinetochores attach to microtubules, the local SAC signal must be rapidly extinguished by at least three different mechanisms: (1) localized MPS1 activity is inhibited (Aravamudhan et al., 2015; Hiruma et al., 2015; Ji et al., 2015), (2) key phosphorylation sites, such as the MELT repeats, are dephosphorylated by KNL1-localized phosphatases (Espert et al., 2014; Espeut et al., 2012; Meadows et al., 2011; Nijenhuis et al., 2014; Rosenberg et al., 2011), and (3) dynein motors physically transport SAC components away from kinetochores down microtubules (Bader and Vaughan, 2010).

One key unexplained aspect of SAC signaling concerns the role of Polo-like kinase 1 (PLK1; Combes et al., 2017). PLK1 interacts via its Polo-box domain (PBD) to phospho-epitopes on various different kinetochore complexes, including two CDK1 phosphorylation sites on the BUB complex (BUB1-pT609 and BUBR1-pT620; Elowe et al., 2007; Qi et al., 2006; Wong and Fang, 2007). PLK1 has similar substrates preferences to MPS1 (Dou et al., 2011; Hennrich et al., 2013), and it shares at least two key substrates that are critical for SAC signaling: the KNL1-MELT motifs and MPS1 itself, including key sites in the MPS1 activation loop (Espeut et al., 2015; Ikeda and Tanaka, 2017; von Schubert et al., 2015). PLK1 can therefore enhance MPS1 kinase activity and also directly phosphorylate the MELT motifs to support SAC signaling, perhaps from its localized binding site on BUB1 (Ikeda and Tanaka, 2017). It is unclear why PLK1 is needed to cooperate with MPS1 in SAC signaling and, importantly, what inhibits PLK1 signaling to allow MELT dephosphorylation and SAC silencing upon microtubule attachment. We set out to address these questions by examining the role of the kinetochore-localized phosphatases PP1-KNL1 and PP2A-B56.

Division of Cellular Medicine, School of Medicine, University of Dundee, Dundee, UK.

*M.H. Cordeiro and R.J. Smith contributed equally to this paper; Correspondence to Adrian T. Saurin: a.saurin@dundee.ac.uk.



# Results and discussion

## PP1-KNL1 and PP2A-B56 antagonize PLK1 recruitment to the BUB complex

Inhibition of PP1-KNL1 or knockdown of PP2A-B56 both enhance PLK1 recruitment to kinetochores (Foley et al., 2011; Liu et al., 2012). To test whether this was due to localized phosphatase inhibition at the BUB complex, we inhibited the recruitment of PP2A-B56 to BUBR1 (BUBR1$^{\Delta PP2A}$) and compared this to a PP1-KNL1 mutant (KNL1$^{\Delta PP1}$), as used previously (Liu et al., 2012; Nijenhuis et al., 2014; note that, in these and all subsequent experiments, siRNA-mediated gene knockdown was used in combination with doxycycline-inducible replacement of the WT or mutant gene from an FRT locus; see Materials and methods). This demonstrated that removal of either PP1 from KNL1 or PP2A-B56 from BUBR1 increased PLK1 levels at unattached kinetochores (Fig. 1 A; nocodazole was added in these and all subsequent experiments to prevent microtubule attachment from indirectly affecting kinetochore signaling). We suspected that the increased PLK1 was due to enhanced binding to the BUB complex, because depletion of BUB1 and/or BUBR1 also removes the BUBR1:PP2A-B56 complex from kinetochores, but this did not enhance PLK1 levels (Fig. S1, A–C). In fact, kinetochore PLK1 was considerably reduced when the whole BUB complex was removed (see siBUB1 + siBUBR1 in Fig. S1, A–C). PLK1 binds via its PBD to CDK1 phosphorylation sites on BUB1 (pT609; Qi et al., 2006) and BUBR1 (pT620; Elowe et al., 2007; Wong and Fang, 2007); therefore, we raised antibodies to these sites and validated their specificity in cells (Fig. S2, A–C). Immunofluorescence analysis demonstrated that phosphorylation of both sites is enhanced at unattached kinetochores in KNL1$^{\Delta PP1}$ or BUBR1$^{\Delta PP2A}$ cells (Fig. 1, B and C; and Fig. S2, B–E). This is the reason that PLK1 kinetochore levels increase when PP2A is removed, because the elevated PLK1 levels can be attenuated by BUBR1-T620A mutation (Fig. 1 D) and completely abolished by additional mutation of BUB1-T609A (Fig. 1 E). Therefore, these data demonstrate that PP1-KNL1 and BUBR1-bound PP2A-B56 antagonize PLK1 recruitment to the BUB complex by inducing the dephosphorylation of key CDK1 phosphorylation sites on BUBR1 (pT620) and BUB1 (pT609).

## PP1-KNL1 and PP2A-B56 antagonize PLK1 to allow SAC silencing

PLK1 is able to support SAC signaling by phosphorylating the MELT motifs directly (Espeut et al., 2015; Ikeda and Tanaka, 2017; von Schubert et al., 2015). Therefore, we hypothesized that the increased BUB–PLK1 levels in KNL1$^{\Delta PP1}$ and BUBR1$^{\Delta PP2A}$ cells could help to sustain MELT phosphorylation and the SAC when MPS1 is inhibited. To address this, we examined the effect of PLK1 inhibition with BI-2536 under these conditions. Fig. 2, A and B, and Fig. S3, A and B, show that MELT dephosphorylation and mitotic exit are both attenuated following MPS1 inhibition in nocodazole-arrested KNL1$^{\Delta PP1}$ or BUBR1$^{\Delta PP2A}$ cells, as demonstrated previously (Espert et al., 2014; Nijenhuis et al., 2014). Importantly, however, these effects are rescued if PLK1 and MPS1 are inhibited together (Fig. 2, C and D; and Fig. S3, C and D; note the thick vertical bars in these violin plots display 95% confidence intervals [CIs], which can be used for statistical

comparison of multiple time points/treatment by eye; see Materials and methods). Therefore, PLK1 is able to maintain the SAC when MPS1 is inhibited in KNL1$^{\Delta PP1}$ or BUBR1$^{\Delta PP2A}$ cells. The sustained SAC phenotype in BUBR1$^{\Delta PP2A}$ cells is due to enhanced PLK1 levels at the BUB complex, because MELT dephosphorylation and SAC silencing are also rescued if MPS1 is inhibited when the PLK1 binding motif on BUBR1 is mutated (BUBR1$^{\Delta PP2A\text{-}T620A}$; Fig. 2, E and F; and Fig. S3 E). Furthermore, additional mutation of BUB1-T609A (BUBR1$^{\Delta PP2A\text{-}T620A}$ + BUB1$^{T609A}$) is sufficient to revert phospho-MELT (pMELT) and SAC strength to levels that are indistinguishable from WT cells (Fig. 2, G and H; and Fig. S3, F–I). Importantly, removal of PLK1 alone from the BUB complex (BUBR1$^{T620A}$ + BUB1$^{T609A}$) reduced basal levels of pMELT and weakened the SAC, implying that the pool of PLK1 that is bound to the BUB complex also enhances basal SAC signaling (Fig. 2, G and H). Collectively, these data demonstrate that PLK1 binds to the BUB complex to enhance MELT phosphorylation, and kinetochore phosphatases must antagonize this recruitment to allow the SAC to silence following MPS1 inhibition. Although the majority of PLK1 is recruited to BUBR1-pT620, there is significant pool that localizes to BUB1-pT609, and removal of PLK1 from both BUBR1 and BUB1 causes the greatest weakening of the SAC (Fig. 1, D and E; and Fig. 2 H).

## BUB1–PLK1 locally amplifies KNL1-MELT signaling without affecting MPS1 activity

As well as phosphorylating the MELT repeats, PLK1 can also phosphorylate MPS1 to enhance its kinase activity (Combes et al., 2018; Ikeda and Tanaka, 2017; von Schubert et al., 2015). Therefore, PLK1 bound to the BUB complex could enhance MELT phosphorylation directly and/or indirectly by elevating MPS1 activity. The effects of PLK1 are apparent following MPS1 inhibition, suggesting these effects are direct. However, pharmacological inhibition is never absolute; therefore, PLK1 could simply enhance MPS1 activity to increase the threshold required for inhibition. To test this possibility, we probed the phosphorylation state of another kinetochore-localized MPS1 substrate that contributes to SAC signaling: MAD1-pT716 (Allan et al., 2020; Faesen et al., 2017; Ji et al., 2018; Ji et al., 2017). Fig. 3, A–C, demonstrates that MAD1-pT716 is dephosphorylated with identical kinetics when MPS1 is inhibited in the presence or absence of kinetochore phosphatases, implying that kinetochore phosphatases do not impact directly on MAD1-pT716 or more generally on kinetochore MPS1 activity. This was surprising given that removal of PP2A-B56 from BUBR1 has been shown recently to enhance MPS1 T-loop phosphorylation and prevent its dephosphorylation following MPS1 inhibition (Hayward et al., 2019). However, it is possible that MPS1 inhibition with 2.5 µM AZ-3146 inhibits downstream MPS1 activity sufficiently, irrespective of the level of MPS1 T-loop phosphorylation. Therefore, we performed additional assays over a range of AZ-3146 doses in BUBR1$^{WT}$ and BUBR1$^{\Delta PP2A}$ cells. These experiments demonstrated that MAD1-pT716 dephosphorylation is completely unaffected by a BUBR1$^{\Delta PP2A}$ mutation (Fig. 3 D), even though BUBR1 removal is strongly inhibited (Fig. 3 E), as expected due to the maintenance of MELT phosphorylation (Fig. 2, A and B; Espert et al., 2014; Nijenhuis et al., 2014). Therefore,

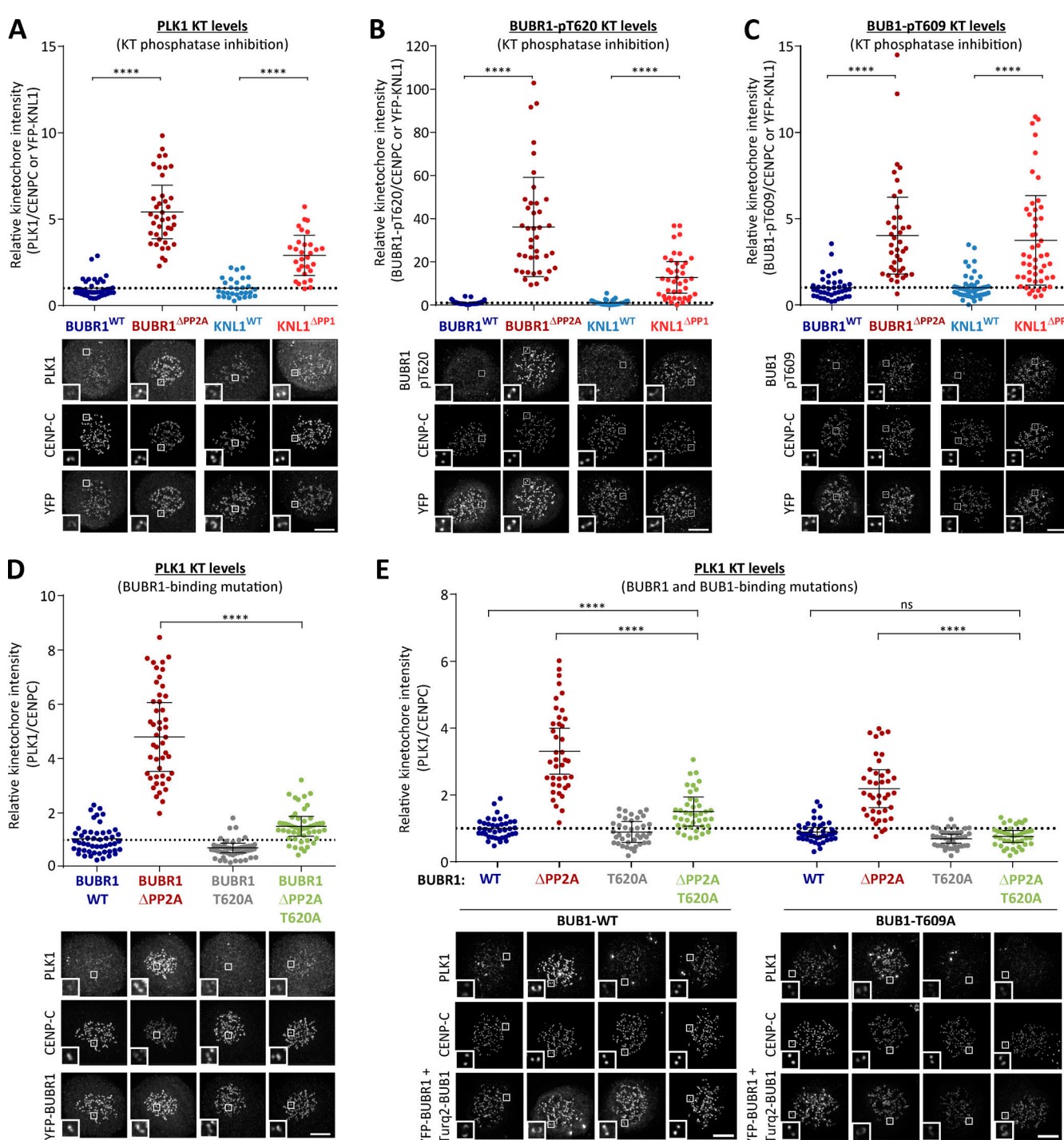

Figure 1. **Kinetochore phosphatases PP1 and PP2A-B56 antagonize PLK1 recruitment to the BUB complex. (A–C)** Effect of phosphatase-binding mutants on levels of PLK1 (A), BUBR1-pT620 (B), and BUB1-pT609 (C) at unattached kinetochores in nocodazole-arrested cells. Mean kinetochore intensities from 30–50 cells, three to five experiments. **(D and E)** Effect of mutating the PLK1 binding site on BUBR1 (pT620) alone (D) or BUBR1 (pT620) and BUB1 (pT609; E) on PLK1 kinetochore levels in nocodazole-arrested BUBR1^WT/ΔPP2A cells. Mean kinetochore intensities from 40 cells per condition, four experiments. All values in E are normalized to BUBR1-WT+BUB1-WT control. For all kinetochore intensity graphs, each dot represents a cell, and the error bars display the variation between the experimental repeats (displayed as ±SD of the experimental means). Two-tailed, nonparametric Mann-Whitney unpaired *t* tests were performed to compare the mean values between experimental groups. Example immunofluorescence images were chosen that most closely resemble the mean values in the quantifications. The insets show magnifications of the outlined regions. Scale bars, 5 µm. Inset size, 1.5 µm. ****, P < 0.0001; ns, nonsignificant.

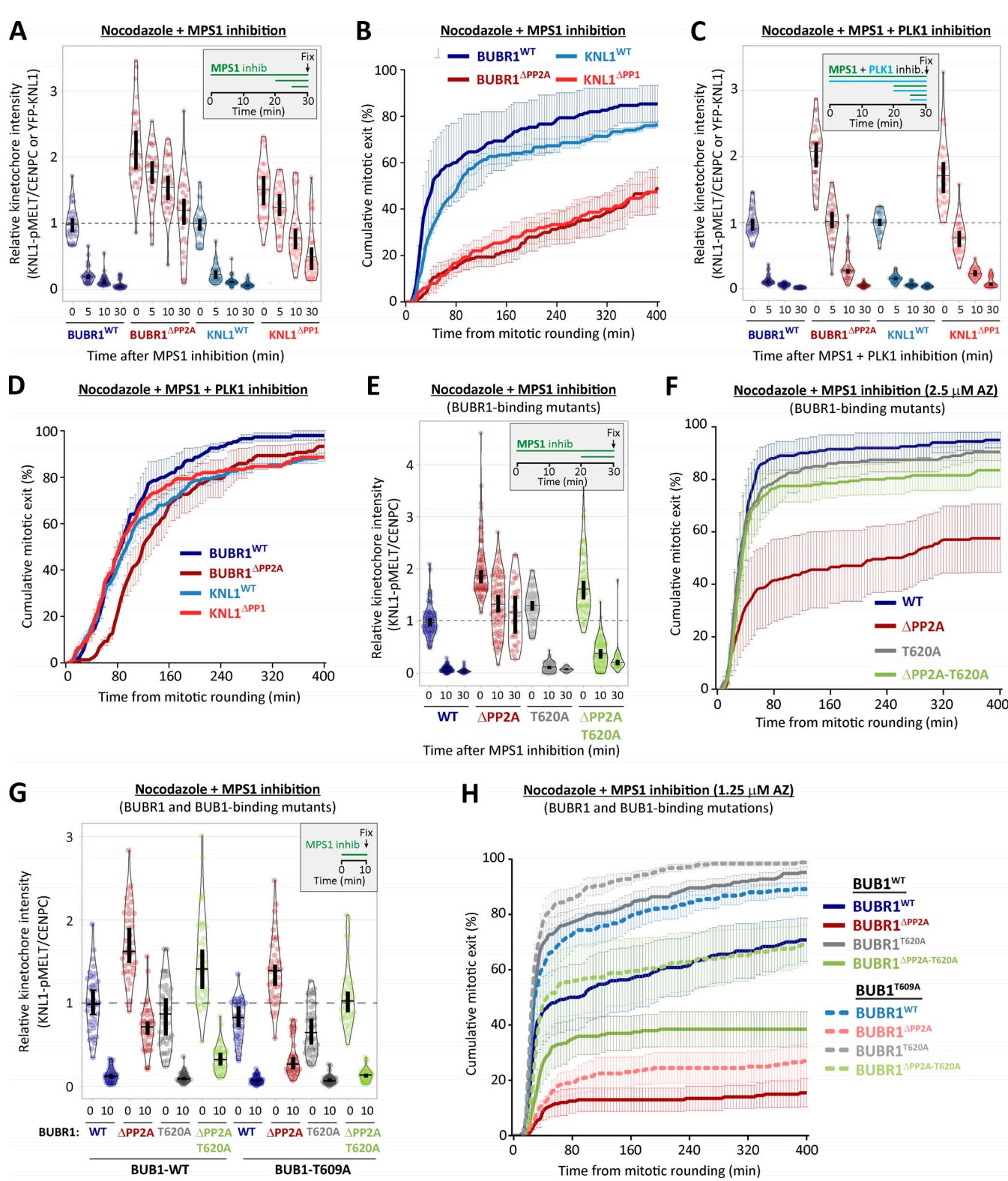

Figure 2. **Kinetochore phosphatases PP1 and PP2A-B56 remove PLK1 from the BUB complex to silence the SAC. (A–D)** Effects of phosphatase-binding mutants on KNL1-MELT dephosphorylation (A and C) and duration of mitotic arrest (B and D) in nocodazole-arrested cells treated with MPS1 inhibitor AZ-3146 (2.5 μM) either alone (A and B) or in combination with the PLK1 inhibitor BI-2536 (100 nM; C and D). MG132 was included in all MELT phosphorylation experiments to prevent mitotic exit after addition of the MPS1 inhibitor. A and C display kinetochore intensities from 30 cells per condition, three experiments. Intensities are relative to CENP-C in BUBR1 cells or YFP-KNL1 in KNL1 cells, and all BUBR1 and KNL1 intensities are normalized to their respective WT, 0 min time point. B and D show mean (±SD) of 150 cells per condition from three experiments. **(E and F)** Effect of mutating the PLK1 binding site on BUBR1 (pT620) on MELT dephosphorylation (E) and mitotic exit (F) in nocodazole-arrested BUBR1^WT/ΔPP2A cells, treated as in A and B. E displays kinetochore intensities of

30–80 cells per condition from three to seven experimental repeats. F shows the means (±SD) of 200 cells from four experiments. **(G and H)** Effect of mutating both PLK1 binding sites BUBR1 (pT620) and BUB1 (pT609) on MELT dephosphorylation (G) and mitotic exit (H) in nocodazole-arrested cells, treated with MPS1 inhibitor AZ-3146 (2.5 μM in G and 1.25 μM in H). The 1.25 μM AZ dose was selected because it is then possible to see effects that weaken or strengthen the WT SAC response (2.5 μM AZ-3146 data are displayed in Fig. S3 I). G displays kinetochore intensities of 40 cells per condition from four experimental repeats, and all intensities are normalized to BUBR1-WT+BUB1-WT control. H shows the means (±SEM) of 200–250 cells from four or five experiments. In all kinetochore intensity graphs, each dot represents the mean kinetochore intensity of a cell, and the violin plots shows the distribution of intensities between cells. The thick vertical lines represent a 95% CI around the median, which can be used for statistical comparison of multiple time points/treatments by eye (see Materials and methods). Timelines indicate treatment regimen before fixation.

PLK1 activity increases on the BUB complex in BUBR1$^{\Delta PP2A}$ cells, and this is able to support MELT phosphorylation and SAC signaling without affecting MPS1 activity toward another key SAC substrate, MAD1.

Collectively, these data suggest that the BUB–PLK1 complex functions as an autonomous kinase module that can bind to phosphorylated MELT motifs and catalyze further MELT phosphorylation to locally amplify SAC signaling (Fig. 3 F). Multiple MELT repeats are active on each KNL1 molecule (Vleugel et al., 2015b; Vleugel et al., 2013; Zhang et al., 2014) and multiple copies of KNL1 are present per kinetochore (Suzuki et al., 2015). This creates >1,000 active MELT repeats on each human kinetochore, which an autonomous BUB1–PLK1 module can use to rapidly amplify SAC signaling downstream of MPS1. Although the benefit of this is likely to be rapid switch-like activation of the SAC, it is important to antagonize this auto-catalytic module to prevent SAC signaling from locking into a constitutive on state. This is clearly an important role of kinetochore phosphatases because in the absence of either KNL1-PP1 or PP2A-B56, PLK1 recruitment is enhanced (Fig. 1), and MELT phosphorylation and mitotic arrest can be maintained when MPS1 is strongly inhibited (Figs. 2 and 3). This raises the important question of whether PLK1 removal from the BUB complex is the only critical role for these phosphatases in the SAC, or whether they are additionally needed to dephosphorylate the MELT repeats, as previously assumed (Espert et al., 2014; Espeut et al., 2012; Meadows et al., 2011; Nijenhuis et al., 2014; Rosenberg et al., 2011). There is an ~5–10-min delay in MELT dephosphorylation between WT and KNL1$^{\Delta PP1}$/BUBR1$^{\Delta PP2A}$ cells when MPS1 and PLK1 are inhibited together (blue versus red symbols in Fig. 2 C). This could indicate a role for PP1/PP2A in MELT dephosphorylation, or alternatively, it may reflect the time it takes for BI-2536 to penetrate cells and inhibit PLK1, since both inhibitors were added together at time point zero in this assay. Therefore, we next sought to dissect if localized PP1 or PP2A-B56 contribute directly to MELT dephosphorylation.

**KNL1-MELT dephosphorylation can still occur normally when kinetochore PP1 and PP2A activities are strongly inhibited**
PLK1 inhibition for 30 min was sufficient to reduce basal pMELT in KNL1$^{\Delta PP1}$ and BUBR1$^{\Delta PP2A}$ cells to levels comparable with WT cells (Figs. 4 A and S4 A). Therefore, we next examined MELT dephosphorylation rates when MPS1 was inhibited immediately after this 30-min inhibition of PLK1. Fig. 4 B and Fig. S4, B and C, show that the MELT motifs are dephosphorylated with very similar kinetics in this assay, irrespective of whether PP1 or PP2A-B56 kinetochore recruitment is inhibited. This was particularly surprising, because it implies that neither PP1 nor

PP2A is essential for dephosphorylating the MELT repeats. This could reflect redundancy between the two phosphatases; therefore, we attempted to remove both phosphatases from kinetochores by combining PP2A-B56 and PP1 knockdown (Fig. 4 C; and Fig. S4, D–F) or performing PP2A-B56 knockdown in KNL1$^{\Delta PP1}$ cells (Fig. 4 D). However, in both of these situations, MELT dephosphorylation was indistinguishable from WT cells if MPS1 and PLK1 were inhibited together. We also pretreated cells with calyculin A, a very potent inhibitor of all PP1 and PP2A phosphatases (IC50 values 1–2 nM; Ishihara et al., 1989). PP1/PP2A were effectively inhibited by either a short 5-min pretreatment with 50 nM calyculin A or a longer 15-min pretreatment with 25 nM calyculin A, because subsequent Aurora B inhibition was unable to translocate RepoMan onto chromatin (an event that depends on the dephosphorylation of RepoMan-pS893 by localized PP1 or PP2A-B56; Fig. S5, A–D; Qian et al., 2013). Note that these protocols pushed the level of PP1/PP2A inhibition to the limit before the extent of cell rounding and toxicity became too high. The toxic effect of PP1/PP2A inhibition reflects the fact that these enzymes are central to most signaling networks, and PP2A inhibition alone causes approximately half of the phospho-proteome to change significantly (Kauko et al., 2020). Notwithstanding these pleotropic effects, calyculin A is still able to inhibit MELT dephosphorylation and BUB complex removal following MPS1 inhibition alone, as expected (Fig. 4 E; Fig. S5, E and F; and Fig. S6, A–C). Importantly, even in this situation, the MELT motifs are still dephosphorylated, and the BUB complex is removed with fast kinetics when MPS1 and PLK1 are inhibited together. Note that there is a very slight delay when PLK1 and MPS1 inhibitors are applied together in this assay, but this delay is abolished by a 30-min pretreatment with the PLK1 inhibitor BI-2536, which is consistent with a similar effect seen in BUBR1$^{\Delta PP2A}$ and KNL1$^{\Delta PP1}$ cells (compare Figs. 2 C and 4 B). As a final test, we combined genetic and pharmacological inhibition by pretreating BUBR1$^{\Delta PP2A}$ cells with calyculin A. Even under these very stringent conditions, the MELT motifs are effectively dephosphorylated when MPS1 and PLK1 are inhibited together (Figs. 4 F and S6 D). There is a slight 5-min delay in dephosphorylation, and the baseline does not drop completely to zero (after 10-min inhibition median, pMELT levels reduce by 97%, from 1 to 0.03, in BUBR1-WT cells, and the reduction is 91%, from 1.72 to 0.16, in BUBR1$^{\Delta PP2A}$ cells containing calyculin A; Fig. 5 F). These small differences could reflect modest effects of PP2A on MELT dephosphorylation, but given the strength of PP2A inhibition under these conditions, and the fact that 43% of phospho-peptides are regulated by knockdown of PP2A subunits alone (Kauko et al., 2020), we prefer the interpretation that these effects are either off-target or due to elevated

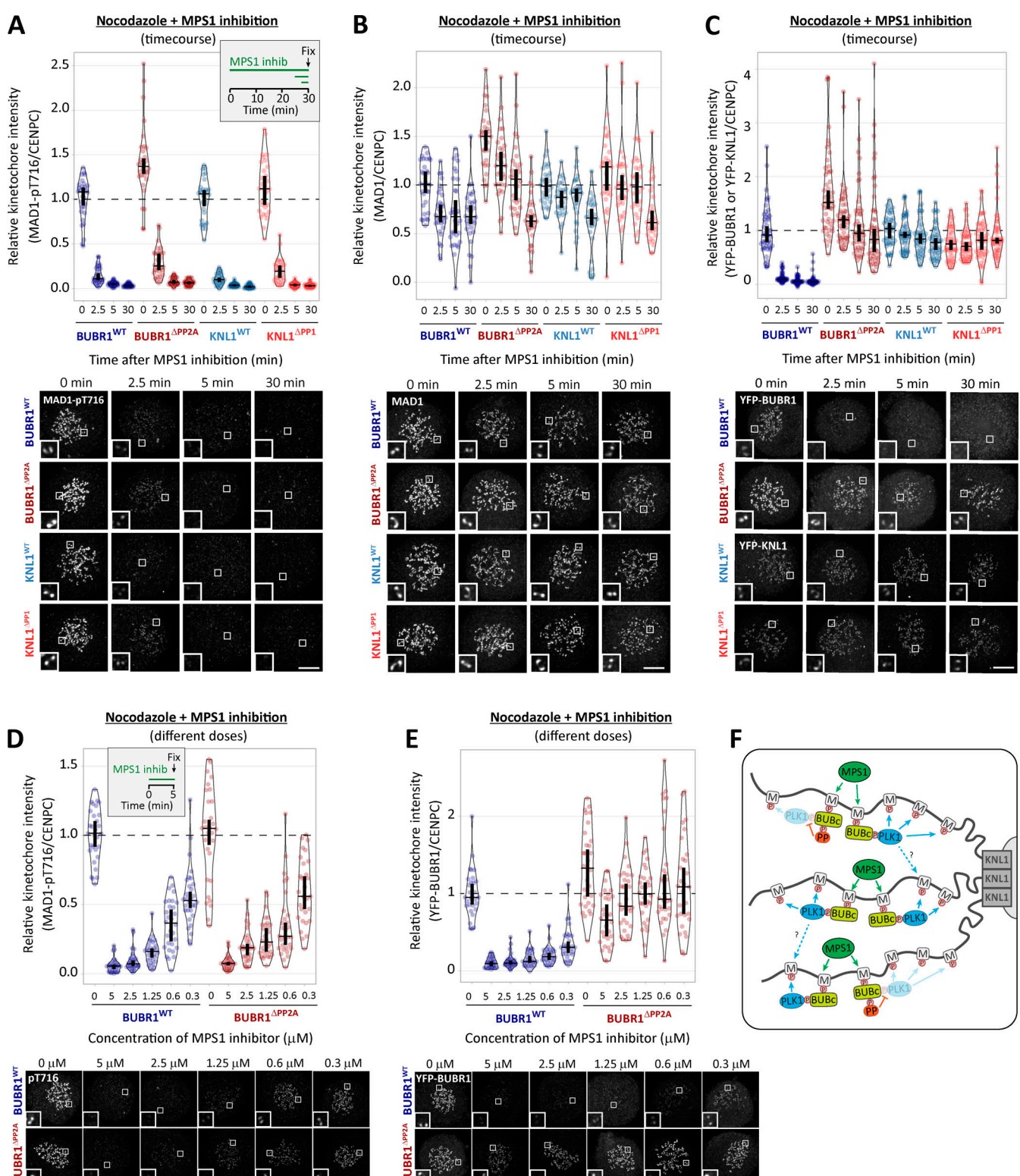

Figure 3. **PP1 and PP2A-B56 removal from kinetochores does not affect dephosphorylation of MAD1-pT716 following MPS1 inhibition. (A)** Effects of phosphatase-binding mutants on MAD1-pT716 dephosphorylation in nocodazole-arrested cells treated with MPS1 inhibitor AZ-3146 (2.5 μM), as indicated in the timeline. Graph displays kinetochore intensities of 30 cells per condition from three experimental repeats. **(B and C)** Total MAD1 (B) and YFP (C) levels in phosphatase-binding mutants treated as in A. B displays kinetochore intensities of 30 cells per condition from three experimental repeats, and C is derived from 60 cells, six experiments. **(D and E)** MAD1-pT716 (D) and YFP-BUBR1 (E) kinetochore levels after 5 min of MPS1 inhibition with different concentrations of AZ-3146 (0.3–5 μM) in nocodazole-arrested cells. Graphs display kinetochore intensities of 30 cells, three experiments. **(F)** Schematic model to illustrate how PLK1 recruitment to multiple MELT motifs can amplify the SAC signal. M, MELT motif; BUBc, BUB complex; PP, PP2A-B56. In all kinetochore intensity graphs, each dot represents the mean kinetochore intensity of a cell, and the violin plots shows the distribution of mean intensities between cells. The thick vertical

lines represent a 95% CI around the median, which can be used for statistical comparison of multiple time point/treatments by eye (see Materials and methods). Timelines indicate treatment regimen before fixation. MG132 was included in combination with MPS1 inhibitor in every case to prevent mitotic exit. Representative images were chosen that most closely resemble the mean values in the quantifications. The insets show magnifications of the outlined regions. Scale bars, 5 µm. Inset size, 1.5 µm.

kinase activity. For example, PLK1 and/or MPS1 activity may be increased—both of these are known targets of PP2A (Hayward et al., 2019; Kim et al., 2019; Wang et al., 2015)—or other phospho-epitopes may be elevated that react with the pMELT antibody.

In summary, these data imply that the primary role of PP1 and PP2A is to remove PLK1 from the BUB complex. This is important because BUB1-bound PLK1 can otherwise prime its own recruitment to maintain a key aspect of the SAC in an autonomous manner (the final model is presented in Fig. 4 G). We can find no evidence that either PP1 or PP2A-B56 are directly required for MELT dephosphorylation, despite multiple attempts to reduce activities as low as possible using a combination of phosphatase inhibition, knockdown, and/or binding motif mutants. We acknowledge that phosphatase inhibition is never absolute under any of these conditions; therefore, it is at least conceivable that PP1 and/or PP2A are so efficient at MELT dephosphorylation that we were unable to detect appreciable effects by any of these treatments. However, we favor the interpretation that an additional phosphatase, perhaps with unregulated basal activity, controls MELT dephosphorylation. We therefore speculate that the regulated phosphatases (PP1/PP2A) are primarily responsible for reducing kinase activity below a threshold required to allow basal phosphatase activity to predominate and dephosphorylate the specific substrates (Fig. S7). One advantage of using a constitutive basal phosphatase in this situation is that it can reverse different signals in a synchronous manner, irrespective of their positions within the kinetochore. This may be difficult to achieve if the executioner phosphatase is a regulated one that binds to a defined location within one protein.

### The PLK1 binding motif is conserved in metazoan MADBUB homologues

The role of kinetochore phosphatases in MELT dephosphorylation and SAC silencing has been extensively documented in different species (Benzi et al., 2020; Espert et al., 2014; Espeut et al., 2012; London et al., 2012; Meadows et al., 2011; Nijenhuis et al., 2014; Pinsky et al., 2009; Rosenberg et al., 2011; Roy et al., 2019; Smith et al., 2019; Vallardi et al., 2019; Vanoosthuyse and Hardwick, 2009). If removal of PLK1 from the BUB complex is a common SAC silencing event in eukaryotes, then one would expect to see coevolution between the phosphatase binding motifs and PLK1 recruitment sites on BUBR1 and/or BUB1. BUBR1/MAD3 and BUB1 are paralogues that arose following the duplication and subfunctionalization of the ancestral MADBUB gene (Tromer et al., 2016). We annotated a list of 152 published eukaryotic MADBUB homologues (Tromer et al., 2016) to display the positions of all consensus Polo binding motifs (PBMs; Ser-Ser/Thr-Pro; Fig. S8; Elia et al., 2003). Note that this binding motif, hereafter called PBM, can only bind PLK1 if accessible and

phosphorylated on the middle Ser/Thr residue; therefore, we excluded any putative PBMs present in established function domains/motifs.

The short linear motif (SLiM) that binds PP2A-B56, LxxIxE (also known as KARD in BUBR1; Hertz et al., 2016; Suijkerbuijk et al., 2012), is present primarily in opisthokonts, which includes fungi and metazoa (Fig. S8). This contrasts with the RVSF SLiM, which binds PP1 to KNL1 and is present throughout the eukaryotic tree of life (Kops et al., 2020; Tromer et al., 2015; van Hooff et al., 2017; Fig. S8). PBMs are predicted to occur every 1 in 400 amino acids by chance; therefore, not surprisingly, they are also present on a wide variety of eukaryotic MADBUBs (88/152 MADBUB homologues). However, there is an enrichment for PBMs within MADBUBs that contain a KARD motif (56/63) and, in particular, in metazoan lineages (35/37; Fig. S8). An unbiased alignment of all eukaryotic PBMs by position, with respect to the KARD, demonstrates a clear enrichment within the regions surrounding, and often just before, the KARD (Fig. S9 A). Sequence alignment of all putative PBMs highlights a conserved motif in a short stretch of ~100 amino acids immediately before the KARD, particularly in metazoan lineages (Fig. S9, B and C). We propose that these are functional PBMs, and the entire metazoan alignment is illustrated in Fig. 5, A–C.

This first notable feature of the PBM is its striking colocalization with the KARD in the same region of MADBUB throughout metazoa (Fig. 5 A and Fig. S9, A–C). The relative activity of PP2A-B56 is strongly influenced by its position with respect to its substrates, and the presence of Pro +1 relative to pSer/Thr is a negative determinant for PP2A-B56 activity (Kruse et al., 2020). Therefore, one interpretation is that the KARD must be close enough to enable dephosphorylation of the PBMs, which are otherwise predicted to be "poor" substrates. In relation to this, it is important to point out that although PP2A-B56 has low activity against Cdk1 sites (Kruse et al., 2020), this does not mean that this phosphatase does not regulate these sites during mitosis. In fact, quite the opposite; it may simply ensure that the bulk of Cdk1 phosphorylation sites are unaffected by constitutive PP2A-B56 activity, while then allowing a subset of these sites to be dephosphorylated in a regulated manner by SLiM-mediated colocalization. An alternative interpretation for the PBM-KARD colocalization is that PLK1 must be close to the KARD to allow subsequent phosphorylation of this motif. In support of this hypothesis, the KARD contains a well-conserved PLK1 phosphorylation site ([E/D]x[S/T]; Alexander et al., 2011; Grosstessner-Hain et al., 2011; Nakajima et al., 2003) in a position that is known to increase PP2A-B56 affinity when phosphorylated (LxxIxEx[S/T]; present in 55/61 eukaryotic and 36/37 metazoan homologues with an LxxIxE motif: Fig. 5, A and C; and Fig. S8; Kruse et al., 2013; Wang et al., 2016a; Wang et al., 2016b). We hypothesize that both of these factors are important, and that the juxtapositioning of the PLK1 and PP2A-binding motifs

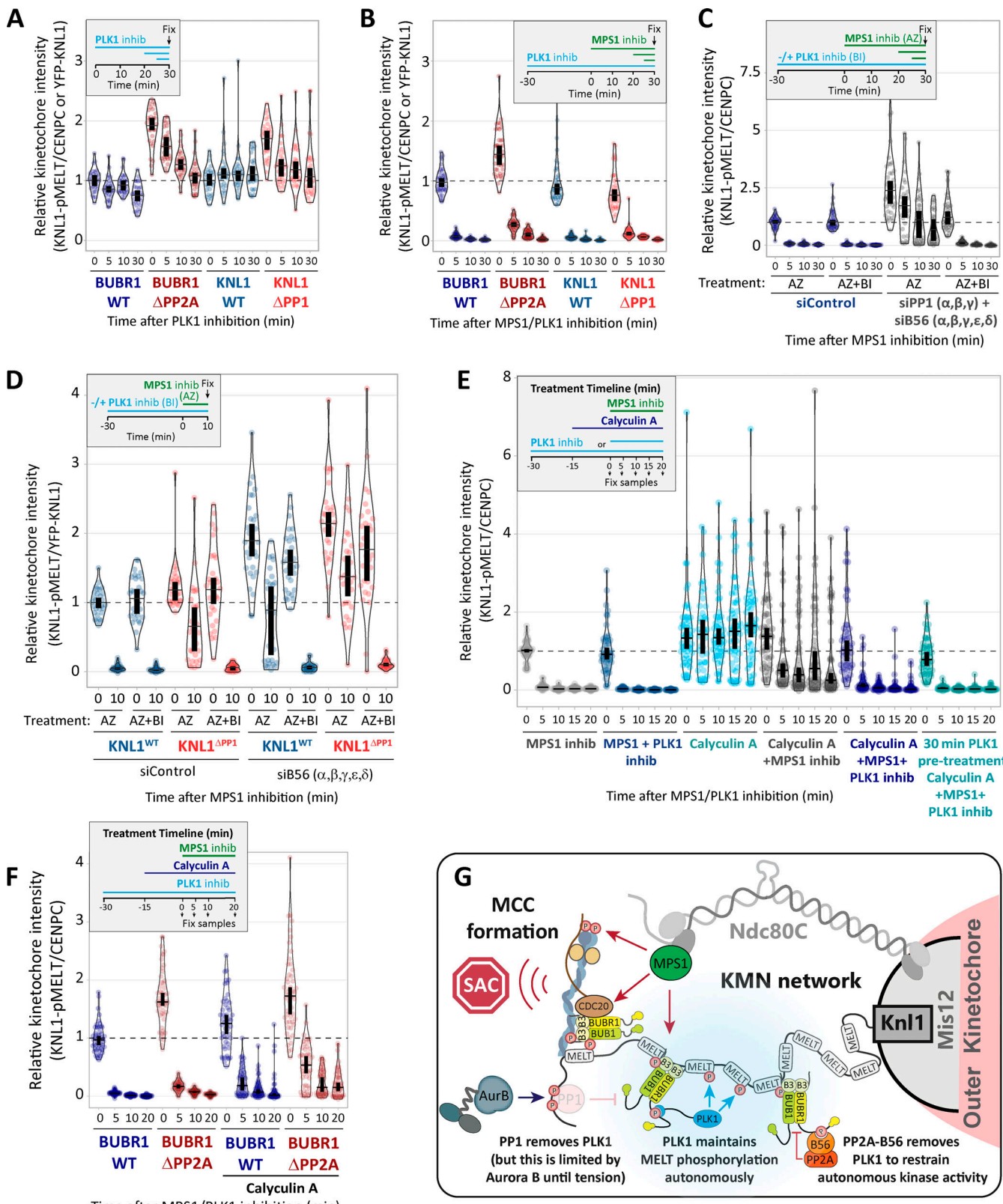

**Figure 4. PP1/PP2A inhibition cannot prevent MELT dephosphorylation if MPS1 and PLK1 are inhibited together. (A and B)** Effects of phosphatase-binding mutants on KNL1-MELT dephosphorylation in nocodazole-arrested cells treated with PLK1 inhibitor BI-2536 (100 nM) alone (A) or in combination with MPS1 inhibitor AZ-3146 (2.5 µM; B), as indicated in the timelines. Intensities are relative to CENP-C in BUBR1 cells or YFP-KNL1 in KNL1 cells, and all BUBR1 and KNL1 intensities are normalized to their respective WT, 0 min time point. **(C and D)** KNL1-MELT phosphorylation levels following combined siRNA-mediated knockdown of all PP1 and B56 isoforms (C) or all B56 isoforms in KNL1^WT/ΔPP1 cells (D). The quantifications are from nocodazole-arrested cells treated with MPS1 inhibitor AZ-3146 (2.5 µM) alone or in combination with PLK1 inhibitor BI-2536 (100 nM), as indicated. Representative images are displayed in Fig. S4, D

and G. **(E)** KNL1-MELT dephosphorylation in Hela FRT cells arrested in nocodazole treated with kinase inhibitors in the presence or absence of the PP1/PP2A phosphatase inhibitor calyculin A (25 nM), as indicated. Representative images are displayed in Fig. S4 H. **(F)** Effects of PP2A-binding mutants in combination with the PP1/PP2A phosphatase inhibitor calyculin A (25 nM) on pMELT dephosphorylation in nocodazole-arrested cells treated with PLK1 (100 nM BI-2536) and MPS1 (2.5 µM AZ-3146) inhibitors, as indicated in the timeline. Representative images are displayed in Fig. S4 I. In all kinetochore intensity graphs, each dot represents the mean kinetochore intensity of a cell, and the violin plots shows the distribution of mean intensities between cells. The thick vertical lines represent a 95% CI around the median, which can be used for statistical comparison of multiple time point/treatments by eye (see Materials and methods). A–D derived from 30–40 cells per condition, three or four experiments. E and F display 40–50 cells, four or five experiments. Timelines indicate treatment regimen before fixation. MG132 was included in whenever MPS1 was inhibited to prevent mitotic exit. **(G)** Schematic model to show how kinetochore phosphatases restrain (PP2A) or extinguish (PP1) autonomous PLK1 activity to control the SAC. MCC, mitotic checkpoint complex.

---

has likely been conserved throughout evolution to allow PLK1/PP2A-B56 cross-regulation on the BUB complex, the full implications of which will be discussed later. It is important to just note that PP2A-B56 interaction within BUBR1 can also regulate phosphorylation within the PBM on BUB1 (BUBR1$^{\Delta PP2A}$ increases BUB1-pT609 in Fig. 1 C), implying that these motifs are close enough to allow cross-regulation. This cross-regulation is also likely to be a conserved feature of vertebrate MAD/BUB paralogues, because four of the annotated vertebrate species have a MAD with a KARD and a BUB that has lost the KARD (*Anolis carolinensis* [lizard], *Ficedula albicollis* [bird], *Callorhinchus milii* [shark], and *Lepisosteus oculatus* [fish]; Fig. S8), and in all of these cases, the BUB has maintained a conserved PBM (Fig. S9 D).

The second notable feature of the PBMs is a strict preference for Thr over Ser in the Cdk1 phosphorylation position (Ser-Thr-Pro in 34/35), which is not observed in other possible PBMs elsewhere in MADBUB (Fig. S9, A–C). A Thr in this position is not a general feature of other PBMs (Lowery et al., 2007) or Cdk1 phosphorylation sites (Holt et al., 2009), but it is thought to confer enhanced activity toward PP2A, in comparison with phospho-Ser (Cundell et al., 2016; Godfrey et al., 2017; Hein et al., 2017; McCloy et al., 2015). We therefore hypothesized that this may enhance activity toward the PBM motif and thereby allow PP2A to antagonize PLK1 recruitment more efficiently. However, Thr-to-Ser conversions in either BUB1 (T609S) or BUBR1 (T620S) did not elevate PLK1 or pMELT levels, either at baseline or after MPS1 inhibition (Fig. S10, A–F). Rather, BUB1-T609S mutation caused a slight reduction in baseline kinetochore PLK1 and pMELT levels and weakened the SAC response (Fig. S10, D–H). We were not able to measure phosphorylation status directly in these mutants, therefore we cannot assess if phosphorylation was increased, as predicted. However, given that PLK1 levels are not increased, but instead slightly decreased, we conclude that Thr is required in the PBM to optimize PLK1 binding (and perhaps turnover) to allow efficient SAC signaling.

A final striking feature of the core PBM is the conserved stretch of hydrophobic residues before and immediately after the Ser-Thr-Pro motif. PLK1 has recently been shown to use a conserved hydrophobic pocket that lies adjacent to the phospho-substrate binding groove to enhance substrate binding affinity (Sharma et al., 2019). Although this pocket is often found in a closed conformation, it also exhibits a variety of crystal packing interactions with hydrophobic residues, demonstrating a high degree of flexibility (Śledź et al., 2011). Two examples of natural phospho-dependent binders that use this pocket are CENP-U/PBIP1, which interacts via Phe71 ($F_{71}$DPPLHS[pT]A; Kang et al., 2006; Śledź et al., 2011), and Partner of Numb (PON), which uses

Phe60 (CFTNAAF$_{60}$SS[pT]P) to bind to the pocket and lock an interaction between two adjacent PBDs, resulting in PLK1 dimerization and activation (Zhu et al., 2016). Interestingly, the CENP-U interaction also appears to dimerize PLK1 (Musacchio, A., personal communication). Therefore, we propose that the conserved hydrophobic amino acids within the MADBUB PBM act to stabilize PLK1 binding, and perhaps additionally to allow PLK1 dimerization and activation at kinetochores. In support of this idea, targeting the hydrophobic pocket on PLK1 directly, using mutations or small molecule inhibition, causes reduced PLK1 recruitment to kinetochores and a mitotic arrest associated with misaligned chromosomes (Sharma et al., 2019). We hypothesize that impaired interaction of PLK1 with the BUB complex is at least partially responsible for these phenotypes. A final parallel between MADBUB and CENP-U/PON is the presence of putative PLK1-regulated PP2A-B56 interaction motifs upstream of the PBMs (Fig. 5 D). It will be important to test whether PP2A-B56 and PLK1 engage in similar cross-regulation at CENP-U/PON, as observed here at the BUB complex, and to examine whether PLK1 dimerizes on the BUB complex, as it does on CENP-U/PON. If similar features have evolved separately in different complexes, then this would imply functional conservation of a PLK1-PP2A module that can be repurposed on different complexes to carry out distinct functions. In relation to this, it is important to point out that PLK1-PP2A cross-regulation on the BUB complex may also regulate other processes as well as the SAC. In particular, kinetochore-microtubule attachments are inhibited by PP2A-B56 loss but rescued by subsequent inhibition of PLK1 (Foley et al., 2011); therefore, we speculate that PLK1-PP2A cross-regulation on the BUB complex may also impact the microtubule attachment process, perhaps by regulating kinetochore Aurora B activity. This will be an important future area of investigation because it may help to explain why this bifunctional kinase-phosphatase module is so well conserved.

Notwithstanding potential effects on other processes, a major role of PLK1-PP2A cross-regulation on the BUB complex is to control SAC activation and silencing. We demonstrate here that PP2A-B56 is required to antagonize BUB1–PLK1 binding and thereby suppress an autonomous kinase module that can otherwise enhance SAC signaling (pMELT→BUB:PLK1→pMELT; Fig. 5 E, P1). We showed recently that PP2A-B56 is unable to silence the SAC on its own, primarily because it binds to BUBR1 in a phospho-dependent manner (Smith et al., 2019). Considering that PLK1 removal is the crucial event in SAC silencing and PLK1 enhances PP2A-B56 recruitment (Elowe et al., 2007; Kruse et al., 2013; Suijkerbuijk et al., 2012; Xu et al., 2013), we speculate that this negative feedback loop operates on the BUB complex

and limits the ability of PP2A-B56 to fully silence the SAC (PP2A⊣PLK1→PP2A; Fig. 5 E, N1). To test this model, it will be important to evaluate how local PLK1 levels affect PP2A-B56 recruitment to the BUB complex. Homeostatic SAC regulation by phospho-dependent PP2A may be important to preserve the BUB complex at kinetochores when microtubules attach in a mono-riented fashion and MPS1 is removed/inhibited (Aravamudhan et al., 2015; Hiruma et al., 2015; Ji et al., 2015). The benefit of preserving the SAC platform in this situation is that SAC signaling can then be rapidly reestablished if microtubules subsequently detach. However, if the correct bioriented state is achieved and the kinetochore comes under tension, then the PP1 arm is engaged (Liu et al., 2010), which, crucially, is not restricted by negative feedback (Fig. 5 E). In fact, we speculate that this tension-dependent switch is reinforced by positive feedback instead, because removal of PLK1 and the BUB complex is predicted to reduce centromeric Aurora B activity (Combes et al., 2017; Hindriksen et al., 2017), thereby enhancing PP1-KNL1 recruitment (PP1⊣BUB:PLK1→ Aurora B⊣PP1; not depicted in Fig. 5 E). An important feature of this model is that PP1 and PP2A-B56 exhibit no intrinsic specificities, but rather, they produce different network effects due to their opposite phospho-dependencies; as previously illustrated with the aid of mathematical modeling (Smith et al., 2019). It will be important to update this model to include the PLK1 regulation demonstrated here; a crucial aspect of which will be to determine whether PP1 or PP2A-B56 activities against the BUBR1-pT620 and/or BUB1-pT609 sites are equivalent.

In summary, by interrogating the specific role of PP1 and PP2A-B56 at kinetochores, we arrive at the conclusion that phosphatase regulation is critical, but primarily to restrain and extinguish localized kinase activity. It will be important in the future to verify whether this is their only critical role in the SAC, and if so, what phosphatase(s) dephosphorylate the MELT motifs. It is interesting to note that other organisms have evolved different circuits to control the SAC, but even in these cases, phosphatase regulation appears to focus back to limit kinase activity. In flies, the phospho-Thr residues on the KNL1-MELT motifs are not conserved, and MPS1/PLK1 activities are not required to recruit the BUB complex to kinetochores (Conde et al., 2013; Schittenhelm et al., 2009). However, in this situation, PP1 binds directly to MPS1 to inhibit its activity and silence the downstream SAC signal (Moura et al., 2017). Therefore, the use of regulated phosphatases to silence local kinase activity may be a conserved feature of SAC signaling.

## Materials and methods
### Cell culture
All cell lines were derived from HeLa Flp-in cells (a gift from S. Taylor, University of Manchester, Manchester, UK; Tighe et al., 2008) and authenticated by STR profiling (Eurofins). The cells were cultured in DMEM supplemented with 9% FBS (Gibco) and 50 µg/ml penicillin/streptomycin (Thermo Fisher). During fluorescence time-lapse analysis, cells were cultured in DMEM (no phenol red) supplemented with 9% FBS and 50 µg/ml penicillin/

streptomycin. Cells were screened every 4–8 wk to ensure they were mycoplasma free.

### Plasmids and cloning
pcDNA5-YFP-BUBR1$^{WT}$ expressing an siRNA-resistant and N-terminally YFP-tagged WT BUBR1 and pcDNA5-YFP-BUBR1$^{\Delta PP2A}$ (also called BUBR1$^{\Delta KARD}$) lacking amino acids 663–680 were previously described (Nijenhuis et al., 2014). These vectors were used to generate pcDNA5-YFP-BUBR1$^{T620A}$, pcDNA5-YFP-BUBR1$^{\Delta PP2A-T620A}$, and pcDNA5-YFP-BUBR1$^{T620S}$ by site-directed mutagenesis using the following primers (Sigma-Aldrich): T620A (forward 5′-TGCCAGAGCAGCTCGTTTT TGTATCCGCTCCTTTTCATGAGATAATGTCCTTG-3′, reverse 5′-CAAGGACATTATCTCATGAAAAGGAGCGGATACAAAACGAGC TGCTCTGGCA-3′) and T620S (forward 5′-TGCCAGAGCAGCTCG TTTTGTATCCTCTCCTTTTCATGAGATAATGTCCTTG-3′, reverse 5′-CAAGGACATTATCTCATGAAAAGGAGAGGATACAAA ACGAGCTGCTCTGGCA-3′). pcDNA5-YFP-BUB1$^{WT}$ expressing an siRNA-resistant and N-terminally YFP-tagged WT BUB1 was described previously (Vleugel et al., 2015a) and used to generate pcDNA5-YFP-BUB1$^{T609A}$ and pcDNA5-YFP-BUB1$^{T609S}$ by site-directed mutagenesis using the following primers (Sigma-Aldrich): T609A (forward 5′-CACATCTGCTGCACAACTTGC GTCTGCACCATTCCACAAGCTTCCAGTGG-3′, reverse 5′-CCA CTGGAAGCTTGTGGAATGGTGCAGACGCAAGTTGTGCAGCAG ATGTG-3′) and T609S (forward 5′-CACATCTGCTGCACAACT TGCGTCTTCACCATTCCACAAGCTTCCAGTGG-3′, reverse 5′-CCACTGGAAGCTTGTGGAATGGTGAAGACGCAAGTTGTGCAG CAGATGTG-3′). pcDNA5-mTurquoise2(Turq2)-BUB1$^{WT}$ and pcDNA5-Turq2-BUB1$^{T609A}$ were created by restriction cloning using Acc65I and BstBI to replace the YFP originally present in pcDNA5-YFP-BUB1$^{WT}$ and pcDNA5-YFP-BUB1$^{T609A}$ (Turq2 subcloned from pcDNA4-mTurq2-BUBR1$^{WT}$; Smith et al., 2019). pcDNA5-YFP-KNL1$^{WT}$ expressing an siRNA-resistant and N-terminally YFP-tagged WT KNL1 and pcDNA5-YFP-KNL1$^{\Delta PP1}$ (with RVSF at amino acids 58–61 mutated to AAAA, also called KNL1$^{4A}$) were previously described (Smith et al., 2019). pcDNA5-YFP-RepoMan$^{WT}$ expressing an siRNA-resistant and N-terminally YFP-tagged WT RepoMan was subcloned by restriction cloning (with NotI/ApaI) to replace BUBR1 in pcDNA5-YFP-BUBR1$^{WT}$ using a synthesized DNA fragment containing the human RepoMan sequence flanked by NotI/ApaI restriction sites (Baseclear). The siRNA-resistant mutations were TGACcG AtCTaACtcGgAA (small letters indicate silent base changes). All plasmids were fully sequenced to verify the transgene was correct.

### Gene expression
HeLa Flp-in cells were used to stably express doxycycline-inducible constructs after transfection with the relevant pcDNA5/FRT/TO vector and the Flp recombinase pOG44 (Thermo Fisher). Cells were then selected for stable integrants at the FRT locus using 200 µg/ml hygromycin B (Santa Cruz Biotechnology) for at least 2 wk. In experiments requiring two transgenes in Fig. 1 E; Fig. 2, G and H; and Fig. S3, F–I, YFP-BUBR1-WT, ΔPP2A, ΔPP2A-T620A, or T620A was transiently transfected into cells stably expressing doxycycline-inducible

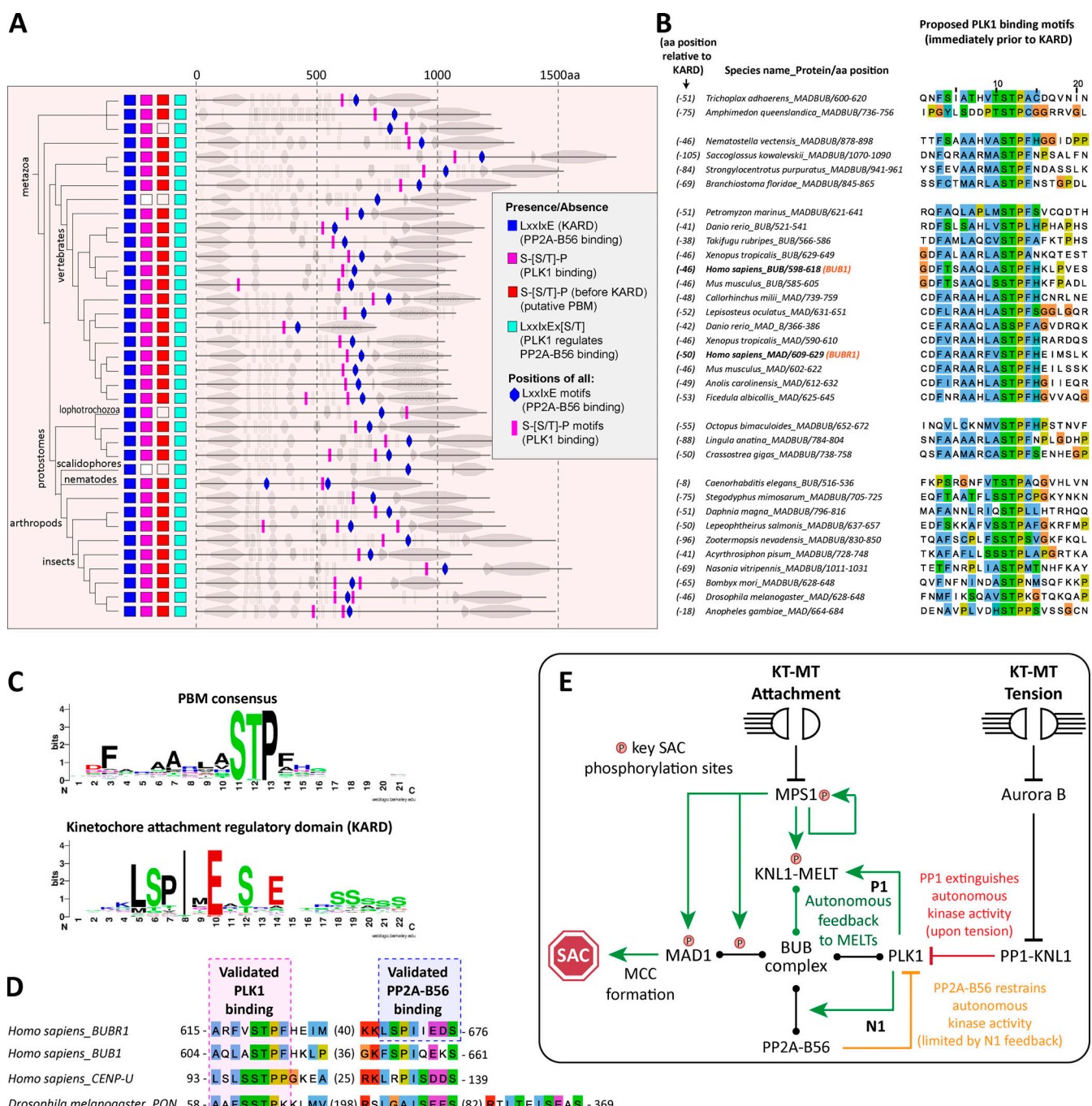

Figure 5. **Evolution of PBMs and PP2A-B56 binding motifs (KARD) in MADBUB homologues. (A)** Annotation of PBMs (Ser-Ser/Thr-Pro) and PP2A-B56 binding motifs (LxxIxE, KARD) positions within metazoan MADBUB homologues that contain a KARD. Adapted from Tromer et al. (2016); see Fig. S8 and Data S1 for complete list of 152 eukaryotic MADBUB homologues. **(B)** Alignment of proposed PBM (located immediately before the KARD) in the species represented in A. **(C)** Consensus sequence of PBMs listed in B (for consensus motif of other PBMs within eukaryotic MADBUB homologues, see Fig. S9) and the KARD within all eukaryotic MADBUB homologues. **(D)** Alignment of PBM and KARD in BUBR1, BUB1, PON, and CENP-U. Note: PON is a *Drosophila* gene with no known homologues in humans, which is involved in mitotic asymmetric division during *Drosophila* brain development (Lu et al., 1998). **(E)** Schematic model to illustrate relevant feedback loops involved in SAC activation and silencing (see Results and discussion for explanations). KT-MT, kinetochore-microtubule.

Turq2-BUB1-WT or T609A mutants. Transfection with these YFP-tagged constructs was done 32 h before endogenous gene knockdown (described below) and at least 72 h before fixation. Plasmids were transfected using Fugene HD (Promega) according to the manufacturer's instructions.

**Gene knockdown**

For all experiments in HeLa Flp-in cells, the endogenous mRNA was knocked down and replaced with an siRNA-resistant mutant. The siRNAs used in this study were BUBR1 (5'-AGAUCC UGGCUAACUGUUC-3'), siBUB1 (5'-GAAUGUAAGCGUUCACGA

A-3′), B56α (PPP2R5A: 5′-UGAAUGAACUGGUUGAGUA-3′), B56β (PPP2R5B: 5′-GAACAAUGAGUAUAUCCUA-3′), B56γ (PPP2R5C: 5′-GGAAGAUGAACCAACGUUA-3′), B56δ (PPP2R5D: 5′-UGACUGAGCCGGUAAUUGU-3′), B56ε (PPP2R5E: 5′-GCA CAGCUGGCAUAUUGUA-3′), PP1α (PP1CA: 5′-GUAGAAACCAUA GAUGCGG-3′), PP1β (PP1CB: 5′-ACAUCAGUAGGUCUCAUAA-3′), PP1γ (PP1CC: 5′-GCAUGAUUUGGAUCUUAUA-3′), RepoMan (5′-UGACAGACUUGACCAGAAA-3′), and GAPDH (control siRNA: 5′-GUCAACGGAUUUGGUCGUA-3′). All siRNAs were synthesized with dTdT overhang by Sigma-Aldrich and used at 20 nM final concentration (i.e., the pools for B56 or PP1 knockdown contain 20 nM of each siRNA). Double-stranded interference RNA was used to knock down endogenous KNL1 (sense: 5′-GCA UGUAUCUCUUAAGGAAGAUGAA-3′; antisense: 5′-UUCAUC UUCCUUAAGAGAUACAUGCAU-3′; Integrated DNA Technologies) at a final concentration of 20 nM. All siRNAs/double-stranded interference RNAs were transfected using Lipofectamine RNAiMAX Transfection Reagent (Life Technologies) according to the manufacturer's instructions.

After 16 h of knockdown, DNA synthesis was prevented by addition of thymidine (2 mM; Sigma-Aldrich) for 24 h. Doxycycline (1 μg/ml; Sigma-Aldrich) was used to induce expression of the BUBR1, BUB1, RepoMan, and KNL1 constructs during and following the thymidine block. Cells were then released from thymidine block into media supplemented with doxycycline and nocodazole (3.3 μM; Sigma-Aldrich) for 8.5 h before processing for fixed analysis. In live-cell imaging experiments in Fig. 2, MPS1 and/or PLK1 were inhibited by adding AZ-3146 (2.5 μM; Sigma-Aldrich; Hewitt et al., 2010) and BI-2536 (100 nM; Selleckbio; Steegmaier et al., 2007) shortly before imaging (6–8 h after thymidine release). For MPS1 and PLK1 inhibition in cells analyzed by immunofluorescence, nocodazole and MG132 (10 μM; Sigma-Aldrich) were added first for 30 min (plus BI-2536 if pretreatment was used), followed by a time course of AZ-3146 and/or BI-2536 in media containing nocodazole and MG132.

A high dose of calyculin A (25 nM or 50 nM; LC Laboratories) was used to inhibit all PP1 and PP2A phosphatases (IC50 values 1–2 nM; Ishihara et al., 1989). In these experiments (Fig. 4 E; Fig. S5, E and F; and Fig. S6, A–C), Hela FRT cells (empty FRT locus) were treated with nocodazole for 4 h followed by a 30-min incubation with media containing nocodazole and MG132 (these two drugs were present in all the subsequent steps of these experiments) before treatment as indicated in the timelines. Efficient phosphatase inhibition was confirmed in similar assays (Fig. S5, A–D) by testing the ability of RepoMan to translocate to chromatin after inhibition of Aurora B activity with 2 μM ZM-447439 (Cayman Chemicals; Ditchfield et al., 2003).

To image nocodazole-arrested cells treated with siRNA targeting BUBR1, BUB1, or both (Fig. S1, A–C), HeLa FRT cells were released from thymidine block (40 h after siRNA treatment) for 7 h before arresting at the G2/M boundary with RO-3306 treatment (10 mM; Tocris) for 2 h. Cells were then washed three times and incubated for 15 min with full growth media before addition of MG132 for 30 min to prevent mitotic exit. This is important so that cells enter mitosis in the presence of nocodazole and MG132, which allows the arrest to be maintained and Cyclin B levels to be preserved, even though the SAC is inhibited. Cells were then fixed and stained as described below.

## Immunofluorescence
Cells, plated on High Precision 1.5H 12-mm coverslips (Marienfeld), were fixed with 4% PFA in PBS for 10 min or preextracted (when using BUB1-pT609 or BUBR1-pT620 antibodies and in double mutant experiments) with 0.1% Triton X-100 in PEM (100 mM Pipes, pH 6.8, 1 mM MgCl$_2$, and 5 mM EGTA) for 1 min before addition of 4% PFA for 10 min. In experiments using calyculin A, coverslips were coated with poly-L-lysine (Sigma-Aldrich) to prevent cell loss. After fixation, coverslips were washed with PBS and blocked with 3% BSA in PBS + 0.5% Triton X-100 for 30 min, incubated with primary antibodies overnight at 4°C, washed with PBS, and incubated with secondary antibodies plus DAPI (Thermo Fisher) for an additional 2–4 h at room temperature in the dark. Coverslips were then washed with PBS and mounted on glass slides using ProLong antifade reagent (Molecular Probes). All images were acquired on a DeltaVision Core or Elite system equipped with a heated 37°C chamber, with a 100×/1.40 NA U Plan S Apochromat oil objective using softWoRx software (Applied Precision). Images were acquired at 1 × 1 binning using a CoolSNAP HQ or HQ2 camera (Photometrics) and processed using softWorx software (deconvolution followed by maximum projection) and ImageJ (National Institutes of Health). For experiments involving YFP-expressing cells, mitotic cells were selected based on good expression of YFP at the kinetochore (KNL1) or cytoplasm (BUBR1 cells). In cases where double mutants (YFP and Turquoise 2) were used (Fig. 1 E, Fig. 2 G, and Fig. S3, F–H), cells were selected based on good YFP signal in the kinetochore and cytoplasm, since the chicken anti-GFP antibody used cannot discriminate between the two fluorescent proteins. All immunofluorescence images displayed are maximum intensity projections of deconvolved stacks and were chosen to most closely represent the mean quantified data. Figure panels were creating using Omero (http://openmicroscopy.org).

## Time-lapse analyses
For fluorescence time-lapse imaging, cells were imaged in 24-well plates in DMEM (no phenol red) with a heated 37°C chamber in 5% CO$_2$. Images were taken every 4 min with a 20×/0.4 NA air objective using a Zeiss Axio Observer 7 with a CMOS Orca flash 4.0 camera at 4 × 4 binning. For brightfield imaging, cells were imaged in a 24-well plate in DMEM in a heated chamber (37°C and 5% CO$_2$) with a 10×/0.5 NA air objective using a Hamamatsu ORCA-ER camera at 2 × 2 binning on a Zeiss Axiovert 200M, controlled by Micro-manager software (open source; https://micro-manager.org/) or with a 20×/0.4 NA air objective using a Zeiss Axio Observer 7 as detailed above. Mitotic exit was defined by cells flattening down in the presence of nocodazole and MPS1 inhibitor. In double mutant assays where both recombinant BUBR1 and BUB1 are expressed (Figs. 2 H and S3 I), cells were selected for quantification based on high levels of YFP as an indication of successful transient transfection of BUBR1 constructs.

## Antibodies

All antibodies were diluted in 3% BSA in PBS. The following primary antibodies were used for immunofluorescence imaging (at the final concentration indicated): chicken anti-GFP (ab13970 from Abcam, 1:5,000), guinea pig anti–CENP-C (PD030 from Caltag + Medsystems, 1:5,000), rabbit anti-BUB1 (A300-373A from Bethyl, 1:1,000), mouse anti-BUBR1 (05–898 from Millipore, 1:1,000), rabbit anti-PLK1 (IHC-00071 from Bethyl, 1:1,000), mouse anti-KNL1 (gift from M. Yanagida, Okinawa Institute of Science and Technology Graduate University, Okinawa, Japan, 1:500), rabbit anti-KNL1 (ab70537 from Abcam, 1:500), mouse anti-MAD1 (MABE867 from Millipore, 1:1,000), rabbit anti–MAD1-pT716 (Allan et al., 2020; custom raised by Biomatik, 1:1,000), and rabbit anti-BUBR1 (A300-386A from Bethyl, 1:1,000). The rabbit anti–pMELT-KNL1 antibody is directed against Thr 943 and Thr 1155 of human KNL1 (Nijenhuis et al., 2014; 1:1,000). The rabbit anti–BUB1-p609 antibody was raised against phospho–Thr 609 of human BUB1 using the following peptide C-AQLAS[pT]PFHKLPVES (custom raised by Biomatik, 1:1,000). The rabbit anti–BUBR1-p620 antibody was raised against phospho–Thr 620 of human BUBR1 using the following peptide C-AARFVS[pT]PFHE (custom raised by Moravian, 1:1,000). Secondary antibodies used were highly cross-absorbed goat, anti-chicken Alexa Fluor 488 (A-11039), anti-rabbit Alexa Fluor 568 (A-11036), anti-mouse Alexa Fluor 488 (A-11029), anti-mouse Alexa Fluor 568 (A-11031), or anti–guinea pig Alexa Fluor 647 (A-21450), all used at 1:1,000 (Thermo Fisher).

The following antibodies were used for Western blotting (at the final concentration indicated): rabbit anti-GFP (custom polyclonal, a gift from G. Kops, Hubrecht Institute, Utrecht, Netherlands, 1:5,000), rabbit anti-BUBR1 (A300-386A from Bethyl, 1:1,000), rabbit anti-BUB1 (A300-373A from Bethyl, 1:1,000), rabbit anti–BUB1-p609 (custom raised by Biomatik, 1:1,000), rabbit anti–BUBR1-p620 (custom raised by Moravian, 1:1,000) and mouse anti–α-tubulin (clone B-5-1-2, T5168, Sigma-Aldrich, 1:10,000). Secondary antibodies used were goat anti-mouse IgG HRP conjugate (170–6516 from Bio-Rad, 1:2,000) and goat anti-rabbit IgG HRP conjugate (170–6515 from Bio-Rad, 1:5,000).

## Immunoprecipitation and immunoblotting

Flp-in HeLa cells were treated with thymidine and doxycycline for 24 h, followed by a treatment with nocodazole and doxycycline for 16 h. Mitotic cells were isolated by mitotic shakeoff and incubated with media containing nocodazole and doxycycline with or without calyculin A (50 nM; LC Laboratories) for 20 min. Cells were lysed in lysis buffer (50 mM Tris, pH 7.5, 150 mM NaCl, 0.5% Triton X-100, 1 mM $Na_3VO_4$, 5 mM β-glycerophosphate, 25 mM NaF, 10 nM calyculin A, and complete protease inhibitor containing EDTA; Roche) on ice for 20 min. The lysate was incubated with GFP-Trap magnetic beads (from ChromoTek) for 2 h at 4°C on a rotating wheel in wash buffer (same as lysis buffer, but without Triton X-100) at a 3:2 ratio of wash buffer:lysate. The beads were washed three times with wash buffer, and the sample was eluted according to the protocol from ChromoTek.

## Image quantification and statistical analysis

For quantification of kinetochore protein levels, images of similarly stained experiments were acquired with identical illumination settings and analyzed using an ImageJ macro, as described previously (Saurin et al., 2011). Briefly, the ImageJ macro uses autothreshold to select all kinetochores and all chromosome areas (excluding kinetochores) using the DAPI and anti-kinetochore antibody channels (CENP-C), respectively. This was used to calculate the relative mean kinetochore intensity of various proteins ([kinetochore–chromosome arm intensity (test protein)]/[kinetochore–chromosome arm intensity (CENP-C)]). RepoMan translocation to chromatin was quantified by determining the mean intensity of YFP in the chromatin (defined by autothreshold in the DAPI channel) and cytoplasm (remaining cell excluding the DAPI area) and plotted as a ratio chromatin/cytoplasm signal. Normality of the data distribution was tested using the D'Agostino & Pearson normality test. Two-tailed, nonparametric Mann-Whitney unpaired $t$ tests were performed to compare the means values between experimental groups in immunofluorescence quantifications from Figs. 1, S1, and S2 (using Prism 6 software). For Fig. 2 onward, when multiple time points and treatments are used to compare the difference in dephosphorylation kinetics, the graphs are plotted as violin plots using PlotsOfData (Postma and Goedhart, 2019; https://huygens.science.uva.nl/PlotsOfData/). This allows the spread of data to be accurately visualized along with the 95% CIs (thick vertical bars) calculated around the median (thin horizontal lines). Statistical comparison can then be made by eye between any treatment and time points, because when the vertical bar of one condition does not overlap with one in another condition, the difference between the medians is statistically significant ($P < 0.05$). In KNL1 cells, kinetochore intensities were normalized to YFP-KNL1 to avoid artificial fluctuation of signal resulting from variability in reexpression levels of YFP-KNL1.

## PLK1 binding motifs analysis in eukaryotic MADBUB homologues

The dataset from Tromer et al., 2016 was used and annotated for the presence of PBM (Ser-Ser/Thr-Pro; Elia et al., 2003), excluding any putative motifs present in established function domains/motifs. The full list of annotated and excluded PBMs is included in Data S1. The majority of KNL1 sequences used to determine presence/absence of the KNL1-RVSF were published previously (Kops et al., 2020; Tromer et al., 2015; van Hooff et al., 2017). Any unannotated genomes were searched by blast using KNL1 orthologues from the closest species. The additional KNL1 orthologues found by this approach are included in Data S2. Consensus binding motifs in Figs. 5 C and S9 were created using WebLogo (Crooks et al., 2004; https://weblogo.berkeley.edu/logo.cgi). Protein alignments were generated using Jalview (Waterhouse et al., 2009).

## Online supplemental material

Fig. S1 shows the effect of BUB1/BUBR1 knockdown in PLK1, BUBR1, and BUB1 levels. Fig. S2 shows the specificity of phospho-specific antibodies BUB1-pT609 and BUBR1-pT620 in cells and

the levels of BUB1 and BUBR1 in phosphatase-binding mutants. Fig. S3 shows representative images of the data represented in Fig. 2, KNL1 levels during MPS1/PLK1 inhibition, and mitotic exit in BUBR1$^{T620A}$+BUB1$^{T609A}$ mutants with 2.5 µM AZ-3146. Fig. S4 shows example images of kinetochore quantifications and KNL1 and BUBR1 levels in cells treated as in Fig. 4. Fig. S5 shows kinetochore quantification of BUBR1 and RepoMan after calyculin A treatment with 25 nM and 50 nM doses. Fig. S6 shows kinetochore quantification of KNL1 and KNL1-pMELT after calyculin A treatment with 25 nM and 50 nM doses. Fig. S7 shows a model for how localized processes could use a combination of regulated and unregulated phosphatase activity to synchronously dephosphorylate spatially resolved substrates. Fig. S8 describes the presence or absence of PBMs and PP2A-B56 binding motifs (KARD) within eukaryotic MADBUB homologues and PP1-binding motifs (RVSF) within KNL1 orthologues. Fig. S9 shows sequence alignment of all PBM in eukaryotic MADBUB homologues with respect to position from the KARD. Fig. S10 shows mitotic exit, representative images, and kinetochore quantifications of pMELT and PLK1 in BUBR1-T620S and BUB1-T609S mutants. Data S1 shows the full list of annotated and excluded PBMs in eukaryotic MADBUB homologues. Data S2 shows the KNL1 orthologues from Fig. S8 that are not published in Kops et al. (2020), Tromer et al. (2015), and van Hooff et al. (2017).

## Acknowledgments

We thank staff at the Dundee Imaging Facility and the Genetic Core Services Unit. We also thank Stephen Taylor for providing the HeLa Flp-in cell line and Mitsuhiro Yanagida for antibodies.

This work was funded by a Cancer Research UK Program Foundation Award to A.T. Saurin (C47320/A21229 and C10988/A22566).

The authors declare no competing financial interests.

Author contributions: A.T. Saurin, R.J. Smith, and M.H. Cordeiro conceived the study, designed the experiments, and interpreted the data. R.J. Smith and M.H. Cordeiro performed the experiments. A.T. Saurin supervised the study and wrote the manuscript with input from M.H. Cordeiro.

Submitted: 5 February 2020

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

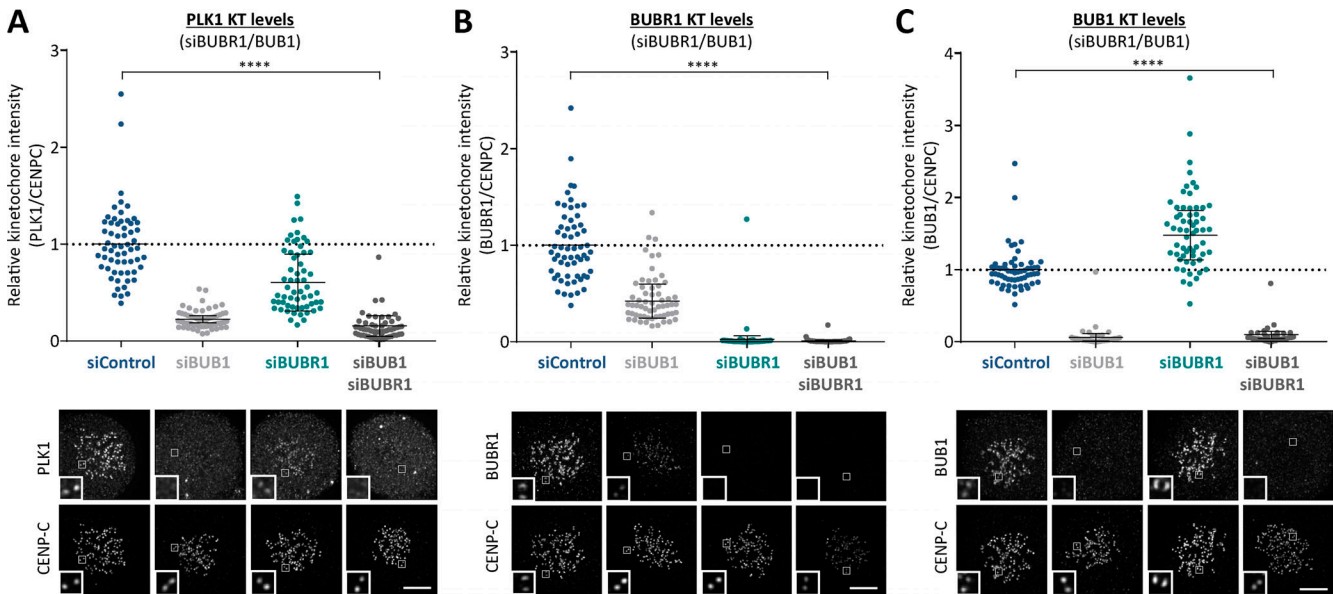

Figure S1. **Effect of BUB/BUBR1 knockdown in PLK1, BUBR1, and BUB1 kinetochore levels (related to Fig. 1). (A–C)** Relative kinetochore levels of PKL1 (A), BUBR1 (B), and BUB1 (C) after siRNA-mediated depletion of BUB1, BUBR1, or both BUB1 and BUBR1 combined in HeLa FRT cells. Graphs show the mean kinetochore intensity of 40–60 cells per condition, four experiments. For all kinetochore intensity graphs, each dot represents a cell, and the error bars display the variation between the experimental repeats (displayed as ±SD of the experimental means). Two-tailed, nonparametric Mann-Whitney unpaired t tests were performed to compare the means values between experimental groups. Example immunofluorescence images were chosen that most closely resemble the mean values in the quantifications. The insets show magnifications of the outlined regions. Scale bars, 5 µm. Inset size, 1.5 µm. ****, P < 0.0001.

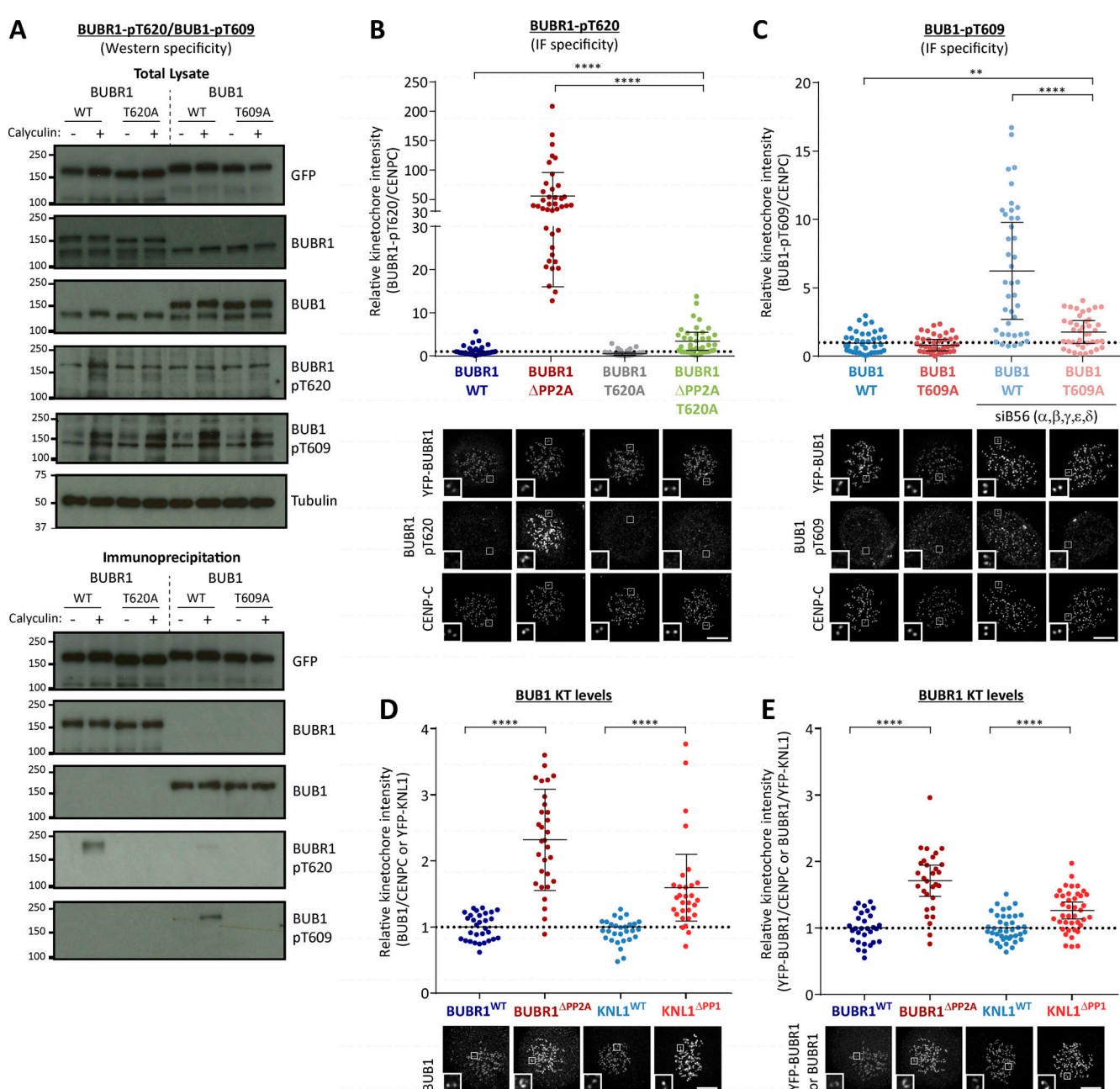

Figure S2. **Testing specificity of phospho-specific antibodies in cells and quantification of kinetochore BUB1 and BUBR1 levels in phosphatase-binding mutants (related to Fig. 1). (A)** Mitotic cells expressing exogenous BUB1 and BUBR1 constructs were treated with/without 50 nM calyculin for 20 min before harvesting for lysis to maximally phosphorylate BUB1/BUBR1. YFP-tagged BUB1 and BUBR1, mutated in the Polo-box binding sites, were immunoprecipitated and blotted with indicated antibodies. Blots from a single experiment were chosen that best represent three independent experiments. **(B and C)** Effect of mutating the Polo-box binding motifs (BUBR1[T620A] and BUB1[T609A]) on antibody detection of BUBR1 pT620 (B) and BUB1 pT609 (C) by immunofluorescence. These experiments were performed in combination with either mutating the phosphatase binding motif on BUBR1 (B) or siRNA-mediated knockdown of all B56 isoforms (C) to elevate BUBR1/BUB1 phosphorylation to maximal levels. The graphs show the mean kinetochore intensity of 40 cells from four experiments. **(D and E)** Kinetochore quantification of BUB1 (D) and BUBR1 (E) levels in phosphatase-binding mutants. D displays 30 cells from three experiments, and E shows 30–40 cells from two or three experiments. For all kinetochore intensity graphs, each dot represents a cell, and the error bars display the variation between the experimental repeats (displayed as ±SD of the experimental means). Two-tailed, nonparametric Mann-Whitney unpaired $t$ tests were performed to compare the means values between experimental groups. Example immunofluorescence images were chosen that most closely resemble the mean values in the quantifications. The insets show magnifications of the outlined regions. Scale bars, 5 µm. Inset size, 1.5 µm. **, $P < 0.01$; ****, $P < 0.0001$.

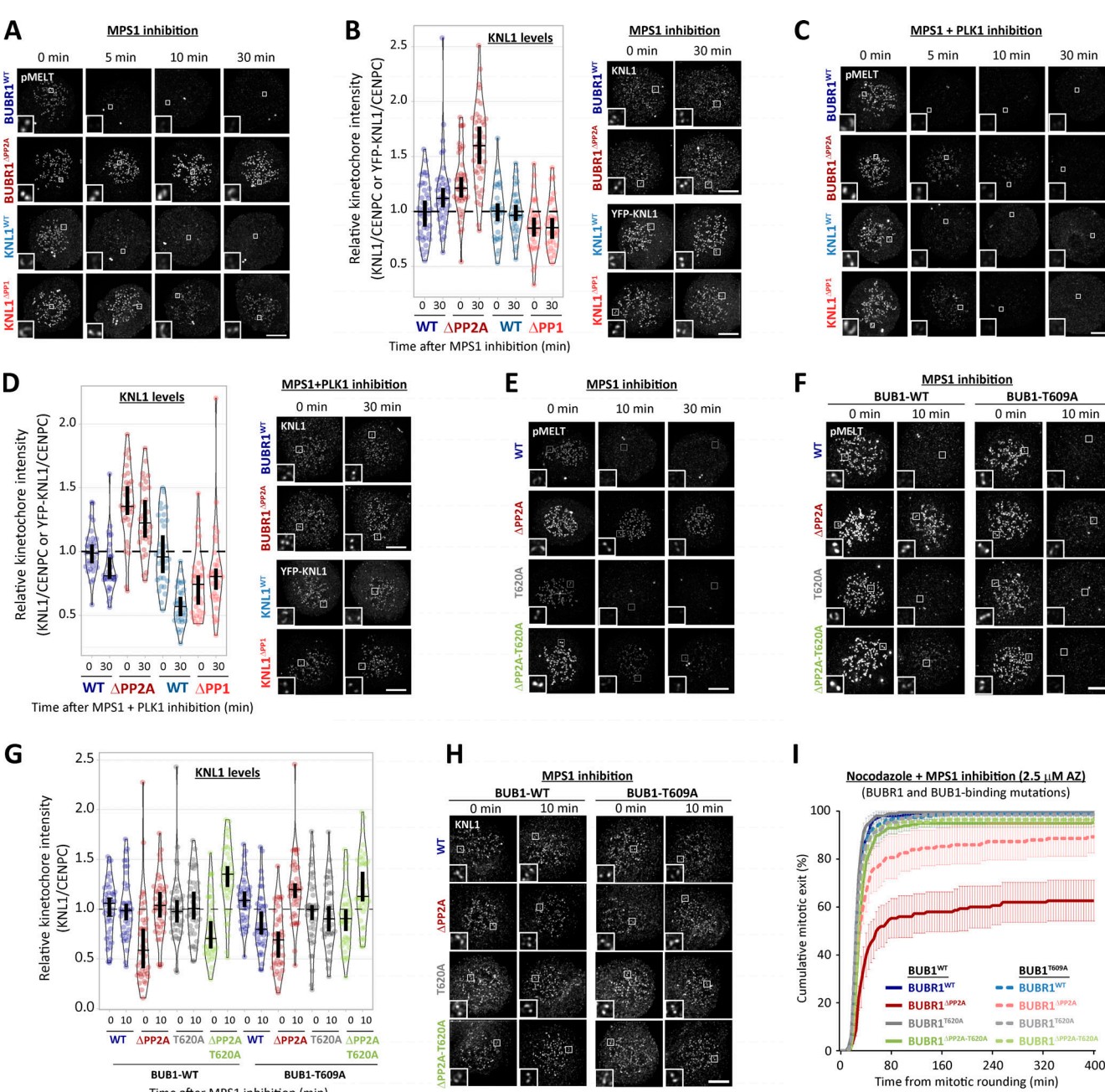

Figure S3. **PLK1 inhibition or removal from BUB complex restores SAC silencing in the phosphatase-binding mutants (related to Fig. 2). (A)** Example immunofluorescence images of the kinetochore quantifications shown in Fig. 2 A. **(B)** Representative images and quantification of KNL1 (BUBR1^{WT/ΔPP2A} cells) or YFP-KNL1 (KNL1^{WT/ΔPP1} cells) after MPS1 inhibition with AZ-3146 (2.5 μM), as in Fig. 2 A. Graph represents 40 cells from four experiments. **(C)** Representative immunofluorescence images of quantifications shown in Fig. 2 C. **(D)** Representative images and quantification of KNL1 (BUBR1 cells) or YFP-KNL1 (KNL1 cells) after MPS1 inhibition with AZ-3146 (2.5 μM) in combination with the PLK1 inhibitor BI-2536 (100 nM), as in Fig. 2 C. Graph represents 30 cells from three experiments. **(E and F)** Example immunofluorescence images of the kinetochore quantifications shown in Fig. 2 E (E) and Fig. 2 G (F). **(G and H)** Quantification of KNL1 levels (G) and representative images (H) of BUBR1/BUB1 Polo-box–binding mutants treated with MPS1 inhibitor (2.5 μM AZ-3146), as in Fig. 2 G. Graph shows median of 40 cells per condition from four experiments. **(I)** Effect of mutating both PLK1 binding sites BUBR1 (pT620) and BUB1 (pT609) on mitotic exit in nocodazole-arrested cells, treated with MPS1 inhibitor AZ-3146 (2.5 μM). Graph shows the means (±SEM) of 150–200 cells from three or four experiments. The images were chosen that most closely resemble the mean values in the quantifications. The insets show magnifications of the outlined regions. Scale bars, 5 μm. Inset size, 1.5 μm.

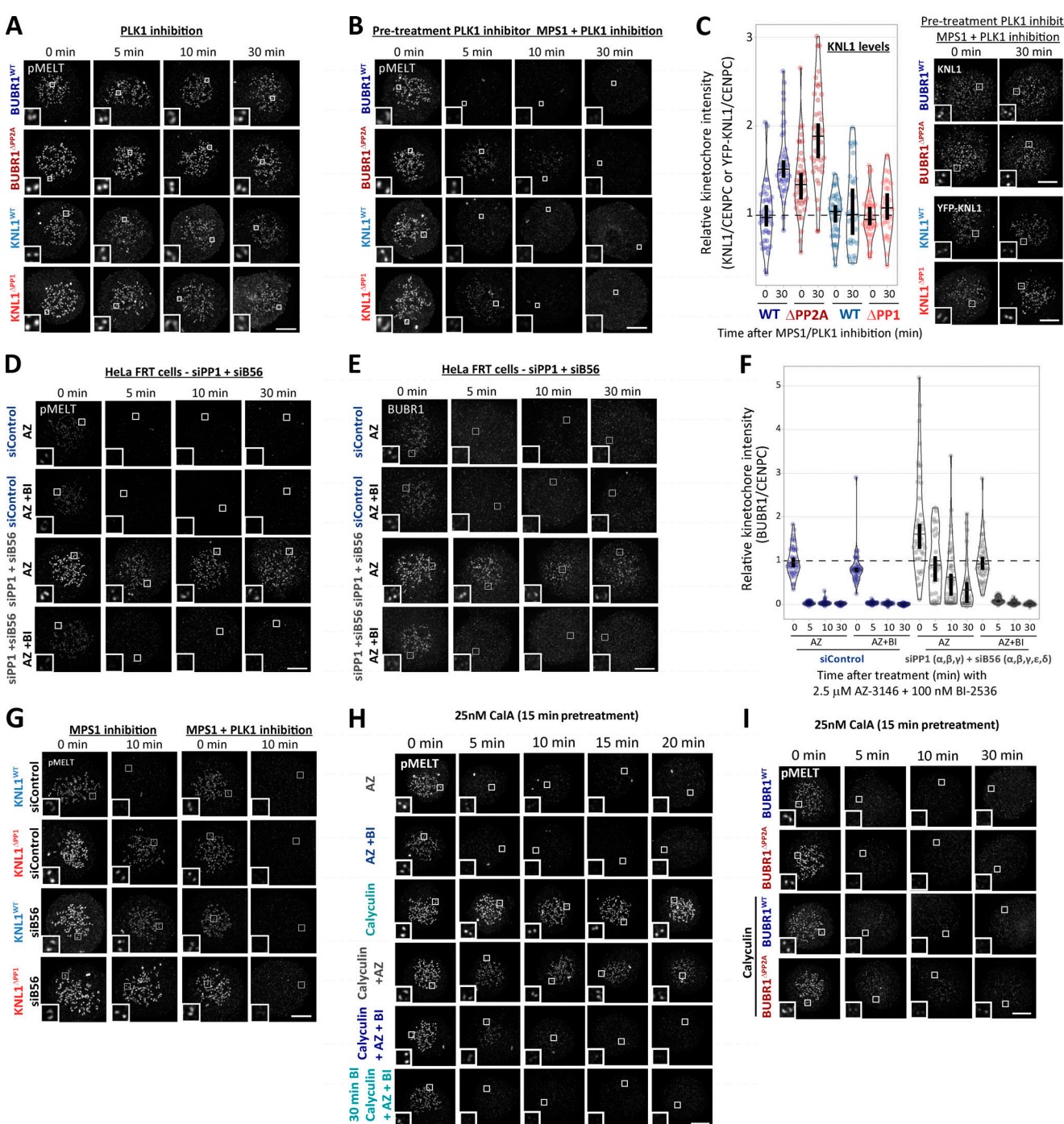

Figure S4. **Representative images from kinetochore quantifications in Fig. 4. (A and B)** Example immunofluorescence images of the kinetochore quantifications shown in Fig. 4 A (A) and Fig. 4 B (B). **(C)** Representative images and quantification of KNL1 (BUBR1 cells) or YFP-KNL1 (KNL1 cells) after 30 min pretreatment with PLK1 inhibitor followed by coinhibition with MPS1 and PLK1 inhibitors, as in Fig. 4 B. Graph represents 40 cells from four experiments. **(D–F)** Immunofluorescence images of pMELT (D) and BUBR1 (E), and kinetochore quantification BUBR1 (F) from cells treated as in Fig. 4 C. Graph shows median of 30–40 cells per condition from three or four experiments. **(G–I)** Representative immunofluorescence images of the kinetochore quantifications shown in Fig. 4 D (G), Fig. 4 E (H), and Fig. 4 F (I). MG132 was included in combination with MPS1 inhibitor in every case to prevent mitotic exit. For all graphs, each dot represents a cell, and vertical bars show 95% CIs. All images were chosen that most closely resemble the mean values in the quantifications. Scale bars, 5 µm. Inset size, 1.5 µm.

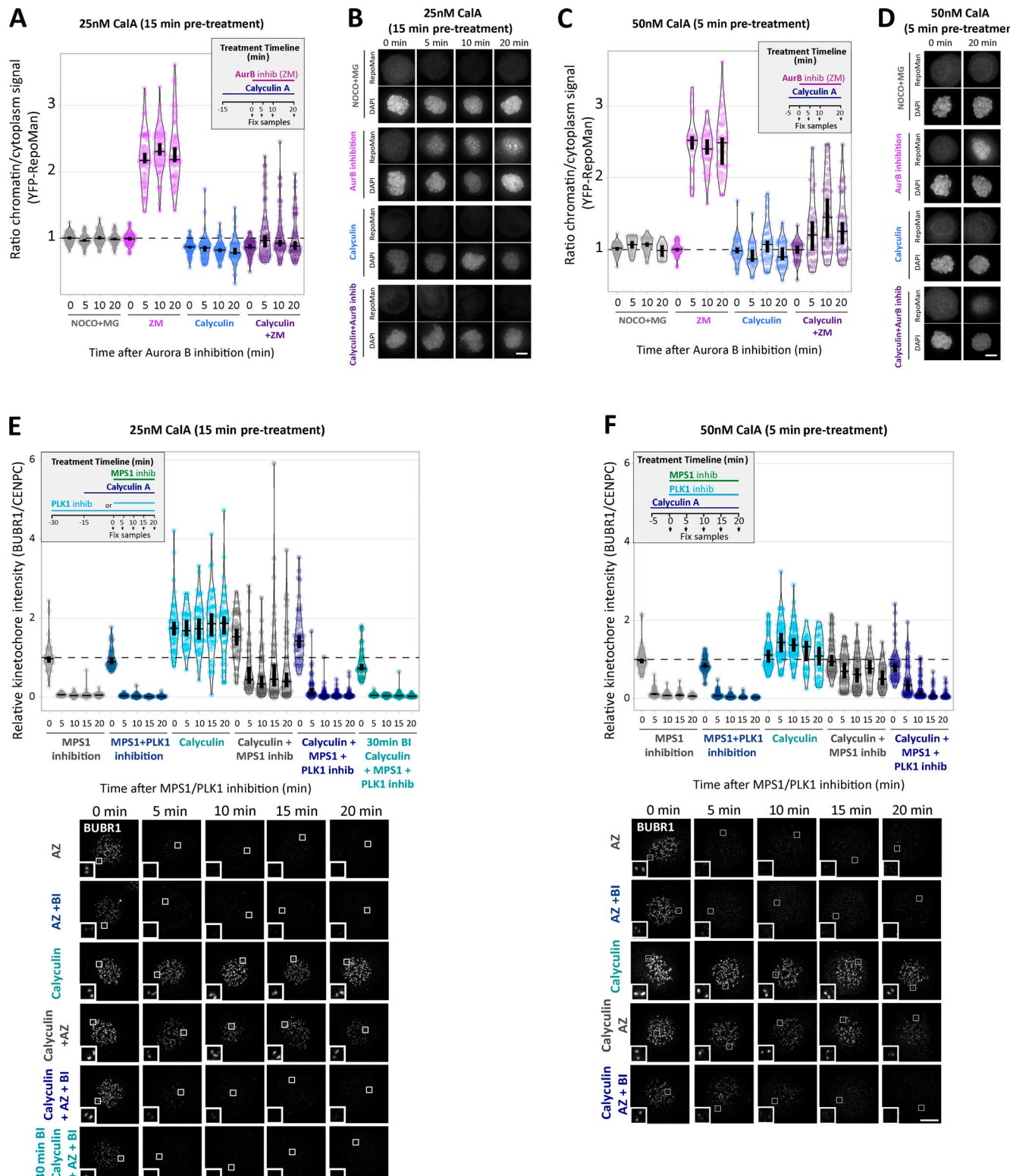

Figure S5. **Example images and kinetochore quantification of BUBR1 and RepoMan after calyculin A treatment with 25 nM and 50 nM doses (related to Fig. 4).** **(A–D)** Quantifications (A and C) and representative images (B and D) of YFP-RepoMan signal on chromatin and cytoplasm to confirm phosphatase inhibition with 25 nM calyculin A (A and B) or 50 nM calyculin A (C and D) treatment regimens. Panel A represents 70 cells from four experiments, and C represents 20–45 cells from two or three experiments. **(E)** Kinetochore BUBR1 quantifications from cells treated with 25 nM calyculin A as in Fig. 4 E. Graph shows median of 40–50 cells per condition from four or five experiments. **(F)** Kinetochore levels of BUBR1 after phosphatase inhibition with 50 nM calyculin A. Graph represents 30–60 cells per condition from three to six experiments. MG132 was included in combination with MPS1 inhibitor in every case to prevent mitotic exit. For all graphs, each dot represents a cell, and vertical bars show 95% CIs. All images were chosen that most closely resemble the mean values in the quantificaions. The insets show magnifications of the outlined regions. Scale bars, 5 µm. Inset size, 1.5 µm.

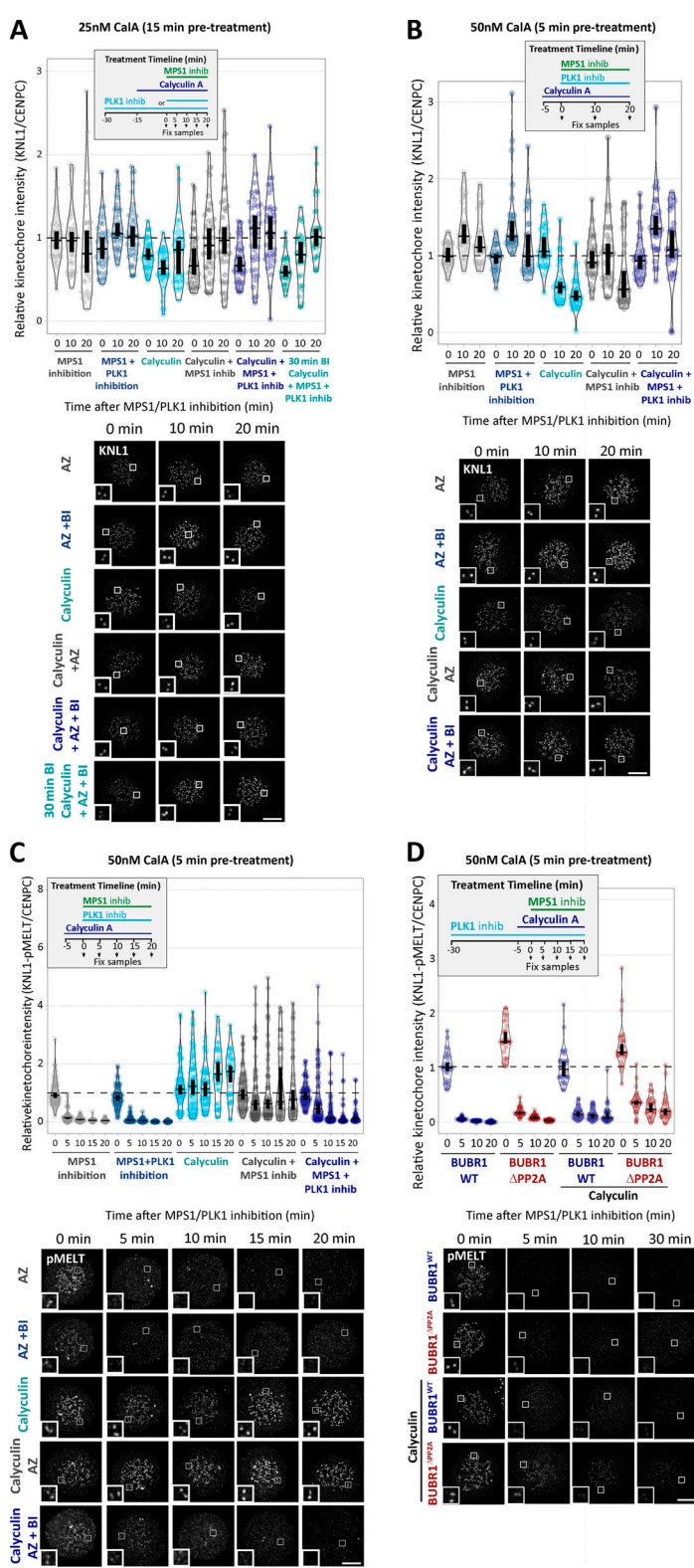

Figure S6. **Example images and kinetochore quantification KNL1 and KNL1-pMELT after calyculin A treatment with 25 nM and 50 nM doses (related to Fig. 4). (A)** Quantification of KNL1 levels in cells treated with 25 nM calyculin as in Fig. 4 E. Graph shows 30–40 cells per condition from three or four experiments. **(B and C)** Kinetochore levels of KNL1 (B) and pMELT (C) after treatment with 50 nM calyculin A. B displays 30 cells from three experiments, and C represents 50–70 cells from five to seven experiments. **(D)** Effects of PP2A-binding mutants in combination with the PP1/PP2A phosphatase inhibitor calyculin A (50 nM) on pMELT dephosphorylation in nocodazole-arrested cells treated with PLK1 (100 nM BI-2536) and MPS1 (2.5 µM AZ-3146) inhibitors, as indicated in the timeline. Graph displays kinetochore intensities of 30 cells per condition from three experimental repeats. MG132 was included in combination with MPS1 inhibitor in every case to prevent mitotic exit. For all graphs, each dot represents a cell, and vertical bars show 95% CIs. All images were chosen that most closely resemble the mean values in the quantifications. The insets show magnifications of the outlined regions. Scale bars, 5 µm. Inset size, 1.5 µm.

**Model for how regulated and unregulated phosphatases can work together**

**Threshold model for substrate dephosphorylation**

PP1

Phosphatase regulation primarily controls kinase localization/activity

PP2A

Point of kinase recruitment/activation

Zone/gradient of kinase activity

Different spatially resolved substrates

Basal (unregulated) phosphatase activity synchronously dephosphorylates substrates

Regulated phosphatase controls kinase activity

Kinase activity

Downstream substrates dephosphorylated

Substrate Phosphorylation

Basal phosphatase activity

Substrate Dephosphorylation

Figure S7. **Model for how localized processes could use a combination of regulated and unregulated phosphatase activity to synchronously dephosphorylate spatially resolved substrates (related to Fig. 4).**

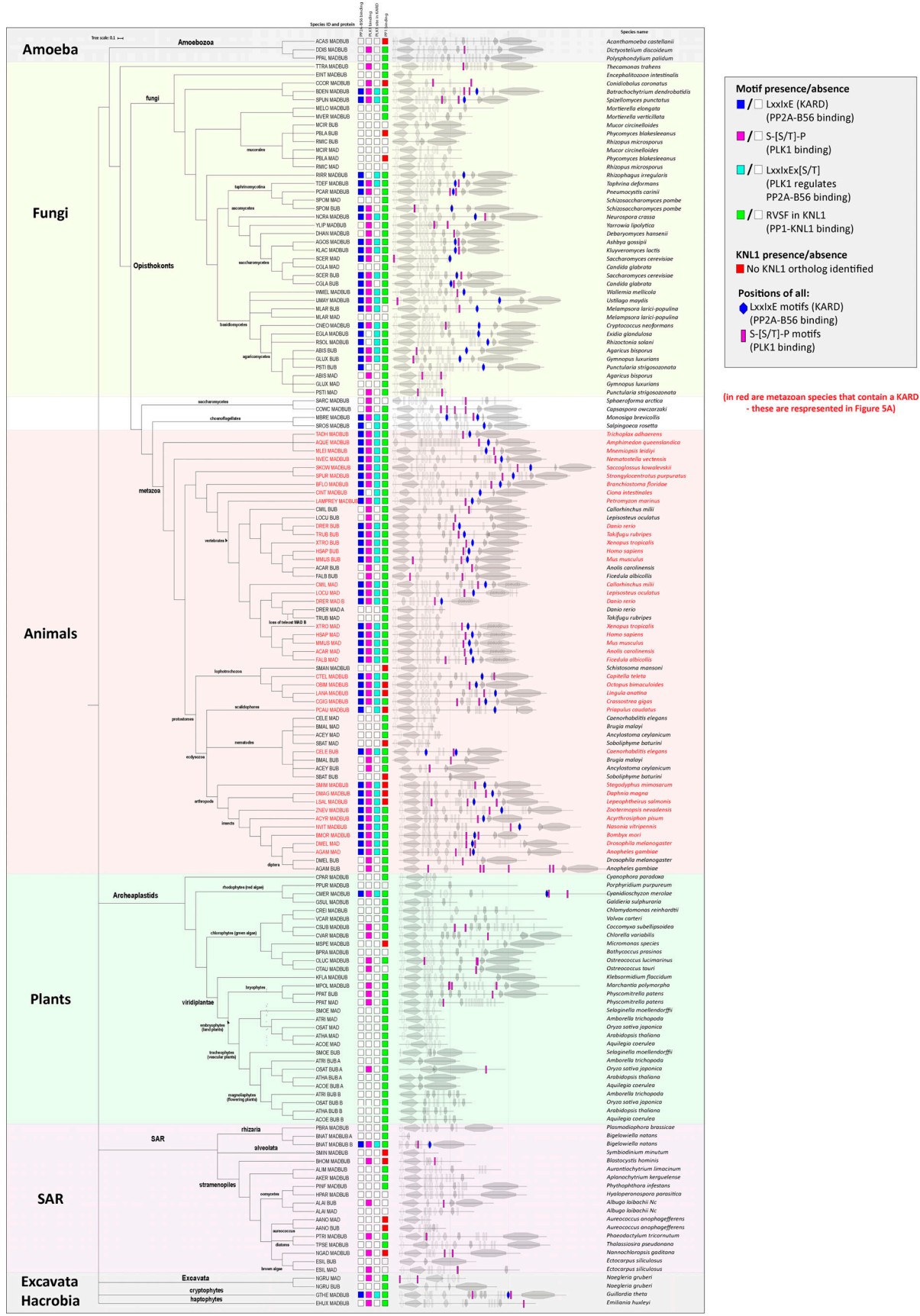

Figure S8. **Annotation of putative PBMs and PP2A-B56 binding motifs (KARD) within eukaryotic MADBUB homologues (related to Fig. 5).** Adapted from Tromer et al. (2016).

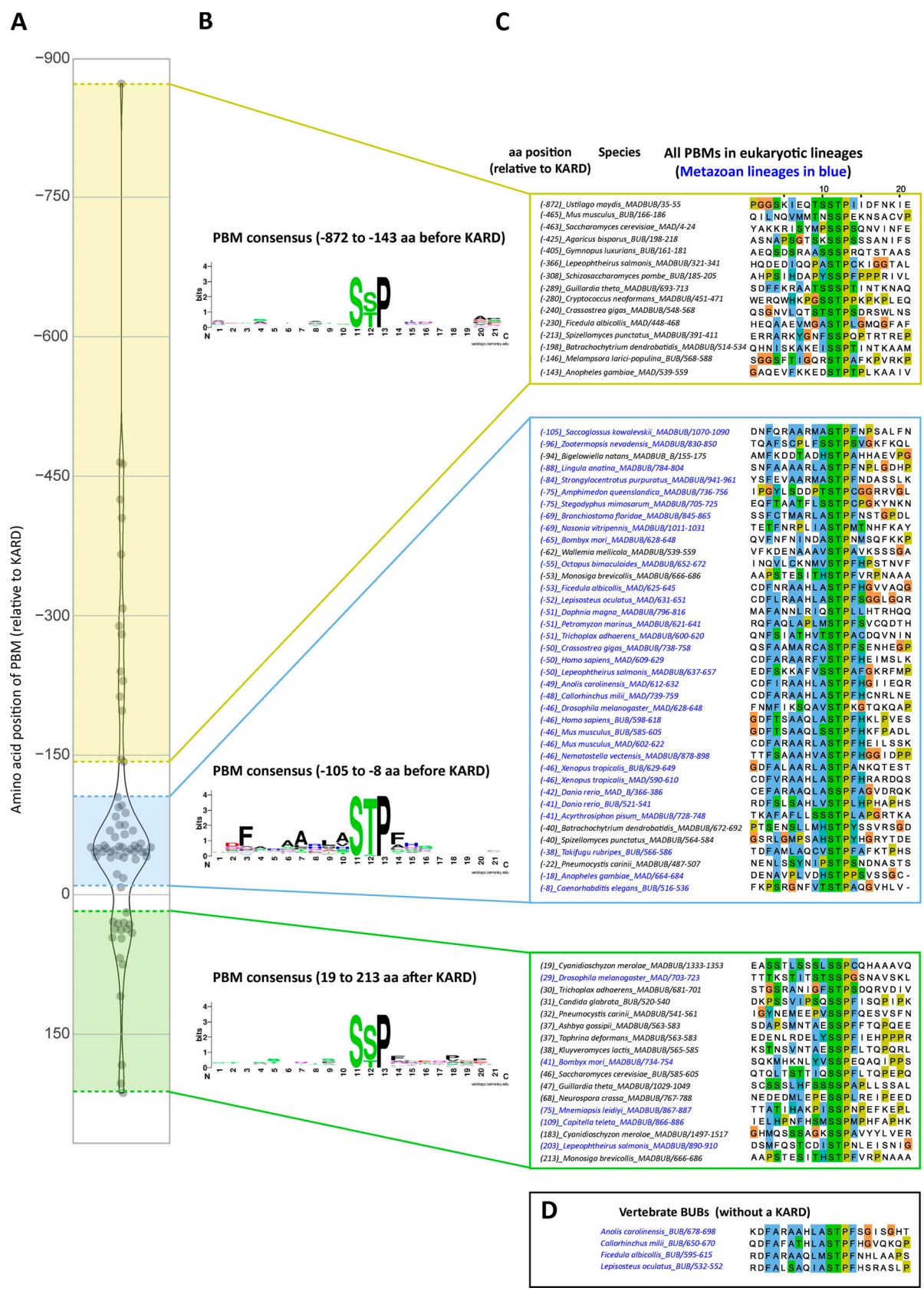

Figure S9. **Sequence alignment of all PBM in eukaryotic MADBUB homologues with respect to position from the KARD (related to Fig. 5). (A)** Distribution of the PBMs with respect to the distance from the KARD position (zero). Positive values represent PBM after KARD and negative values represent PBM before KARD. **(B and C)** Consensus sequence (B) and sequence alignment (C) of the PBM depending on the relative position from KARD motif. **(D)** PBM sequence alignment of vertebrate BUBs without a KARD.

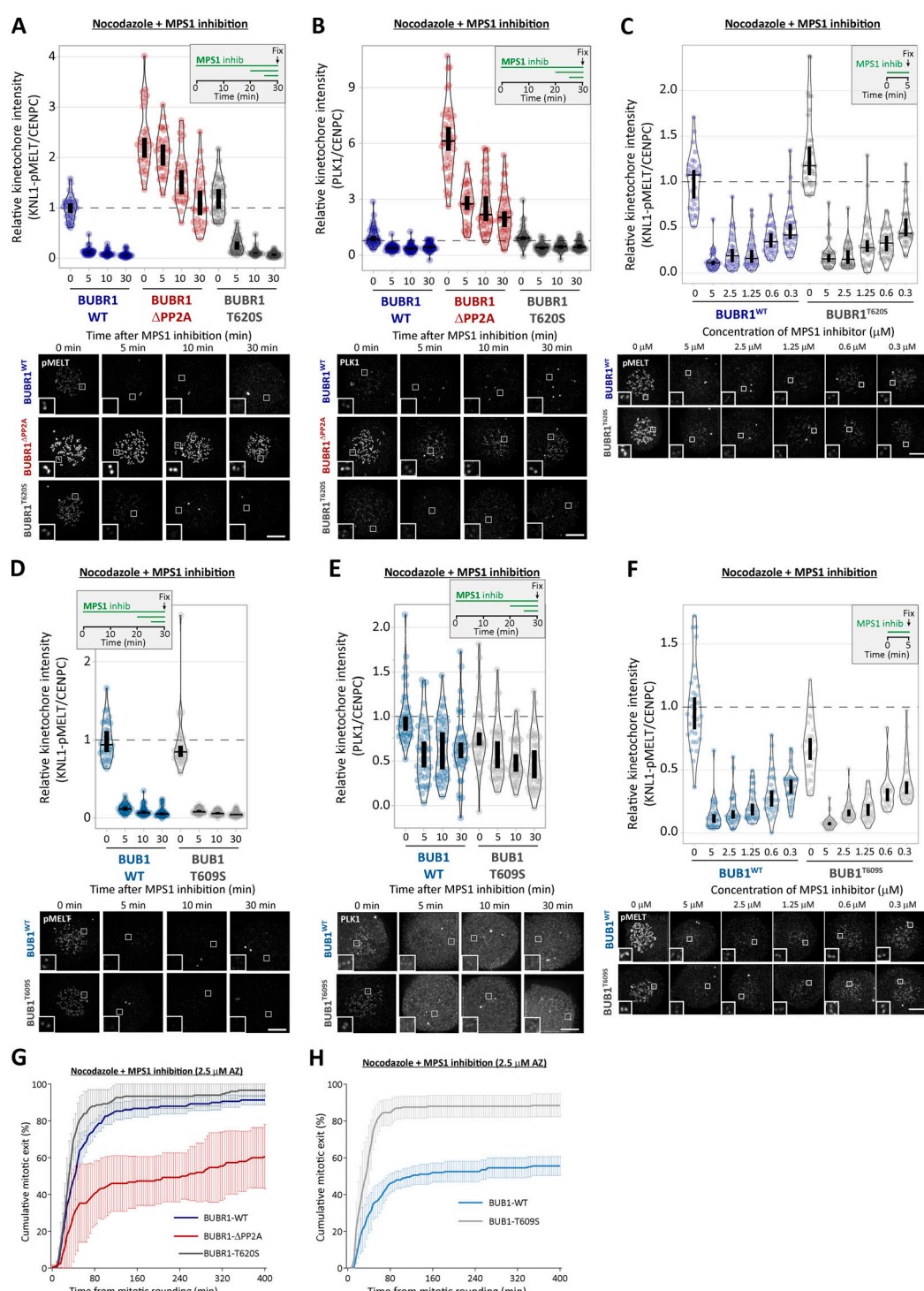

**Figure S10.  Thr-to-Ser conversion in the PBM of BUB1 or BUBR1 does not increase PLK1 levels or delay MELT dephosphorylation after MPS1 inhibition (related to Fig. 5). (A and B)** Effect of Thr-to-Ser conversion in BUB1 (T609S) on kinetochore levels of pMELT (A) and PLK1 (B) after MPS1 inhibition with 2.5 µM AZ-3146. A shows 30 cells per condition from three experiments, and B represents 30–40 cells from three or four experiments. **(C)** MELT phosphorylation levels in BUB1-T609S cells after 5 min of MPS1 inhibition with different concentrations of AZ-3146 (0.3–5 µM) in nocodazole-arrested cells. Graphs display kinetochore intensities of 30 cells, three experiments. **(D and E)** Effect of Thr-to-Ser conversion in BUBR1 (T620S) in pMELT (D) and PLK1 (E) levels after MPS1 inhibition (2.5 µM AZ-3146). D represents 30 cells from three experiments, and E shows 40 cells from four experiments. **(F)** MELT phosphorylation levels in BUBR1-T620S cells after 5 min of MPS1 inhibition with different concentrations of AZ-3146 (0.3–5 µM) in nocodazole-arrested cells. Graphs display kinetochore intensities of 20–30 cells, two or three experiments. **(G and H)** Effect of BUBR1-T620S (G) and BUB1-T609S (H) mutations on mitotic exit in nocodazole-arrested cells treated with MPS1 inhibitor (2.5 µM AZ-3146). G shows the means (±SEM) of 150 cells from three experiments, and H shows 200 cells from four experiments. MG132 was included in combination with MPS1 inhibitor in every case to prevent mitotic exit. For all graphs, each dot represents a cell, and vertical bars show 95% CIs. All images were chosen that most closely resemble the mean values in the quantifications. Scale bars, 5 µm. Inset size, 1.5 µm.

**Data S1 shows the full list of annotated and excluded PBMs in eukaryotic MADBUB homologues. Data S2 shows the KNL1 orthologues from Fig. S8 that are not published in Kops et al. (2020), Tromer et al. (2015), or van Hooff et al. (2017).**

