## [Peer Review File · The Journal of Cell Biology]

Kinetochore phosphatases suppress autonomous PLK1 activity to control the mitotic checkpoint

Marilia Cordeiro, Richard Smith, and Adrian Saurin

Corresponding Author(s): Adrian Saurin, University of Dundee

Review Timeline:

Submission Date:	2020-02-05
Editorial Decision:	2020-02-19
Revision Received:	2020-08-20
Editorial Decision:	2020-09-17
Revision Received:	2020-10-07

Monitoring Editor: Timothy Yen

Scientific Editor: Tim Spencer

Transaction Report:

DOI: <https://doi.org/10.1083/jcb.202002020>

Revision 0

Review #1

1. How much time do you estimate the authors will need to complete the suggested revisions:

Estimated time to Complete Revisions (Required)

(Decision Recommendation)

Cannot tell / Not applicable

2. Evidence, reproducibility and clarity:

Evidence, reproducibility and clarity (Required)

The manuscript by Cordeiro et al provides a series of compelling evidences to support a provocative conclusion: PP2A-B56 and PP1 are critical for SAC silencing mainly by restraining and extinguishing autonomous kinase activity at kinetochores. This finding challenges the prevailing view of PP2A-B56/PP1-mediated KNL1-MELT dephosphorylation as a major SAC silencing event. This represents a paradigm change in the field and opens an important goal for future research: determine the phosphatases that dephosphorylate the MELTs. In my view this paper delivers an important clarification on how PP1-KNL1 and PP2A-B56 actually drive SAC silencing. This is a nice study and will move the field forward. The manuscript is globally solid, very well written and the conclusions are generally supported by the experimental data. However, I do have some issues with the following points, which in my view, if unaddressed, may leave the conclusion a bit fragile: ****Minor comments:**** 1) The authors propose that PP1-KNL1 and BUBR1-bound PP2A-B56 continuously antagonise PLK1 association with the BUB complex by dephosphorylating the CDK1 phosphorylation sites on BUBR1 (pT620) and BUB1 (pT609). It is therefore expected that converting these residues to aspartate would increase PLK1 recruitment. It would be interesting to verify if this hypothesis fits with the proposed model. 2) In Figure 1E, are the mean values for BubR1WT+BubWT and BubR1WT+Bub1T609 both normalized to 1? If so, this fails to reveal the contribution of Bub1 T609 for the recruitment of PLK1 when PP2A-B56 is allowed to localize at kinetochores. 3) What underlies the increase in Bub1 levels at unattached kinetochores of siBubR1 cells (Figure S1C)? Is this caused by an increase in Bub1 T609 phosphorylation and consequently unopposed PLK1 recruitment, which consequently increases MELT phosphorylation? 4) Although the immunoblotting from Figure S1D indicates that BubR1T620A and Bub1T609A are expressed at similar levels as their respective WT counterparts, some degree of single-cell variability is expected to occur. As a complement to Figure 1B,C and Figure S1E,F could the authors plot the kinetochore intensity of BubR1 pT620 and Bub1T609 relative to the YFP-BubR1 and YFP-Bub1 signal, respectively? 5)

The authors nicely show that excessive PLK1 levels at the BUB complex are able to maintain MELT phosphorylation and the SAC (independently of MPS1) when KNL1-localised phosphatases are removed (Figures 2A,B). However, it should be noted that PLK1 is able to promote MPS1 activation at kinetochores and so, whether AZ-3146 at 2.5 μ M efficiently inhibits MPS1 under conditions of excessive PLK1 recruitment should be confirmed. Can the authors provide a read-out for MPS1 activation status or activity (other than p-MELTs) to exclude a potential contribution of residual MPS1 activity in maintaining the p-MELTs and SAC? 6) To examine whether PLK1 removal is the major role of PP1-KNL1 and PP2A-B56 in the SAC or whether they are additionally needed to dephosphorylate the MELTs, the authors monitored MELT dephosphorylation when MPS1 was inhibited immediately after 30-minute of BI2356. This revealed similar dephosphorylation kinetics, irrespective of compromised PP1-KNL1 or PP2A-B56 activity, thus suggesting that these pools of phosphatases are not required to dephosphorylate MELTs. To confirm this and exclude phosphatase redundancy, the authors simultaneously depleted all PP1 and B56 isoforms or treated cells with Calyculin A to inhibit all PP1 and PP2A phosphatases. In both of these situations, the kinetics of MELT dephosphorylation was indistinguishable from wild type cells if MPS1 and PLK1 were inhibited together. These observations led to the conclusion that neither PP1 or PP2A are required to dephosphorylate the MELT motifs. Instead they are needed to remove PLK1 from the BUB complex. This set of experiments is well-designed and the results support the conclusion. However, it would be of value if the authors provide evidence for the efficiency of PP1 and B56 isoforms depletion and for the efficiency of phosphatase inhibition by Calyculin A. An alternative read-out for the activity of PP1 and PP2A-B56 (other than p-MELT dephosphorylation) clearly confirming that both phosphatases are compromised when MPS1 and PLK1 are inhibited together could make a stronger case in excluding the contribution of residual PP1 or PP2A to the observed dephosphorylation of MELT motifs. To summarize, this is a very good paper and will definitely cause an important impact in the field of mitosis.

3. Significance:

Significance (Required)

This manuscript provides an important conceptual advance for the field of mitosis, specifically to the topic of mitotic checkpoint regulation. It remains elusive how the spindle assembly checkpoint is silenced. While previous studies have shown that PP1-KNL1 and PP2A-B56 contribute to suppress SAC signaling, how they do so is unclear. This study provides important insight into this matter. Cordeiro and colleagues demonstrate that in contrast with previous expectations, PP1 and PP2A promote SAC silencing, not by directly dephosphorylating MELT motifs on KNL1, but instead by removing PLK1 from the Bub complex. The authors find that these phosphatases antagonise CDK1- phosphorylations on BubR1 and Bub1 to dampen PLK1 levels. This activity is crucial to prevent PLK1 from maintaining MELT phosphorylation in an autocatalytic manner, thus (probably) allowing prompt SAC silencing following stable kinetochore-microtubule attachments. The described mechanism extends our view of how the SAC is regulated and should be of interest to those in the field of mitosis. The findings described in this paper allow us to better understand how cells silence the SAC. This is a top priority in the field, as the inability to timely quench SAC signaling can result in chromosome segregation

errors. Determining the phosphatases that actually dephosphorylate the MELT motifs will be an essential next step forward

Review #2

1. How much time do you estimate the authors will need to complete the suggested revisions:

Estimated time to Complete Revisions (Required)

(Decision Recommendation)

Cannot tell / Not applicable

2. Evidence, reproducibility and clarity:

Evidence, reproducibility and clarity (Required)

****Summary:**** The work focuses on the role of kinetochore localized protein phosphatases in the dephosphorylation of MELT motifs and SAC silencing. The focus is on PP1 bound to KNL1 and PP2A-B56 bound to BubR1 and uses largely RNAi rescue experiments in human cell lines combined with immunofluorescence analysis and time-lapse imaging. The authors show that kinetochore localized phosphatases antagonize the localization of the Plk1 mitotic kinase to kinetochores. This is due to the dephosphorylation of BubR1 T620 and Bub1 T609 that are binding sites for Plk1 on the kinetochore. The main conclusion is that if Plk1 kinetochore localisation is prevented then there is no longer a need for kinetochore phosphatases for SAC silencing and MELT dephosphorylation. ****Major comments:**** 1) In its current state I am not convinced that the key conclusions are fully supported by the experiments and alternative conclusions/interpretations can be drawn. For example the level of MELT phosphorylation will be determined by the balance of kinase and phosphatase activity and if they do not achieve 100% inhibition of Mps1 in their assays then they are not strictly monitoring dephosphorylation kinetics in their assays. If the combination of Mps1 and Plk1 inhibition then more strongly inhibits Mps1 then dephosphorylation kinetics becomes faster. Thus subtle differences in Mps1 activity under their different conditions could lead to misleading conclusions but in its present state a careful analysis of Mps1 activity is not provided. This lack of complete inhibition also applies to the phosphatases and the experiments in Figure 3E indicates that their Calyculin preparation is not really active as at steady state MELT phosphorylation levels are much less affected than in for instance BubR1 del PP2A (Figure 2A as an example). Thus they likely still have phosphatase activity in the experiment in figure 3E making it difficult to draw the conclusions they do. A more careful analysis of kinase and phosphatase activities in their different perturbations would be recommendable and should be possible within a reasonable time frame. 2) A more stringent test of their model would also be needed. What happens if Plk1 is

artificially maintained in the Bub complex? The prediction would be that SAC silencing should be severely delayed even when Mps1 is inhibited. This is a straightforward experiment to do that should not take too long. If the polobox can bind phosphoSer then one could also make BubR1 T620S to slow down dephosphorylation of this site (PPPs work slowly on Ser while Cdk1 have almost same activity for Ser and Thr). 3) Another issue is the relevance of Plk1 removal under normal conditions. As their quantification shows in figure 1D-E (I think there is something wrong with figure 1E - should likely be Bub1) the contribution of BubR1 T620 and Bub1 T609 to Plk1 kinetochore localisation seems minimal. Thus upon SAC satisfaction there is not really a need to remove Plk1 through dephosphorylation as it is already at wild type levels. It is only in their BubR1 and KNL1 mutants that there is this effect so one has to question the impact in a normal setting. This is consistent with the data in Figure S1D showing no phosphorylation of these sites under unperturbed conditions. 4) They write that in the absence of phosphatase activity Plk1 becomes capable of supporting SAC independently (of Mps1 is implied). They do not show this - only that MELT phosphorylation is maintained. As Mps1 has other targets required for SAC activity I would rephrase this. 5) The method section is extensive and contains sufficient information for reproducing data. 6) Data and statistical analysis is ok.

3. Significance:

Significance (Required)

The advance is clearly conceptual and provides a new way of thinking about the kinetochore localized phosphatases. These phosphatases and the SAC have been immensely studied but this work brings in a new angle. The discussion would benefit from some evolutionary perspectives as the PP1 and PP2A-B56 binding sites are very conserved but the Plk1 docking sites on Bubs less so. This will be of interest to people in the field of cell division and researchers interested in phospho-mediated signaling. Field of expertise: kinetochore/phosphatases/bub proteins Jakob Nilsson

Review #3

1. How much time do you estimate the authors will need to complete the suggested revisions:

Estimated time to Complete Revisions (Required)

(Decision Recommendation)

Cannot tell / Not applicable

2. Evidence, reproducibility and clarity:

Evidence, reproducibility and clarity (Required)

The Spindle Assembly Checkpoint (SAC) is a conserved surveillance device that responds to errors in kinetochore-microtubule attachments to ultimately prevent the onset of anaphase until all chromosomes are bipolarly attached. Current models of SAC posit that the Mps1 kinase initiates the SAC signalling cascade by phosphorylating the KNL1/Blinkin kinetochore scaffold at MELT repeats, in order to create phospho-docking sites for the hetero-tetrameric BUB complex made by BUB1-BUB3-BUB3-BUBR1. The BUB complex, in turn, promotes the assembly the Mitotic Checkpoint Complex (MCC), which prevents anaphase onset by inhibiting the E3 ubiquitin ligase Anaphase-Promoting Complex bound to its activator Cdc20 (APCCdc20). The polo-like kinase PLK1, which is recruited to kinetochores through its binding to BUBR1, contributes to the robustness of SAC signalling in human cells by cooperating with Mps1 in KNL1/Blinkin phosphorylation and by phosphorylating MPS1 itself, thereby enhancing its catalytic activity. While in human cells MPS1 is the predominant kinase in SAC signalling, aided by PLK1, in other organisms where MPS1 is absent, such as in nematodes, PLK1 functionally replaces MPS1 and is necessary for SAC activation. Once all chromosomes are bipolarly attached, SAC signalling is extinguished. Key to this process are the PP1 and PP2A-B56 phosphatases that antagonise KNL1 phosphorylation by MPS1 and PLK1 and also dephosphorylate the T-loop of MPS1 to lower its catalytic activity. Current models envision that PP1 and PP2A-B56 dephosphorylate the MELT repeats of KNL1 directly. Importantly, this has been formally tested for both PP2A-B56 in human cells (Espert et al., 2014) and PP1 in yeast (London et al., 2012). In the present manuscript, the above model is challenged with the proposal that the main contribution of PP1 and PP2A-B56 to SAC silencing is to lower the levels of PLK1 at kinetochores, rather than to dephosphorylate KNL1. By interfering with the levels of these opposing kinases and phosphatases at kinetochores the authors describe an interesting interplay that confirms an overlapping function of PLK1 and MPS1 in KNL1 phosphorylation and highlights a role for the phosphatases in dampening PLK1 kinetochore levels. Consistently, inhibition of both Mps1 and PLK1 is sufficient to bring about KNL1 dephosphorylation upon inhibition of both phosphatases at kinetochores. The hypothesis is interesting and experiments are in general carefully designed and performed. It is clear from the presented data that PP1 and PP2A-B56 antagonize PLK1 kinetochore localisation and that the MELT repeats of KNL1 can be dephosphorylated even in the absence of phosphatases, provided that MPS1 and PLK1 are inhibited. However, in my opinion the results do not rule out that phosphatases actually have a primary and direct role in KNL1 dephosphorylation. ****Major comments:**** 1. An important limitation of this study is that KNL1 dephosphorylation at MELT repeats is monitored only by indirect immunofluorescence using phospho-specific antibodies. Thus, reduction of phospho-KNL1 kinetochore signals could be due to protein turnover at kinetochores, rather than to dephosphorylation. This is a serious issue that could be addressed by checking KNL1 dephosphorylation during time course experiments by western blot using phospho-specific antibodies, as previously done (Espert et al., 2014). 2. For obvious technical reasons, the shortest time point at which authors compare KNL1 dephosphorylation upon MPS1-PLK1 inhibition is 5 minutes. Based on immunofluorescence data, authors conclude that kinetics of KNL1 dephosphorylation are similar when kinases are inhibited, independent of whether or not kinetochore-bound phosphatases are active. However, in most experiments (e.g. Fig. 3B, 3C, 3E) lower levels of MELT phosphorylation are detected after 5 minutes of kinase inhibition when phosphatases are present than when they are absent, suggesting that phosphatases likely do

contribute to KNL1 dephosphorylation. I suspect that differences between the presence and absence of phosphatases might even be more obvious if authors were to look at shorter time points, when phosphatases conceivably accomplish their function. I would therefore suggest that the authors tone down their conclusions, as their data complement but do not disprove the previous model. 3. In all experiments cells are kept mitotically arrested through nocodazole treatment, which is not quite a physiological condition to study SAC silencing. This could potentially mask the real contribution of phosphatases in MELT dephosphorylation. Indeed, it is possible that higher amounts of phosphatases are recruited to kinetochores during SAC silencing than during SAC signalling (e.g. during SAC signalling Aurora B phosphorylates the RVSF motif of KNL1 to keep PP1 binding at low levels; Liu et al., 2010). What would happen in a nocodazole wash-out? Would phosphatases be dispensable in these conditions for normal kinetics of MELT dephosphorylation and anaphase onset if PLK1 is inhibited? 4. Other data are overinterpreted. For instance, the evidence that CDK1-dependent phosphorylation sites in Bub1 and BubR1 is enhanced when PP1 and PP2A-B56 are absent at kinetochores suggests but does not "demonstrate that PP1-KNL1 and BUBR1-bound PP2A-B56 antagonise PLK1 recruitment to the BUB complex by dephosphorylating key CDK1 phosphorylation sites on BUBR1 (pT620) and BUB1 (pT609)(Figure 1F)". Similarly, the claim "when kinetochore phosphatase recruitment is inhibited, PLK1 becomes capable of supporting the SAC independently" referred to Fig. 2C-D is an overstatement, as residual MPS1 kinase could be still active in the presence of the AZ-3146 inhibitor. **Minor comments:** 1. In many graphs (Fig. 1A-C, Fig. 2A,C) relative kinetochore intensities are quantified over "CENPC or YFP-KNL1". Authors should clarify when it is one versus the other. 2. The drawing in Fig. 1F depicts the action of PP1 and PP2A-B56 in antagonising PLK1 at kinetochores. Thus, the output should be SAC silencing, rather than activation. 3. In the Discussion authors speculate that KNL1 dephosphorylation relies on a constitutive phosphatase with unregulated basal activity. Would a phosphatase be needed at all when MPS1 and PLK1 are inhibited? Could phosphorylated KNL1 be actively degraded? 4. What happens to MPS1 when KNL1-bound PP1 and BUBR1-bound PP2A are absent? Do its kinetochore levels increase as observed for PLK1? And what about the kinetochore levels of Bub1 and BubR1?

3. Significance:

Significance (Required)

The nature of the advance is conceptual. This paper challenges (although I would rather say "integrates") the prevailing model of spindle checkpoint silencing. The current model of SAC silencing envisions that PP1 and PP2A-B56 phosphatases oppose SAC kinases (Mps1 and Polo kinase) by directly dephosphorylating some of their targets (e.g. the kinetochore scaffold KNL1 and MPS1 itself). This work proposes instead that the main function of the above phosphatases is to keep low levels of the polo kinase PLK1 at kinetochores, which would otherwise boost KNL1 phosphorylation and assembly of SAC complexes. People working in the fields of mitosis, chromosome segregation, aneuploidy, spindle checkpoint, kinases/phosphatases could be interested by these findings. Reviewer's field of expertise: Cell cycle, mitosis, spindle assembly checkpoint

Authors' response to the reviews for the manuscript 'Review Commons Refereed Preprint #RC-2019-00133'

Response to Reviewers Comments

We would like to thank all reviewers for carefully considering our manuscript and providing useful suggestions/ideas. The general consensus was that our study provides an important conceptual advance that reveals a new way of thinking about kinetochore phosphatases. However, in light of our surprising findings, it was suggested that additional experiments would be required to fully validate our conclusions. In particular, it was seen as important to test whether PLK1 can activate MPS1 from the BUB complex and to confirm that PP1 and PP2A are effectively inhibited in situations where MELT dephosphorylation can occur normally (Figure 3).

In general, we agree with these and the other points raised by the reviewers, therefore we plan to address all comments as outlined in detail below.

The major new additions to the final paper will be the following:

- 1) Experiments to test how BUB-bound PLK1 affects MPS1 activity.
- 2) Experiments to determine the efficiency of phosphatase inhibition in figure 3.
- 3) Experiments to test whether maintaining PLK1 at the BUB complex causes SAC silencing defects
- 4) Evolutionary analysis demonstrating that the PLK1 and PP2A-binding modules have co-evolved in the kinetochore BUB complex. This analysis, which has been performed already, strengthens our manuscript because it provides additional independent evidence for a functional relationship between PLK1 and PP2A on the BUB complex.

Reviewer #1 (Evidence, reproducibility and clarity (Required)):

The manuscript by Cordeiro et al provides a series of compelling evidences to support a provocative conclusion: PP2A-B56 and PP1 are critical for SAC silencing mainly by restraining and extinguishing autonomous kinase activity at kinetochores. This finding challenges the prevailing view of PP2A-B56/PP1-mediated KNL1-MELT dephosphorylation as a major SAC silencing event. This represents a paradigm change in the field and opens an important goal for future research: determine the phosphatases that dephosphorylate the MELTs. In my view this paper delivers an important clarification on how PP1-KNL1 and PP2A-B56 actually drive SAC silencing. This is a nice study and will move the field forward. The manuscript is globally solid, very well written and the conclusions are generally supported by the experimental data. However, I do have some issues with the following points, which in my view, if unaddressed, may leave the conclusion a bit fragile:

****Minor comments:****

- 1) The authors propose that PP1-KNL1 and BUBR1-bound PP2A-B56 continuously antagonise PLK1 association with the BUB complex by dephosphorylating the CDK1 phosphorylation sites on BUBR1 (pT620) and BUB1 (pT609). It is therefore expected that converting these residues to aspartate would increase PLK1 recruitment. It would be interesting to verify if this hypothesis fits with the proposed model.

The general idea to maintain PLK1 at the BUB complex is a good one, but unfortunately polo-box domains do not bind to acidic negatively charged residues. Instead we will attempt to maintain PLK1 at the BUB complex using alternatively approaches (as suggested by reviewer 2).

2) In Figure 1E, are the mean values for BubR1WT+BubWT and BubR1WT+Bub1T609 both normalized to 1? If so, this fails to reveal the contribution of Bub1 T609 for the recruitment of PLK1 when PP2A-B56 is allowed to localize at kinetochores.

The values will be updated and normalised to the BubR1WT+BUB1WT control. We have also performed additional experiments already and overall the results reveal a small reduction in kinetochore PLK1 following BUB1-T609A mutation and a larger reduction upon combined BUBR1-T620A mutation.

3) What underlies the increase in Bub1 levels at unattached kinetochores of siBubR1 cells (Figure S1C)? Is this caused by an increase in Bub1 T609 phosphorylation and consequently unopposed PLK1 recruitment, which consequently increases MELT phosphorylation?

We suspect that PLK1 is not the cause of the increased BUB1 levels because PLK1 kinetochore levels are actually decreased in this situation (Figure S1A).

4) Although the immunoblotting from Figure S1D indicates that BubR1T620A and Bub1T609A are expressed at similar levels as their respective WT counterparts, some degree of single-cell variability is expected to occur. As a complement to Figure 1B,C and Figure S1E,F could the authors plot the kinetochore intensity of BubR1 pT620 and Bub1T609 relative to the YFP-BubR1 and YFP-Bub1 signal, respectively?

There is indeed variability in the level of re-expression of BUBR1/BUB1 on a single cell level, which can at least partially explain the variation on BUBR1-pT620 and BUB1-pT609 observed within in each condition. We can upload these scatter plots at resubmission and include in the supplementary, if required.

5) The authors nicely show that excessive PLK1 levels at the BUB complex are able to maintain MELT phosphorylation and the SAC (independently of MPS1) when KNL1-localised phosphatases are removed (Figures 2A,B). However, it should be noted that PLK1 is able to promote MPS1 activation at kinetochores and so, whether AZ-3146 at 2.5 μ M efficiently inhibits MPS1 under conditions of excessive PLK1 recruitment should be confirmed. Can the authors provide a read-out for MPS1 activation status or activity (other than p-MELTs) to exclude a potential contribution of residual MPS1 activity in maintaining the p-MELTs and SAC?

This is a good point because although PLK1 can phosphorylate the MELTs it can also activate MPS1, although it is unknown whether it can do this from the BUB complex. We had left a dotted line in Figure 4B to include this possibility, but we will now test this directly with additional experiments.

6) To examine whether PLK1 removal is the major role of PP1-KNL1 and PP2A-B56 in the SAC or whether they are additionally needed to dephosphorylate the MELTs, the authors monitored MELT dephosphorylation when MPS1 was inhibited immediately after 30-minute of BI2356. This revealed similar dephosphorylation kinetics, irrespective of compromised PP1-KNL1 or PP2A-B56 activity, thus

suggesting that these pools of phosphatases are not required to dephosphorylate MELTs. To confirm this and exclude phosphatase redundancy, the authors simultaneously depleted all PP1 and B56 isoforms or treated cells with Calyculin A to inhibit all PP1 and PP2A phosphatases. In both of these situations, the kinetics of MELT dephosphorylation was indistinguishable from wild type cells if MPS1 and PLK1 were inhibited together. These observations led to the conclusion that neither PP1 or PP2A are required to dephosphorylate the MELT motifs. Instead they are needed to remove PLK1 from the BUB complex. This set of experiments is well-designed and the results support the conclusion. However, it would be of value if the authors provide evidence for the efficiency of PP1 and B56 isoforms depletion and for the efficiency of phosphatase inhibition by Calyculin A. An alternative read-out for the activity of PP1 and PP2A-B56 (other than p-MELT dephosphorylation) clearly confirming that both phosphatases are compromised when MPS1 and PLK1 are inhibited together could make a stronger case in excluding the contribution of residual PP1 or PP2A to the observed dephosphorylation of MELT motifs.

This is also a good point. We had attempted many different combinations in Figure 3 to inhibit PP1/PP2A activity as efficiently as possible. This is especially important considering the “negative” results on pMELT are very surprising. However, we will now test how efficiently we have inhibited PP1 and PP2A phosphatase function in these experiments.

To summarize, this is a very good paper and will definitely cause an important impact in the field of mitosis.

Reviewer #1 (Significance (Required)):

This manuscript provides an important conceptual advance for the field of mitosis, specifically to the topic of mitotic checkpoint regulation. It remains elusive how the spindle assembly checkpoint is silenced. While previous studies have shown that PP1-KNL1 and PP2A-B56 contribute to suppress SAC signaling, how they do so is unclear. This study provides important insight into this matter. Cordeiro and colleagues demonstrate that in contrast with previous expectations, PP1 and PP2A promote SAC silencing, not by directly dephosphorylating MELT motifs on KNL1, but instead by removing PLK1 from the Bub complex. The authors find that these phosphatases antagonise CDK1-phosphorylations on BubR1 and Bub1 to dampen PLK1 levels. This activity is crucial to prevent PLK1 from maintaining MELT phosphorylation in an autocatalytic manner, thus (probably) allowing prompt SAC silencing following stable kinetochore-microtubule attachments. The described mechanism extends our view of how the SAC is regulated and should be of interest to those in the field of mitosis. The findings described in this paper allow us to better understand how cells silence the SAC. This is a top priority in the field, as the inability to timely quench SAC signaling can result in chromosome segregation errors. Determining the phosphatases that actually dephosphorylate the MELT motifs will be an essential next step forward

Reviewer #2 (Evidence, reproducibility and clarity (Required)):

****Summary:****

The work focuses on the role of kinetochore localized protein phosphatases in the dephosphorylation of MELT motifs and SAC silencing. The focus is on PP1 bound to KNL1 and PP2A-

B56 bound to BubR1 and uses largely RNAi rescue experiments in human cell lines combined with immunofluorescence analysis and time-lapse imaging. The authors show that kinetochore localized phosphatases antagonize the localization of the Plk1 mitotic kinase to kinetochores. This is due to the dephosphorylation of BubR1 T620 and Bub1 T609 that are binding sites for Plk1 on the kinetochore. The main conclusion is that if Plk1 kinetochore localisation is prevented then there is no longer a need for kinetochore phosphatases for SAC silencing and MELT dephosphorylation.

****Major comments:****

1) In its current state I am not convinced that the key conclusions are fully supported by the experiments and alternative conclusions/interpretations can be drawn. For example the level of MELT phosphorylation will be determined by the balance of kinase and phosphatase activity and if they do not achieve 100% inhibition of Mps1 in their assays then they are not strictly monitoring dephosphorylation kinetics in their assays. If the combination of Mps1 and Plk1 inhibition then more strongly inhibits Mps1 then dephosphorylation kinetics becomes faster. Thus subtle differences in Mps1 activity under their different conditions could lead to misleading conclusions but in its present state a careful analysis of Mps1 activity is not provided. This lack of complete inhibition also applies to the phosphatases and the experiments in Figure 3E indicates that their Calyculin preparation is not really active as at steady state MELT phosphorylation levels are much less affected than in for instance BubR1 del PP2A (Figure 2A as an example). Thus they likely still have phosphatase activity in the experiment in figure 3E making it difficult to draw the conclusions they do. A more careful analysis of kinase and phosphatase activities in their different perturbations would be recommendable and should be possible within a reasonable time frame.

These are good points and we will now more carefully assess MPS1 and PP1/PP2A activities.

2) A more stringent test of their model would also be needed. What happens if Plk1 is artificially maintained in the Bub complex? The prediction would be that SAC silencing should be severely delayed even when Mps1 is inhibited. This is a straightforward experiment to do that should not take too long. If the polobox can bind phosphoSer then one could also make BubR1 T620S to slow down dephosphorylation of this site (PPPs work slowly on Ser while Cdk1 have almost same activity for Ser and Thr).

These are good suggestions and we will try to see if maintaining PLK1 at the BUB complex produces effects on the SAC.

3) Another issue is the relevance of Plk1 removal under normal conditions. As their quantification shows in figure 1D-E (I think there is something wrong with figure 1E - should likely be Bub1) the contribution of BubR1 T620 and Bub1 T609 to Plk1 kinetochore localisation seems minimal. Thus upon SAC satisfaction there is not really a need to remove Plk1 through dephosphorylation as it is already at wild type levels. It is only in their BubR1 and KNL1 mutants that there is this effect so one has to question the impact in a normal setting. This is consistent with the data in Figure S1D showing no phosphorylation of these sites under unperturbed conditions.

The major finding of this study is that kinetochore phosphatases are primarily needed to suppress PLK1 activity on the BUB complex and thereby prevent excessive MELT phosphorylation. The relevance of this continued PLK1 removal under normal conditions is clear, because when it cannot occur (i.e. if the phosphatases are removed) then the SAC cannot be silenced unless PLK1 is inhibited. Therefore, whilst it is true that PLK1 localisation to the BUB complex is low under normal

conditions, that is because the phosphatases are working to keep it that way. The relevance of that continual removal is an interesting, but in our opinion, separate question that will require a new body of work to resolve. One possibility is that PLK1 recruitment is a continual dynamic process, that is perhaps coupled to a particular stage in MCC assembly. For example, PLK1 could bind the BUB complex to recruit PP2A to BUBR1, before being immediately removed by PP2A. In this sense, PLK1 binding could still be functionally important even if it is only occurs transiently and steady state PLK1 levels are low. We will add a line to the discussion to highlight that it would be interesting to test PLK1 dynamics on the BUB complex in future.

4) They write that in the absence of phosphatase activity Plk1 becomes capable of supporting SAC independently (of Mps1 is implied). They do not show this - only that MELT phosphorylation is maintained. As Mps1 has other targets required for SAC activity I would rephrase this.

Good point, this will be rephrased.

5) The method section is extensive and contains sufficient information for reproducing data.

6) Data and statistical analysis is ok.

Reviewer #2 (Significance (Required)):

The advance is clearly conceptual and provides a new way of thinking about the kinetochore localized phosphatases. These phosphatases and the SAC have been immensely studied but this work brings in a new angle. The discussion would benefit from some evolutionary perspectives as the PP1 and PP2A-B56 binding sites are very conserved but the Plk1 docking sites on Bubs less so. This will be of interest to people in the field of cell division and researchers interested in phospho-mediated signaling.

Since the paper was submitted, we performed evolutionary analysis to examine this point. We discovered that the PLK1 docking sites are surprisingly well conserved and, in fact, they appear to have co-evolved within the same region of MAD/BUB along with the PP2A-B56 binding motif. We believe this new data strengthens our manuscript because it argues strongly for an important functional relationship between PLK1 and PP2A. A new figure containing this evolutionary analysis will be included in the final version.

Field of expertise: kinetochore/phosphatases/bub proteins

Jakob Nilsson

Reviewer #3 (Evidence, reproducibility and clarity (Required)):

The Spindle Assembly Checkpoint (SAC) is a conserved surveillance device that responds to errors in kinetochore-microtubule attachments to ultimately prevent the onset of anaphase until all chromosomes are bipolarly attached. Current models of SAC posit that the Mps1 kinase initiates the SAC signalling cascade by phosphorylating the KNL1/Blinkin kinetochore scaffold at MELT repeats, in order to create phospho-docking sites for the hetero-tetrameric BUB complex made by BUB1-BUB3-BUB3-BUBR1. The BUB complex, in turn, promotes the assembly the Mitotic Checkpoint Complex

(MCC), which prevents anaphase onset by inhibiting the E3 ubiquitin ligase Anaphase-Promoting Complex bound to its activator Cdc20 (APCCdc20). The polo-like kinase PLK1, which is recruited to kinetochores through its binding to BUBR1, contributes to the robustness of SAC signalling in human cells by cooperating with Mps1 in KNL1/Blinkin phosphorylation and by phosphorylating MPS1 itself, thereby enhancing its catalytic activity. While in human cells MPS1 is the predominant kinase in SAC signalling, aided by PLK1, in other organisms where MPS1 is absent, such as in nematodes, PLK1 functionally replaces MPS1 and is necessary for SAC activation.

Once all chromosomes are bipolarly attached, SAC signalling is extinguished. Key to this process are the PP1 and PP2A-B56 phosphatases that antagonise KNL1 phosphorylation by MPS1 and PLK1 and also dephosphorylate the T-loop of MPS1 to lower its catalytic activity. Current models envision that PP1 and PP2A-B56 dephosphorylate the MELT repeats of KNL1 directly. Importantly, this has been formally tested for both PP2A-B56 in human cells (Espert et al., 2014) and PP1 in yeast (London et al., 2012).

In the present manuscript, the above model is challenged with the proposal that the main contribution of PP1 and PP2A-B56 to SAC silencing is to lower the levels of PLK1 at kinetochores, rather than to dephosphorylate KNL1. By interfering with the levels of these opposing kinases and phosphatases at kinetochores the authors describe an interesting interplay that confirms an overlapping function of PLK1 and MPS1 in KNL1 phosphorylation and highlights a role for the phosphatases in dampening PLK1 kinetochore levels. Consistently, inhibition of both Mps1 and PLK1 is sufficient to bring about KNL1 dephosphorylation upon inhibition of both phosphatases at kinetochores.

The hypothesis is interesting and experiments are in general carefully designed and performed. It is clear from the presented data that PP1 and PP2A-B56 antagonize PLK1 kinetochore localisation and that the MELT repeats of KNL1 can be dephosphorylated even in the absence of phosphatases, provided that MPS1 and PLK1 are inhibited. However, in my opinion the results do not rule out that phosphatases actually have a primary and direct role in KNL1 dephosphorylation.

****Major comments:****

1. An important limitation of this study is that KNL1 dephosphorylation at MELT repeats is monitored only by indirect immunofluorescence using phospho-specific antibodies. Thus, reduction of phospho-KNL1 kinetochore signals could be due to protein turnover at kinetochores, rather than to dephosphorylation. This is a serious issue that could be addressed by checking KNL1 dephosphorylation during time course experiments by western blot using phospho-specific antibodies, as previously done (Espert et al., 2014).

This is an important point that we feel is best addressed by examining total KNL1 levels at kinetochores (instead of simply total cellular levels by western blots). The reason is that KNL1 could potentially still be lost from kinetochores even if the total protein is not degraded. In all experiments involving YFP-KNL1 we observe no change in kinetochore KNL1 levels and this data will be included in the final version. We will also perform new experiments to examine total KNL1 levels in the BUBR1-WT/ Δ PP2A situation to test whether KNL1 kinetochore levels are similarly maintained in these cells following MPS1 inhibition.

2. For obvious technical reasons, the shortest time point at which authors compare KNL1 dephosphorylation upon MPS1-PLK1 inhibition is 5 minutes. Based on immunofluorescence data, authors conclude that kinetics of KNL1 dephosphorylation are similar when kinases are inhibited,

independent of whether or not kinetochore-bound phosphatases are active. However, in most experiments (e.g. Fig. 3B, 3C, 3E) lower levels of MELT phosphorylation are detected after 5 minutes of kinase inhibition when phosphatases are present than when they are absent, suggesting that phosphatases likely do contribute to KNL1 dephosphorylation. I suspect that differences between the presence and absence of phosphatases might even be more obvious if authors were to look at shorter time points, when phosphatases conceivably accomplish their function. I would therefore suggest that the authors tone down their conclusions, as their data complement but do not disprove the previous model.

We appreciate that small differences can be seen in figure 3B and 3E at the 5-minute timepoint (between the WT and phosphatase inhibited situations). This may reflect a role for the phosphatases in dephosphorylation or in the ability of drugs such as BI-2536 (3B) or Calyculin A (3E) to fully inhibit their targets in the short timeframe. We will perform additional experiments to examine MPS1 and phosphatase activity under these conditions, in response to comments by reviewers 1 and 2. In the final version we will carefully interpret the new and existing data and, if required, modify the conclusions appropriately.

3. In all experiments cells are kept mitotically arrested through nocodazole treatment, which is not quite a physiological condition to study SAC silencing. This could potentially mask the real contribution of phosphatases in MELT dephosphorylation. Indeed, it is possible that higher amounts of phosphatases are recruited to kinetochores during SAC silencing than during SAC signalling (e.g. during SAC signalling Aurora B phosphorylates the RVSF motif of KNL1 to keep PP1 binding at low levels; Liu et al., 2010). What would happen in a nocodazole wash-out? Would phosphatases be dispensable in these conditions for normal kinetics of MELT dephosphorylation and anaphase onset if PLK1 is inhibited?

All SAC silencing assays were performed in nocodazole for 2 main reasons: 1) PP2A-B56, PP1 or PLK1 can all regulate kinetochore-microtubule attachments, and thereby control the SAC indirectly. Therefore, performing our assays in the absence of microtubules allows us to make specific and direct conclusions about SAC regulation; 2) Previous work on pMELT regulation by PP1/PP2A in human cells was also performed following MPS1 inhibition in nocodazole (Espert et al 2014, Nijenhuis et al, 2014). Therefore, we are able to directly compare the contribution of PLK1 to the previously observed phenotypes, which allowed us to conclude that PLK1 has a major influence.

Nevertheless, we appreciate the point that the influence of PLK1 could, in theory, be different during a normal mitosis when microtubule attachment can form. Therefore, we will attempt to address whether PLK1 inhibition can bypass a requirement for PP1/PP2A in SAC silencing during an unperturbed mitosis.

4. Other data are overinterpreted. For instance, the evidence that CDK1-dependent phosphorylation sites in Bub1 and BubR1 is enhanced when PP1 and PP2A-B56 are absent at kinetochores suggests but does not "demonstrate that PP1-KNL1 and BUBR1-bound PP2A-B56 antagonise PLK1 recruitment to the BUB complex by dephosphorylating key CDK1 phosphorylation sites on BUBR1 (pT620) and BUB1 (pT609)(Figure 1F)". Similarly, the claim "when kinetochore phosphatase recruitment is inhibited, PLK1 becomes capable of supporting the SAC independently" referred to Fig. 2C-D is an overstatement, as residual MPS1 kinase could be still active in the presence of the AZ-3146 inhibitor.

These are good points and the indicated statements will be reworded.

****Minor comments:****

1. In many graphs (Fig. 1A-C, Fig. 2A,C) relative kinetochore intensities are quantified over "CENPC or YFP-KNL1". Authors should clarify when it is one versus the other.

This will be clarified in the axis and in the methods.

2. The drawing in Fig. 1F depicts the action of PP1 and PP2A-B56 in antagonising PLK1 at kinetochores. Thus, the output should be SAC silencing, rather than activation.

The SAC symbol will be removed from the schematic to avoid confusion and because it is not actually the focus of figure 1 anyway.

3. In the Discussion authors speculate that KNL1 dephosphorylation relies on a constitutive phosphatase with unregulated basal activity. Would a phosphatase be needed at all when MPS1 and PLK1 are inhibited? Could phosphorylated KNL1 be actively degraded?

We will insert total KNL1 immunofluorescence quantification so show that KNL1 KT levels are not decreased in this situation. KNL1 remains anchored at kinetochore but the MELTs must be dephosphorylated to remove the BUB complex.

4. What happens to MPS1 when KNL1-bound PP1 and BUBR1-bound PP2A are absent? Do its kinetochore levels increase as observed for PLK1? And what about the kinetochore levels of Bub1 and BubR1?

We have demonstrated previously that BUB1/BUBR1 increase in this situation in line with the MELTs (Nijenhuis et al 2014; I Smith et al, 2019) – these papers will be referenced in relation to this. We will also address the effect of phosphatase removal on MPS1 activity, in response to comments by reviewers 1 and 2.

Reviewer #3 (Significance (Required)):

The nature of the advance is conceptual. This paper challenges (although I would rather say "integrates") the prevailing model of spindle checkpoint silencing.

The current model of SAC silencing envisions that PP1 and PP2A-B56 phosphatases oppose SAC kinases (Mps1 and Polo kinase) by directly dephosphorylating some of their targets (e.g. the kinetochore scaffold KNL1 and MPS1 itself). This work proposes instead that the main function of the above phosphatases is to keep low levels of the polo kinase PLK1 at kinetochores, which would otherwise boost KNL1 phosphorylation and assembly of SAC complexes.

People working in the fields of mitosis, chromosome segregation, aneuploidy, spindle checkpoint, kinases/phosphatases could be interested by these findings.

Reviewer's field of expertise: Cell cycle, mitosis, spindle assembly checkpoint

February 19, 2020

Re: JCB manuscript #202002020T

Dr. Adrian Saurin
University of Dundee
Jacqui Wood Cancer Centre
James Arrott Drive
Dundee DD1 9SY
United Kingdom

Dear Dr. Saurin,

Thank you for submitting your manuscript entitled "Kinetochore phosphatases suppress autonomous kinase activity to control the spindle assembly checkpoint". We apologize for the delay in providing you with a decision.

In any case, we have now had an opportunity to assess your paper and the reviewer reports provided by Review Commons as well as your revision plan/rebuttal. We feel that the paper could be a good fit for JCB as a short Report and so we invite you to submit a revision if you can address the reviewers' key concerns, as outlined in your revision plan.

Please note that we cannot make any guarantees about the ultimate outcome of the paper as we will likely need to seek feedback from the reviewers on the revised manuscript (and there are points in the revision plan where you indicate that you plan to address the indicated issue but do not explain how you will do so and, thus, we will need to assess the fully revised manuscript before we can commit to moving forward). However, we are optimistic that the paper will be suitable for JCB once fully revised.

Please note that, for JCB Report papers, the results and discussion sections should be merged, so you will need to rewrite that section of the paper accordingly. Also, please note that Reports may have a maximum of 5 main figures and 3 supplemental figures - while the main figure limit is immutable, we can allow a bit more space in the supplemental information, if you find that you need more room to fully address each of the reviewers' concerns.

Please see here for our formatting guidelines:

<https://rupress.org/jcb/pages/submission-guidelines#manuscript-prep>

GENERAL GUIDELINES:

Text limits: Character count for a Report is < 20,000, not including spaces. Count includes title page, abstract, introduction, 'Results & Discussion' section, and acknowledgments. Count does not include materials and methods, figure legends, references, tables, or supplemental legends.

Figures: Transfers may have up to 5 main text figures. To avoid delays in production, figures must

be prepared according to the policies outlined in our Instructions to Authors, under Data Presentation, <http://jcb.rupress.org/site/misc/ifora.xhtml>. All figures in accepted manuscripts will be screened prior to publication.

IMPORTANT: It is JCB policy that if requested, original data images must be made available. Failure to provide original images upon request will result in unavoidable delays in publication. Please ensure that you have access to all original microscopy and blot data images before submitting your revision.

Supplemental information: There are strict limits on the allowable amount of supplemental data. Transfers may have up to 3 supplemental figures. Up to 10 supplemental videos or flash animations are allowed. A summary of all supplemental material should appear at the end of the Materials and methods section.

Our typical timeframe for revisions is three months; if submitted within this timeframe, novelty will not be reassessed at the final decision. Please note that papers are generally considered through only one revision cycle, so any revised manuscript will likely be either accepted or rejected.

Thank you for this interesting contribution to Journal of Cell Biology. You can contact us at the journal office with any questions, cellbio@rockefeller.edu or call (212) 327-8588.

Sincerely,

Timothy Yen, PhD
Monitoring Editor
Journal of Cell Biology

Tim Spencer, PhD
Executive Editor
Journal of Cell Biology

Response to all reviewers

We would like to thank all reviewers for carefully considering our manuscript and providing excellent suggestions for improvement. The general consensus was that our study provides an important conceptual advance which reveals a new way of thinking about kinetochore phosphatases. However, in light of our surprising findings, it was suggested that additional experiments would be required to fully validate our conclusions. In particular, it was seen as important to test whether PLK1 can activate MPS1 from the BUB complex and to confirm that PP1 and PP2A are effectively inhibited with Calyculin A in situations where MELT dephosphorylation can occur normally. It was also suggested that the manuscript would benefit from evolutionary analysis of the PLK1 binding motif on the BUBs to compare to the phosphatase binding motifs that are generally well conserved.

We now submit a revised manuscript with extensive new experiments to respond to all of these points. These are discussed at length in a point-by-point response to each reviewer below, but a summary of the main changes follows:

- A new figure (Fig. 3) to explore whether PLK1 can regulate MPS1 activity from the BUB complex
- Additional experiments to: 1) optimise Calyculin A dosing protocol (Figure 4E-F, S4H-I and S5), 2) demonstrate that Calyculin A inhibits PP1 and PP2A activity (Figure S5A-D), and 3) reduce phosphatase activity even further by combining genetic mutation with phosphatase inhibition (Figs.4F and S5J).
- Substantial new evolutionary analysis (Figs.5, S7, S8 and S9) that demonstrates a strong co-evolution between the PLK1 and PP2A binding motifs on the BUB complex in metazoa and demonstrates new conserved features of the polo-binding motif
- A number other figure panels in the main text and supplementary data to address specific points as outline in the point-by-point rebuttal below.
- Finally, we have also modified the text in a number of places to be more careful in our interpretations.

Although it has been incredibly challenging to perform so many additional experiments given the current constraints imposed by the COVID19 pandemic, we feel that the effort has been worthwhile since this new data has considerably strengthened our original manuscript. We thank all reviewers for helping to improve this study.

Reviewer #1 (Evidence, reproducibility and clarity (Required)):

The manuscript by Cordeiro et al provides a series of compelling evidences to support a provocative conclusion: PP2A-B56 and PP1 are critical for SAC silencing mainly by restraining and extinguishing autonomous kinase activity at kinetochores. This finding challenges the prevailing view of PP2A-B56/PP1-mediated KNL1-MELT dephosphorylation as a major SAC silencing event. This represents a paradigm change in the field and opens an important goal for future research: determine the phosphatases that dephosphorylate the MELTs. In my view this paper delivers an important clarification on how PP1-KNL1 and PP2A-B56 actually drive SAC silencing. This is a nice study and will move the field forward. The manuscript is globally solid, very well written and the conclusions are generally supported by the experimental data. However, I do have some issues with the following points, which in my view, if unaddressed, may leave the conclusion a bit fragile:

Minor comments:

1) The authors propose that PP1-KNL1 and BUBR1-bound PP2A-B56 continuously antagonise PLK1 association with the BUB complex by dephosphorylating the CDK1 phosphorylation sites on BUBR1 (pT620) and BUB1 (pT609). It is therefore expected that converting these residues to aspartate would increase PLK1 recruitment. It would be interesting to verify if this hypothesis fits with the proposed model.

The general idea to maintain PLK1 at the BUB complex is a good one, but unfortunately polo-box domains do not bind to negatively charged acidic residues. Instead we tried to maintain PLK1 at the BUB complex by performing Thr-Ser mutations (on BUB1 and BUBR1) in an attempt to antagonise PP2A dephosphorylation and enhance PBM phosphorylation (as suggested by reviewer 2). This was also driven by our new evolutionary analysis (Figures 5, S7 and S8) which demonstrates a strong conservation of serine in this position. Unfortunately, this approach did not increase either kinetochore PLK1 or pMELT or delay their dephosphorylation after MPS1 inhibition. In fact, BUB1-T609S mutation led to a slight reduction in PLK1/pMELT levels and a weakening of the SAC. Although we cannot test phosphorylation of this mutant directly, we presume that even if it is elevated it cannot effectively enhance PLK1 binding. The new data is in figure S9.

2) In Figure 1E, are the mean values for BubR1WT+BubWT and BubR1WT+Bub1T609 both normalized to 1? If so, this fails to reveal the contribution of Bub1 T609 for the recruitment of PLK1 when PP2A-B56 is allowed to localize at kinetochores.

The values have been updated and normalised to the BubR1WT + BUB1WT control. We also performed additional experiments on this and overall the results reveal a small reduction in kinetochore PLK1 following just BUB1-T609A mutation and a larger reduction upon combined BUBR1-T620A mutation (figure 1E). This correlates with a reduction in basal pMELT levels under these conditions (Figure 2G). We performed new SAC strength analysis in these situations, and this now demonstrates that BUB1-T609A reduces SAC strength and additional BUBR1-T620A reduces this even further (Figure 2H). Therefore, basal recruitment of PLK1 to BUB1 and BUBR1 is important to maintain a robust SAC response.

3) What underlies the increase in Bub1 levels at unattached kinetochores of siBubR1 cells (Figure S1C)? Is this caused by an increase in Bub1 T609 phosphorylation and consequently unopposed PLK1 recruitment, which consequently increases MELT phosphorylation?

We suspect that PLK1 is not the cause of the increased BUB1 levels because PLK1 kinetochore levels are actually decreased in this situation (Figure S1A).

4) Although the immunoblotting from Figure S1D indicates that BubR1T620A and Bub1T609A are expressed at similar levels as their respective WT counterparts, some degree of single-cell variability is expected to occur. As a complement to Figure 1B,C and Figure S1E,F could the authors plot the kinetochore intensity of BubR1 pT620 and Bub1T609 relative to the YFP-BubR1 and YFP-Bub1 signal, respectively?

There is indeed variability in the level of re-expression of BUBR1/BUB1 on a single cell level, which can at least partially explain the variation on BUBR1-pT620 and BUB1-pT609 observed within each condition. We plot the requested scatter plots below. We also plotted the BUBR1-pT620 and BUB1-pT609 with respect to level of the YFP-BUBR1 or YFP-BUB1 proteins to demonstrate how this reduces some of the variability compared with to the values normalised to Cenp C (shown alongside for comparison). For space restrictions in the main text we propose to leave these plots in the reviewer responses, which we will elect to publish with the main article

Figure 1B - BUBR1-pT620

Figure 1C - BUB1-pT609

Figure S2B - BUBR1-pT620

Figure S2C - BUB1-pT609

5) The authors nicely show that excessive PLK1 levels at the BUB complex are able to maintain MELT phosphorylation and the SAC (independently of MPS1) when KNL1-localised phosphatases are removed (Figures 2A,B). However, it should be noted that PLK1 is able to promote MPS1 activation at kinetochores and so, whether AZ-3146 at 2.5 μ M efficiently inhibits MPS1 under conditions of excessive PLK1 recruitment should be confirmed. Can the authors provide a read-out for MPS1 activation status or activity (other than p-MELTS) to exclude a potential contribution of residual MPS1 activity in maintaining the p-MELTS and SAC?

This is a good point and we now include a new main figure to address this (Figure 3). We chose to use MAD1-pT716 which is another validated MPS1 SAC target at kinetochores. The new data demonstrate that neither phosphatase binding mutant has any effect on the kinetics MAD1-pS716 dephosphorylation after MPS1 inhibition (Figure 3A-C). We also used a range of AZ-3146 dose to partially inhibit MPS1 and did not observe differences in MAD1-pT716 levels between BUBR1-WT and Δ PP2A cells (Figures 3D and E). This demonstrates that the PLK1 effects on MELT cannot be explained by a general effect on MPS1 activity, and it also implies that neither PP1 or PP2A-B56 are required for MAD1 dephosphorylation.

6) To examine whether PLK1 removal is the major role of PP1-KNL1 and PP2A-B56 in the SAC or whether they are additionally needed to dephosphorylate the MELTs, the authors monitored MELT dephosphorylation when MPS1 was inhibited immediately after 30-minute of BI2356. This revealed similar dephosphorylation kinetics, irrespective of compromised PP1-KNL1 or PP2A-B56 activity, thus suggesting that these pools of phosphatases are not required to dephosphorylate MELTs. To confirm this and exclude phosphatase redundancy, the authors simultaneously depleted all PP1 and B56 isoforms or treated cells with Calyculin A to inhibit all PP1 and PP2A phosphatases. In both of these situations, the kinetics of MELT dephosphorylation was indistinguishable from wild type cells if MPS1 and PLK1 were inhibited together. These observations led to the conclusion that neither PP1 or PP2A are required to dephosphorylate the MELT motifs. Instead they are needed to remove PLK1 from the BUB complex. This set of experiments is well-designed and the results support the conclusion. However, it would be of value if the authors provide evidence for the efficiency of PP1 and B56 isoforms depletion and for the efficiency of phosphatase inhibition by Calyculin A. An alternative read-out for the activity of PP1 and PP2A-B56 (other than p-MELT dephosphorylation) clearly confirming that both phosphatases are compromised when MPS1 and PLK1 are inhibited together could make a stronger case in excluding the contribution of residual PP1 or PP2A to the observed dephosphorylation of MELT motifs.

We appreciate the need to solidify this figure given the importance of conclusively ruling out a role for PP1 and PP2A in MELT dephosphorylation. We have now done this by the addition of several new experiments:

Firstly, we performed analysis on RepoMan, which we believe is an excellent control to demonstrate PP1/PP2A inhibition. Localised PP1/PP2A-B56 antagonise an Aurora B phosphorylation site on RepoMan (pSer893) that prevents its chromatin association. When Aurora B is inhibited during prometaphase, RepoMan translocates onto chromatin as a result of PP1/PP2A-mediated dephosphorylation of pSer893, and this translocation can be prevented by either inhibiting phosphatase binding to RepoMan or by using Calyculin A (Qian et al, 2013). Using this assay, we were able to demonstrate that the identical Calyculin A treatment protocols used to assess MELT dephosphorylation during prometaphase, were also able to prevent RepoMan translocation (Fig. S5A-D)

We also attempted an additional Calyculin A treatment regime to maximise phosphatase inhibition, which was limited in duration previously due to cell rounding/toxicity (now fully explained in the text). The new optimised protocol allows effective PP1/PP2A inhibition but did not prevent MELT dephosphorylation (Figs.4E, S4H and S5)

Finally, we tried combining Calyculin A with genetic inhibition (BUBR1- Δ PP2A) as a stringent method to reduce phosphatase activity as low as possible (Figs. 4F and S5J). These experiments also provided no evidence that PP1 or PP2A are involved in MELT dephosphorylation.

We believe that these new experiments have significantly strengthened our original conclusion, however we have nevertheless elected to be more careful in our interpretations. In particular, to acknowledge the fact that none of these conditions are able to fully inhibit phosphatase activity. This statement has been added after the relevant results in Figure 4:

“We can find no evidence that either PP1 or PP2A-B56 are directly required for MELT dephosphorylation, despite multiple attempts to reduce activities as low as possible using a combination of phosphatase inhibition, knockdown and/or binding-motif mutants. We acknowledge that phosphatase inhibition is never absolute under any of these conditions, therefore it is at least conceivable that PP1 and/or PP2A are so efficient at MELT dephosphorylation that we were unable to detect appreciable effects by any of these treatments. However, we favour the interpretation that an additional phosphatase, perhaps with unregulated basal activity, controls MELT dephosphorylation.”

To summarize, this is a very good paper and will definitely cause an important impact in the field of mitosis.

Thank you for this balanced and useful review

Reviewer #1 (Significance (Required)):

This manuscript provides an important conceptual advance for the field of mitosis, specifically to the topic of mitotic checkpoint regulation. It remains elusive how the spindle assembly checkpoint is silenced. While previous studies have shown that PP1-KNL1 and PP2A-B56 contribute to suppress SAC signalling, how they do so is unclear. This study provides important insight into this matter. Cordeiro and colleagues demonstrate that in contrast with previous expectations, PP1 and PP2A promote SAC silencing, not by directly dephosphorylating MELT motifs on KNL1, but instead by removing PLK1 from the Bub complex. The authors find that these phosphatases antagonise CDK1- phosphorylations on BubR1 and Bub1 to dampen PLK1 levels. This activity is crucial to prevent PLK1 from maintaining MELT phosphorylation in an autocatalytic manner, thus (probably) allowing prompt SAC silencing following stable kinetochore-microtubule attachments. The described mechanism extends our view of how the SAC is regulated and should be of interest to those in the field of mitosis. The findings described in this paper allow us to better understand how cells silence the SAC. This is a top priority in the field, as the inability to timely quench SAC signalling can result in chromosome segregation errors. Determining the phosphatases that actually dephosphorylate the MELT motifs will be an essential next step forward

Reviewer #2 (Evidence, reproducibility and clarity (Required)):

****Summary:****

The work focuses on the role of kinetochore localized protein phosphatases in the dephosphorylation of MELT motifs and SAC silencing. The focus is on PP1 bound to KNL1 and PP2A-B56 bound to BubR1 and uses largely RNAi rescue experiments in human cell lines combined with immunofluorescence analysis and time-lapse imaging. The authors show that kinetochore localized phosphatases antagonize

the localization of the Plk1 mitotic kinase to kinetochores. This is due to the dephosphorylation of BubR1 T620 and Bub1 T609 that are binding sites for Plk1 on the kinetochore. The main conclusion is that if Plk1 kinetochore localisation is prevented then there is no longer a need for kinetochore phosphatases for SAC silencing and MELT dephosphorylation.

****Major comments:****

1) In its current state I am not convinced that the key conclusions are fully supported by the experiments and alternative conclusions/interpretations can be drawn. For example the level of MELT phosphorylation will be determined by the balance of kinase and phosphatase activity and if they do not achieve 100% inhibition of Mps1 in their assays then they are not strictly monitoring dephosphorylation kinetics in their assays. If the combination of Mps1 and Plk1 inhibition then more strongly inhibits Mps1 then dephosphorylation kinetics becomes faster. Thus subtle differences in Mps1 activity under their different conditions could lead to misleading conclusions but in its present state a careful analysis of Mps1 activity is not provided. This lack of complete inhibition also applies to the phosphatases and the experiments in Figure 3E indicates that their Calyculin preparation is not really active as at steady state MELT phosphorylation levels are much less affected than in for instance BubR1 del PP2A (Figure 2A as an example). Thus they likely still have phosphatase activity in the experiment in figure 3E making it difficult to draw the conclusions they do. A more careful analysis of kinase and phosphatase activities in their different perturbations would be recommendable and should be possible within a reasonable time frame.

These are all valid points and we have now included a significant number of new experiments to address these issues.

Firstly, we now include a new figure 3 to address the possibility that PLK1 could generally enhance MPS1 activity. Using MAD1-pT716 as an alternative readout for kinetochore MPS1 activity we detect no difference in the kinetics of dephosphorylation in the presence or absence phosphatase binding (figure 3A-C). We also tested whether subtle difference in MPS1 activity maybe present by using different doses of AZ-3146 to partially inhibit MPS1 (Figure 3D and E). Again, we could detect no difference using this assay between BUBR1-WT and Δ PP2A cells. We therefore conclude that BUB-PLK1 does not generally increase MPS1 activity, and additionally, PP1 or PP2A-B56 are not responsible for dephosphorylating MAD1-pT716 directly.

To address the potential for residual phosphatase activity following Calyculin A treatment we performed a number of new experiments. Firstly, we assessed RepoMan chromatin translocation to RepoMan as another readout that has been shown previously to rely on localised PP1 or PP2A activity (Qian et al, 2013). This demonstrate that our Calyculin A prep and dosing schedule is sufficient to effectively inhibit PP1/PP2A activity at RepoMan (Fig.S5A-D).

Regarding the point about our old Fig.3E, we appreciate that pMELT did not rise to the same degree as in the Δ PP2A, but this is not due to the activity of our Calyculin A preparation. Instead, it is perhaps due to the fact that treatment with Calyculin A at the previous 50nM dose must be limited to less than 25 mins to prevent complete cell rounding and loss of viability – this likely reflects the pleotropic effect PP1/PP2A inhibition, which is now discussed in the text. This meant that only a 5-min pre-treatment was possible, therefore we instead tried a 25nM dose which allowed a 15 mins pre-treatment schedule. This new schedule better enhanced basal pMELT and completely prevented RepoMan translocation following Aurora B inhibition, but still had no effect on PP1/PP2A dephosphorylation (see figures 4E, S4 and S5).

Finally, we attempted to sensitize phosphatase inhibition even further by combining Calyculin A with a BUBR1- Δ PP2A mutant. This experiment was also unable to provide evidence for a role of PP1/PP2A in MELT dephosphorylation (Figure 4F and S5J).

Nevertheless, we acknowledge the fact that none of these methods are able to completely abolish phosphatase activity, therefore we have added the following statement to our interpretation regarding the new figure 4.

“We can find no evidence that either PP1 or PP2A-B56 are directly required for MELT dephosphorylation, despite multiple attempts to reduce activities as low as possible using a combination of phosphatase inhibition, knockdown and/or binding-motif mutants. We acknowledge that phosphatase inhibition is never absolute under any of these conditions, therefore it is at least conceivable that PP1 and/or PP2A are so efficient at MELT dephosphorylation that we were unable to detect appreciable effects by any of these treatments. However, we favour the interpretation that an additional phosphatase, perhaps with unregulated basal activity, controls MELT dephosphorylation.”

2) A more stringent test of their model would also be needed. What happens if Plk1 is artificially maintained in the Bub complex? The prediction would be that SAC silencing should be severely delayed even when Mps1 is inhibited. This is a straightforward experiment to do that should not take too long. If the polobox can bind phosphoSer then one could also make BubR1 T620S to slow down dephosphorylation of this site (PPPs work slowly on Ser while Cdk1 have almost same activity for Ser and Thr).

We tried the Thr-Ser switch, as suggested, which we also deemed worthwhile because our new evolutionary analysis demonstrated strong conservation of serine in this position (see below). Unfortunately, neither the mutation of BUB1-T609S or BUBR1-T620S caused an effect on PLK1 level or MELT dephosphorylation. In fact, PLK1 binding and MELT levels were slightly decrease in BUB1-T609S cells, suggesting that threonine may be needed to bind PLK1 effectively (Fig. S9).

3) Another issue is the relevance of Plk1 removal under normal conditions. As their quantification shows in figure 1D-E (I think there is something wrong with figure 1E - should likely be Bub1) the contribution of BubR1 T620 and Bub1 T609 to Plk1 kinetochore localisation seems minimal. Thus upon SAC satisfaction there is not really a need to remove Plk1 through dephosphorylation as it is already at wild type levels. It is only in their BubR1 and KNL1 mutants that there is this effect so one has to question the impact in a normal setting. This is consistent with the data in Figure S1D showing no phosphorylation of these sites under unperturbed conditions.

The major finding of this study is that kinetochore phosphatases are primarily needed to suppress PLK1 activity on the BUB complex and thereby prevent excessive MELT phosphorylation. The relevance of this continued PLK1 removal under normal conditions is clear, because when it cannot occur (i.e. if the phosphatases are removed) then the SAC cannot be silenced unless PLK1 is inhibited. Therefore, whilst it is true that PLK1 localisation to the BUB complex is low under normal conditions, that is because the phosphatases are working to keep it that way. However, to directly address if this low-level BUB-PLK1 is relevant for basal SAC signalling we performed additional experiments to removed PLK1 from the BUB complex under conditions where the phosphatases are not inhibited. This new data demonstrates that MELT phosphorylation and SAC strength are reduced in BUBR1-T620A cells, and decreased even further in BUBR1-T620A+BUB1-T609A cells (Figure 2G-H). This demonstrates that PLK1 recruitment to these sites has a role in basal SAC signalling. Therefore, even though PLK1 levels on the BUB complex are apparently low, this pool of PLK1 is still able to contribute positively to SAC signalling.

4) They write that in the absence of phosphatase activity Plk1 becomes capable of supporting SAC

independently (of Mps1 is implied). They do not show this - only that MELT phosphorylation is maintained. As Mps1 has other targets required for SAC activity I would rephrase this.

Good point, the text has now been changed considerably and this statement has been removed. The text has also been modified in other parts to make more careful interpretations.

5) The method section is extensive and contains sufficient information for reproducing data.

6) Data and statistical analysis is ok.

Reviewer #2 (Significance (Required)):

The advance is clearly conceptual and provides a new way of thinking about the kinetochore localized phosphatases. These phosphatases and the SAC have been immensely studied but this work brings in a new angle. The discussion would benefit from some evolutionary perspectives as the PP1 and PP2A-B56 binding sites are very conserved but the Plk1 docking sites on Bubs less so. This will be of interest to people in the field of cell division and researchers interested in phospho-mediated signaling.

The manuscript now contains throughout evolutionary analysis in 152 eukaryotic species containing MADBUB homologs (figures 5, S7 and S8). This new analysis shows a striking co-evolution between the PLK1 and PP2A binding motifs within the same region of MAD/BUB in metazoa, which argues for a strong functional role for PLK1-PP2A cross-regulation. There are three separate interesting features identified by this evolutionary analysis and these are now discussed extensively in the text.

Field of expertise: kinetochore/phosphatases/bub proteins

Jakob Nilsson

Reviewer #3 (Evidence, reproducibility and clarity (Required)):

The Spindle Assembly Checkpoint (SAC) is a conserved surveillance device that responds to errors in kinetochore-microtubule attachments to ultimately prevent the onset of anaphase until all chromosomes are bipolarly attached. Current models of SAC posit that the Mps1 kinase initiates the SAC signalling cascade by phosphorylating the KNL1/Blinkin kinetochore scaffold at MELT repeats, in order to create phospho-docking sites for the hetero-tetrameric BUB complex made by BUB1-BUB3-BUB3-BUBR1. The BUB complex, in turn, promotes the assembly the Mitotic Checkpoint Complex (MCC), which prevents anaphase onset by inhibiting the E3 ubiquitin ligase Anaphase-Promoting Complex bound to its activator Cdc20 (APCCdc20). The polo-like kinase PLK1, which is recruited to kinetochores through its binding to BUBR1, contributes to the robustness of SAC signalling in human cells by cooperating with Mps1 in KNL1/Blinkin phosphorylation and by phosphorylating MPS1 itself, thereby enhancing its catalytic activity. While in human cells MPS1 is the predominant kinase in SAC signalling, aided by PLK1, in other organisms where MPS1 is absent, such as in nematodes, PLK1 functionally replaces MPS1 and is necessary for SAC activation. Once all chromosomes are bipolarly attached, SAC signalling is extinguished. Key to this process are the PP1 and PP2A-B56 phosphatases that antagonise KNL1 phosphorylation by MPS1 and PLK1 and also dephosphorylate the T-loop of MPS1 to lower its catalytic activity. Current models envision that PP1 and PP2A-B56 dephosphorylate the MELT repeats of KNL1 directly. Importantly, this has been formally tested for both PP2A-B56 in human cells (Espert et al., 2014) and PP1 in yeast (London et al., 2012).

In the present manuscript, the above model is challenged with the proposal that the main contribution of PP1 and PP2A-B56 to SAC silencing is to lower the levels of PLK1 at kinetochores, rather than to dephosphorylate KNL1. By interfering with the levels of these opposing kinases and phosphatases at kinetochores the authors describe an interesting interplay that confirms an overlapping function of PLK1 and MPS1 in KNL1 phosphorylation and highlights a role for the phosphatases in dampening PLK1 kinetochore levels. Consistently, inhibition of both Mps1 and PLK1 is sufficient to bring about KNL1 dephosphorylation upon inhibition of both phosphatases at kinetochores.

The hypothesis is interesting and experiments are in general carefully designed and performed. It is clear from the presented data that PP1 and PP2A-B56 antagonize PLK1 kinetochore localisation and that the MELT repeats of KNL1 can be dephosphorylated even in the absence of phosphatases, provided that MPS1 and PLK1 are inhibited. However, in my opinion the results do not rule out that phosphatases actually have a primary and direct role in KNL1 dephosphorylation.

****Major comments:****

1. An important limitation of this study is that KNL1 dephosphorylation at MELT repeats is monitored only by indirect immunofluorescence using phospho-specific antibodies. Thus, reduction of phospho-KNL1 kinetochore signals could be due to protein turnover at kinetochores, rather than to dephosphorylation. This is a serious issue that could be addressed by checking KNL1 dephosphorylation during time course experiments by western blot using phospho-specific antibodies, as previously done (Espert et al., 2014).

This is an important point that we felt was best addressed by examining total KNL1 levels at kinetochores (instead of simply total cellular levels by western blots). The reason is that KNL1 could potentially still be lost from kinetochores even if the total protein is not degraded. We have now included analysis of KNL1 levels throughout, particularly in figures S3B, D, G, H; S4C and S5G, H. This new analysis demonstrates that the observed phospho-dependent changes cannot be explained by changes to KNL1 kinetochore levels.

2. For obvious technical reasons, the shortest time point at which authors compare KNL1 dephosphorylation upon MPS1-PLK1 inhibition is 5 minutes. Based on immunofluorescence data, authors conclude that kinetics of KNL1 dephosphorylation are similar when kinases are inhibited, independent of whether or not kinetochore-bound phosphatases are active. However, in most experiments (e.g. Fig. 3B, 3C, 3E) lower levels of MELT phosphorylation are detected after 5 minutes of kinase inhibition when phosphatases are present than when they are absent, suggesting that phosphatases likely do contribute to KNL1 dephosphorylation. I suspect that differences between the presence and absence of phosphatases might even be more obvious if authors were to look at shorter time points, when phosphatases conceivably accomplish their function. I would therefore suggest that the authors tone down their conclusions, as their data complement but do not disprove the previous model.

We appreciate that small differences can be seen in figure 3B and 3E at the 5-minute timepoint (between the WT and phosphatase inhibited situations). This may reflect a role for the phosphatases in dephosphorylation or in the ability of drugs such as BI-2536 (Fig. 4B) or Calyculin A (Fig. 4E) to fully inhibit their targets in the short timeframe. We have now performed extensive additional analysis to address these issues, in response to specific comments by reviewers 1 and 2. The new data demonstrate that the timing of drug penetration is an issue - compare the kinetics of dephosphorylation in the PLK1 + Calyculin A sample in new figure 4E (old 3E, but now including PLK1 pre-treatment). In this case a 30

min PLK1 pre-treatment produced similar fast dephosphorylation after 5mins. In addition, we have confirmed Calyculin A inhibits PP1/PP2A activity towards other substrates (RepoMan, fig. S5A-D), we have tried a new calyculin treatment protocol that allows longer Calyculin A pre-treatment (figs. S5) and we have sensitized phosphatase inhibition even further by combining Calyculin A with a BUBR1- Δ PP2A mutant – our strongest way to inhibit phosphatases activity (figs.4F and S5J). None of these approaches provides evidence for the involvement of PP1/PP2A in MELT dephosphorylation.

Nevertheless, we appreciate your suggestion to use caution when interpreting these results because phosphatase inhibition is never absolute under any of these conditions. Therefore, we have added the following statement after presenting the results in figure 4 and we have also toned-down interpretations at other places in the text, including the abstract.

“We can find no evidence that either PP1 or PP2A-B56 are directly required for MELT dephosphorylation, despite multiple attempts to reduce activities as low as possible using a combination of phosphatase inhibition, knockdown and/or binding-motif mutants. We acknowledge that phosphatase inhibition is never absolute under any of these conditions, therefore it is at least conceivable that PP1 and/or PP2A are so efficient at MELT dephosphorylation that we were unable to detect appreciable effects by any of these treatments. However, we favour the interpretation that an additional phosphatase, perhaps with unregulated basal activity, controls MELT dephosphorylation.”

3. In all experiments cells are kept mitotically arrested through nocodazole treatment, which is not quite a physiological condition to study SAC silencing. This could potentially mask the real contribution of phosphatases in MELT dephosphorylation. Indeed, it is possible that higher amounts of phosphatases are recruited to kinetochores during SAC silencing than during SAC signalling (e.g. during SAC signalling Aurora B phosphorylates the RVSF motif of KNL1 to keep PP1 binding at low levels; Liu et al., 2010). What would happen in a nocodazole wash-out? Would phosphatases be dispensable in these conditions for normal kinetics of MELT dephosphorylation and anaphase onset if PLK1 is inhibited?

All SAC silencing assays were performed in nocodazole for 2 main reasons: 1) PP2A-B56, PP1 or PLK1 can all regulate kinetochore-microtubule attachments, and thereby control the SAC indirectly. Therefore, performing our assays in the absence of microtubules allows us to make specific and direct conclusions about SAC regulation; 2) Previous work on pMELT regulation by PP1/PP2A in human cells was also performed following MPS1 inhibition in nocodazole (Espert et al 2014, Nijenhuis et al, 2014). Therefore, we are able to directly compare the contribution of PLK1 to the previously observed phenotypes, which allowed us to conclude that PLK1 has a major influence.

The suggested nocodazole washout experiments are very challenging, because PLK1 inhibition is needed for bipolar spindle assembly and PP2A-B56 is needed for initial kinetochore microtubule attachments. Therefore nocodazole-washout would need to be performed just in KNL1- Δ PP1 mutants and PLK1 would then need to be inhibited at a sufficient time after nocodazole washout to allow bipolar spindle assembly first, but prior to natural pMELT dephosphorylation which would occur quickly afterwards. Due to the complicated nature of these experiments and the restrictions imposed on us by COVID19, we elected to concentrate on other key areas instead. However, we have been careful not to make statements about natural SAC silencing in the presence of microtubules, except to state at the very end of the discussion that the engagement of PP1-KNL1 upon tension is likely a key event in SAC silencing.

4. Other data are overinterpreted. For instance, the evidence that CDK1-dependent phosphorylation sites in Bub1 and BubR1 is enhanced when PP1 and PP2A-B56 are absent at kinetochores suggests but does not "demonstrate that PP1-KNL1 and BUBR1-bound PP2A-B56 antagonise PLK1 recruitment to the BUB complex by dephosphorylating key CDK1 phosphorylation sites on BUBR1 (pT620) and BUB1 (pT609)(Figure 1F)". Similarly, the claim "when kinetochore phosphatase recruitment is inhibited, PLK1

becomes capable of supporting the SAC independently" referred to Fig. 2C-D is an overstatement, as residual MPS1 kinase could be still active in the presence of the AZ-3146 inhibitor.

These are good points and the indicated statements have been reworded. We have also carefully checked the text and reworded other interpretations where necessary.

****Minor comments:****

1. In many graphs (Fig. 1A-C, Fig. 2A,C) relative kinetochore intensities are quantified over "CENPC or YFP-KNL1". Authors should clarify when it is one versus the other.

This are now clarified in the legends of the relevant figures.

2. The drawing in Fig. 1F depicts the action of PP1 and PP2A-B56 in antagonising PLK1 at kinetochores. Thus, the output should be SAC silencing, rather than activation.

The SAC symbol has been removed from the schematic to avoid confusion, since it is not actually the focus of figure 1 anyway.

3. In the Discussion authors speculate that KNL1 dephosphorylation relies on a constitutive phosphatase with unregulated basal activity. Would a phosphatase be needed at all when MPS1 and PLK1 are inhibited? Could phosphorylated KNL1 be actively degraded?

Our new inserted KNL1 quantifications show that KNL1 KT levels are not decreased in this situation (figures S3B, D, G, H; S4C and S5G, H). KNL1 remains anchored at kinetochores, but the MELTs must be dephosphorylated to remove the BUB complex. While it is possible that phosphorylates KNL1 protein is released and replaced by new unphosphorylated KNL1, we feel this is unlikely.

4. What happens to MPS1 when KNL1-bound PP1 and BUBR1-bound PP2A are absent? Do its kinetochore levels increase as observed for PLK1? And what about the kinetochore levels of Bub1 and BubR1?

We have now analysed another kinetochore MPS1 substrate as a downstream readout for MPS1 activity (MAD1-pT716: see new Figure 3). This demonstrates that MPS1 kinetochore activity is not generally elevated by phosphatase inhibition. We have also performed additional experiments to show the effect of phosphatase binding mutants on BUB1 and BUBR1 levels (Fig. S2D-E). BUB1/BUBR1 do rise at kinetochores in these conditions, as shown previously (Nijenhuis et al 2014; Smith et al, 2019), but the increase is much less than that seen for BUBR1-pT620 and BUB1-pT609 (Fig. 1B,C).

Reviewer #3 (Significance (Required)):

The nature of the advance is conceptual. This paper challenges (although I would rather say "integrates") the prevailing model of spindle checkpoint silencing.

The current model of SAC silencing envisions that PP1 and PP2A-B56 phosphatases oppose SAC kinases (Mps1 and Polo kinase) by directly dephosphorylating some of their targets (e.g. the kinetochore scaffold KNL1 and MPS1 itself). This work proposes instead that the main function of the above

phosphatases is to keep low levels of the polo kinase PLK1 at kinetochores, which would otherwise boost KNL1 phosphorylation and assembly of SAC complexes.

People working in the fields of mitosis, chromosome segregation, aneuploidy, spindle checkpoint, kinases/phosphatases could be interested by these findings.

Reviewer's field of expertise: Cell cycle, mitosis, spindle assembly checkpoint

September 17, 2020

RE: JCB Manuscript #202002020R

Dr. Adrian Saurin
University of Dundee
Jacqui Wood Cancer Centre
James Arrott Drive
Dundee DD1 9SY
United Kingdom

Dear Adrian:

Thank you for submitting your revised manuscript entitled "Kinetochore phosphatases suppress autonomous Polo-like kinase 1 activity to control the mitotic checkpoint". The paper has now been seen by the original reviewers (two of whom decided to sign their reviews, as you will see in their reports) and they now all recommend acceptance. We would therefore be happy to publish your paper in JCB pending final revisions necessary to meet our formatting guidelines (see details below).

As you will see in the reviewer reports below this email, reviewers #2 and 3 have voiced a few lingering concerns that should be addressed prior to publication. These are relatively minor issues that will not require new experiments and can be addressed by further discussion and/or comments in the main text.

****Please be sure to provide a point-by-point rebuttal to these remaining reviewer concerns.****

A. MANUSCRIPT ORGANIZATION AND FORMATTING:

Full guidelines are available on our Instructions for Authors page, <https://jcb.rupress.org/submission-guidelines#revised>. ****Submission of a paper that does not conform to JCB guidelines will delay the acceptance of your manuscript.****

1) Text limits: Character count for Reports is < 20,000, not including spaces. Count includes title page, abstract, introduction, results, discussion, and acknowledgments. Count does not include materials and methods, figure legends, references, tables, or supplemental legends. As you know, you are a bit over this limit but we should be able to give you the extra space this time.

2) Figure formatting: Scale bars must be present on all microscopy images, including inset magnifications. Molecular weight or nucleic acid size markers must be included on all gel electrophoresis.

3) Statistical analysis: Error bars on graphic representations of numerical data must be clearly described in the figure legend. The number of independent data points (n) represented in a graph must be indicated in the legend. Statistical methods should be explained in full in the materials and methods. For figures presenting pooled data the statistical measure should be defined in the figure

legends. Please also be sure to indicate the statistical tests used in each of your experiments (both in the figure legend itself and in a separate methods section) as well as the parameters of the test (for example, if you ran a t-test, please indicate if it was one- or two-sided, etc.). Also, since you used parametric tests in your study (e.g. t-tests, ANOVA, etc.), you should have first determined whether the data was normally distributed before selecting that test. In the stats section of the methods, please indicate how you tested for normality. If you did not test for normality, you must state something to the effect that "Data distribution was assumed to be normal but this was not formally tested."

4) Materials and methods: Should be comprehensive and not simply reference a previous publication for details on how an experiment was performed. Please provide full descriptions (at least in brief) in the text for readers who may not have access to referenced manuscripts. The text should not refer to methods "...as previously described."

5) Please be sure to provide the sequences for all of your primers/oligos and RNAi constructs in the materials and methods. You must also indicate in the methods the source, species, and catalog numbers (where appropriate) for all of your antibodies.

6) Microscope image acquisition: The following information must be provided about the acquisition and processing of images:

- a. Make and model of microscope
- b. Type, magnification, and numerical aperture of the objective lenses
- c. Temperature
- d. imaging medium
- e. Fluorochromes
- f. Camera make and model
- g. Acquisition software
- h. Any software used for image processing subsequent to data acquisition. Please include details and types of operations involved (e.g., type of deconvolution, 3D reconstitutions, surface or volume rendering, gamma adjustments, etc.).

7) References: There is no limit to the number of references cited in a manuscript. References should be cited parenthetically in the text by author and year of publication. Abbreviate the names of journals according to PubMed.

8) Supplemental materials: As you know, JCB Reports usually may have up to 3 supplemental figures. However, as discussed previously, we will allow you to have the extra space this time. Please also note that tables, like figures, should be provided as individual, editable files. A summary of all supplemental material should appear at the end of the Materials and methods section.

9) Conflict of interest statement: JCB requires inclusion of a statement in the acknowledgements regarding competing financial interests. If no competing financial interests exist, please include the following statement: "The authors declare no competing financial interests." If competing interests are declared, please follow your statement of these competing interests with the following statement: "The authors declare no further competing financial interests."

10) A separate author contribution section is required following the Acknowledgments in all research manuscripts. All authors should be mentioned and designated by their first and middle initials and full surnames. We encourage use of the CRediT nomenclature (<https://casrai.org/credit/>).

11) ORCID IDs: ORCID IDs are unique identifiers allowing researchers to create a record of their various scholarly contributions in a single place. At resubmission of your final files, please consider providing an ORCID ID for as many contributing authors as possible.

B. FINAL FILES:

-- High-resolution figure and video files: See our detailed guidelines for preparing your production-ready images, <https://jcb.rupress.org/fig-vid-guidelines>.

Thank you for this interesting contribution, we look forward to publishing your paper in Journal of Cell Biology.

Sincerely,

Timothy Yen, PhD
Monitoring Editor
Journal of Cell Biology

Tim Spencer, PhD
Executive Editor
Journal of Cell Biology

Reviewer #1 (Comments to the Authors (Required)):

This is a good and provocative study that in my opinion provides important conceptual advance to the field of mitosis. The revised manuscript by Cordeiro and colleagues, adequately addresses most of my previous concerns, thus strengthening the authors' initial conclusions. I appreciate the effort that the authors have put in performing additional experiments during the COVID19 pandemic.

Unfortunately, the authors' strategy of converting Thr to Ser on Bub1 and BubR1 failed to increment the interaction with PLK1, which leaves their model partially unchallenged. I also realize that, despite commendable effort, the authors cannot guarantee 100% inhibition of PP2A and PP1. This prevents the authors from excluding with absolute confidence the MELTs as possible substrates of PP2A and PP1. However, these two issues should not discourage publication. The manuscript recognizes these caveats and provides an honest interpretation and discussion of the data. Furthermore, the revised version of the manuscript includes additional data that supports the role of BUB-associated PLK1 in maintaining basal SAC signaling, which further strengthens the relevance of its continued removal (by PP1 and PP2A) to ensure responsiveness/timely SAC silencing. Also, the inclusion of evolutionary analysis demonstrating a strong co-evolution between the PLK1 and PP2A binding motifs on the BUB complex, as well as uncovering new conserved features of the polo-binding domain, represents a nice addition to the manuscript and clearly improves this study.

I also find it interesting that phosphorylation of Mad1 Thr716 does not seem to be directly antagonized by either PP2A-B56 or PP1-KNL1. I agree with the authors that MAD1-pT716 is dephosphorylated with identical kinetics when MPS1 is inhibited in the presence or absence PP2A-B56 or PP1-KNL1, thus implying that kinetochore phosphatases do not impact directly on MAD1-pT716 or more generally on kinetochore MPS1 activity (At least in nocodazole-treated cells. It is still possible that additional microtubule-dependent pools of PP1 may impact on Mps1 activity or even on MAD1-pT716).

There's a typo in the last sentence of page 6: bindexs > binds.

In brief, the main findings of this study are:

- 1- PP1-KNL1 and BUBR1-bound PP2A-B56 reduces PLK1 recruitment to the BUB complex by antagonizing key CDK1 phosphorylation sites on BUBR1 (pT620) and BUB1 (pT609). - This conclusion is well supported by the data (Figure 1).
- 2- PLK1 binds to the BUB complex to enhance MELT phosphorylation and PP1-KNL1 and PP2-B56 antagonize this recruitment to allow SAC silencing following MPS1 inhibition. - This conclusion is well supported by the data depicted in Figure 2 and the appropriate controls are presented in the corresponding supplementary Figures.
- 3- The PLK1-dependent increment of MELT phosphorylation and SAC function observed in BUBR1 Δ PP2A cells treated with AZ-3146 is not driven by increased Mps1 activity - This conclusion is in part supported by the observation that under these circumstances MAD1-pT716 is severely reduced as in wild type cells. It would have been useful to check the phosphorylation status of Mps1 T-loop as well (or other autophosphorylation sites), since the kinase might be much more

efficient in phosphorylating MELTs than MAD1. Showing that Mps1 activation is not incremented would help to rule out this possibility. Nevertheless, the conclusion that PLK1 is able to support MELT phosphorylation and SAC signaling without affecting MPS1 activity towards another key SAC substrate is valid. It is however puzzling (at least to me) that given the critical role attributed to MAD1-pT716 in MCC assembly, that BUBR1 Δ PP2A cells treated with AZ-3146 are still quite competent in their SAC function. Is the rate of MCC formation in the absence of Mad1-T716 phosphorylation sufficient to sustain the SAC? A brief discussion of this point could be useful.

4- The primary role of PP1 and PP2A in SAC silencing is to remove PLK1 from the BUB complex. - The authors have now included several new experiments that, in my opinion, support this conclusion with a higher degree of confidence.

5- PBM and KARD motifs exhibit strong co-evolution and marked co-localization in the same region of MADBUB throughout metazoan.

Collectively, this paper nicely shows that phosphatase regulation at kinetochores is critical, but mostly to restrain and extinguish localized kinase activity. This is a feature of SAC signaling that seems to be conserved throughout evolution. Based on these points, I recommend the publication of this manuscript in The Journal of Cell Biology.

Carlos Conde

Reviewer #2 (Comments to the Authors (Required)):

The authors have done a good job in addressing several of the points raised by the reviewers. I support the publication in JCB following a few minor adjustments:

1) I assume that BubR1 T620A might have reduced levels of PP2A-B56 bound to the LxxIxE motif but I am not sure if the authors have checked this. I would at least mention this possibility as this is important for interpreting some of the results.

2) I would mention how much of Plk1 is recruited by BubR1 vs. Bub1 based on their IF data. It looks to me that BubR1 is mediating the bulk of Plk1 recruitment.

3) Reviewer 3 had an important point on checking KNL phosphorylation status independently of IF. This they have not done and so a line mentioning that it is assumed that antibody accessibility is constant and not affected by any of the treatments would be important to have.

4) Could the authors comment on whether they favour that PP1 is dephosphorylating T620 and T609 and the effect of mutating PP2A binding site is indirectly through affecting the RVxF motif.

5) On page 5 they cite the 2012 Kops paper for identifying the LxxIxE motif. This is not a correct citation as the Kops lab argued for similarity to Sgol. I would recommend citing either Kruse 2013 or Hertz 2016 here. Further down they cite Wang 2016 papers for showing that phosphorylation of LxxIxE motif by Plk1 enhances PP2A-B56 binding. Here the correct citations would be the Kops 2012 paper and Kruse 2013.

6) The authors focus very much on SAC silencing. It could be a possibility that Plk1 recruits PP2A-B56 to BuBR1 to have dynamic phosphorylation of MELT motifs and other Plk1/Mps1 phosphorylation sites which is important for alignment. Mentioning this in the discussion could be useful because the way the discussion is written now one cannot help think why this system is so conserved if you do not need the PPPs.

Jakob Nilsson

Reviewer #3 (Comments to the Authors (Required)):

The authors have answered satisfactorily to my previous comments and criticisms. I therefore support publication in JCB.

One point that needs further clarification is why authors use CENPC kinetochore levels to normalise phosphorylation of KNL1 at MELT repeats, instead of using KNL1 kinetochore levels (e.g. in BUBR1delPP2A cells), which would be a more appropriate control. This is particularly relevant in light of the fact that in several experiments (e.g. Fig. S3B,D related to Fig. 2A,C) kinetochore levels of KNL1 are elevated in BUBR1delPP2A relative to WT cells. To what extent is the fluorescence intensity of pMELT-KNL1 accounted for by increased levels of KNL1 in BUBR1delPP2A cells?

There are a few typos in the text and figures. Some examples (not an exhaustive list):
inhibitior instead of inhibitor (Fig. 3D-E, S9F)
calycuin instead of calyculin (Fig. 4E)

In the legend of Fig. S3 I believe that authors mean "restore" rather than "rescue"

Reviewer #1 (Comments to the Authors (Required)):

I also find it interesting that phosphorylation of Mad1 Thr716 does not seem to be directly antagonized by either PP2A-B56 or PP1-KNL1. I agree with the authors that MAD1-pT716 is dephosphorylated with identical kinetics when MPS1 is inhibited in the presence or absence PP2A-B56 or PP1-KNL1, thus implying that kinetochore phosphatases do not impact directly on MAD1-pT716 or more generally on kinetochore MPS1 activity (At least in nocodazole-treated cells. It is still possible that additional microtubule-dependent pools of PP1 may impact on Mps1 activity or even on MAD1-pT716).

- *We agree, our data cannot rule out a role for PP1 (or PP2A) after microtubules attach to kinetochores*

3- The PLK1-dependent increment of MELT phosphorylation and SAC function observed in BUBR1 Δ PP2A cells treated with AZ-3146 is not driven by increased Mps1 activity - This conclusion is in part supported by the observation that under these circumstances MAD1-pT716 is severely reduced as in wild type cells. It would have been useful to check the phosphorylation status of Mps1 T-loop as well (or other autophosphorylation sites), since the kinase might be much more efficient in phosphorylating MELTs than MAD1. Showing that Mps1 activation is not incremented would help to rule out this possibility. Nevertheless, the conclusion that PLK1 is able to support MELT phosphorylation and SAC signaling without affecting MPS1 activity towards another key SAC substrate is valid. It is however puzzling (at least to me) that given the critical role attributed to MAD1-pT716 in MCC assembly, that BUBR1 Δ PP2A cells treated with AZ-3146 are still quite competent in their SAC function. Is the rate of MCC formation in the absence of Mad1-T716 phosphorylation sufficient to sustain the SAC? A brief discussion of this point could be useful.

- *We suspect that these cells are competent in SAC function because pMAD1-pT716 does not drop to baseline levels at the 2.5 μ M dose of AZ that we used in most of assays. MAD1 dephosphorylation may indeed explain why the SAC is still weakened in Δ PP2A cells as the dose of MPS1 inhibition is increased (compare Figures. S3J and 2H).*

Reviewer #2 (Comments to the Authors (Required)):

1) I assume that BubR1 T620A might have reduced levels of PP2A-B56 bound to the LxxIxE motif but I am not sure if the authors have checked this. I would at least mention this possibility as this is important for interpreting some of the results.

- *We have added a statement to the discussion to mention that it will be important in future to determine whether local PLK1 on the BUB complex is needed for PP2A-B56 recruitment.*

2) I would mention how much of Plk1 is recruited by BubR1 vs. Bub1 based on their IF data. It looks to me that BubR1 is mediating the bulk of Plk1 recruitment.

- *We have added a statement to reinforce that BUBR1 is mediating the bulk of PLK1 recruitment.*

3) Reviewer 3 had an important point on checking KNL phosphorylation status independently of IF. This they have not done and so a line mentioning that it is assumed that antibody accessibility is constant and not affected by any of the treatments would be important to have.

- *Perhaps there was a misunderstanding regarding the previous point by reviewer 3 who stated that: “reduction of phospho-KNL1 kinetochore signals could be due to protein turnover at kinetochores, rather than to dephosphorylation”. The “turnover” in this case could mean protein degradation of KNL1 (as we had assumed) or protein binding/release of KNL1-binding proteins, which in principle could mask epitopes (which presumably you refer to now). If it was protein degradation, then we did address this in the most appropriate way by re-assessing kinetochore KNL1 levels extensively. I am presuming that this was the original intended meaning of reviewer 3 considering that they are now satisfied with our inclusion of these total KNL1 intensities. You are correct in also pointing out that epitope masking is an additional possible concern, but this applies to any immunofluorescence staining against any epitope involved in a protein-protein interaction. This includes most functional phospho-epitopes on the kinetochore and yet we have never seen this statement added to another publication, therefore we are not sure why it is seen as important in this specific case.*

4) Could the authors comment on whether they favour that PP1 is dephosphorylating T620 and T609 and the effect of mutating PP2A binding site is indirectly through affecting the RVxF motif.

- *At this stage it is difficult to say, but we have now added a line to the discussion to state that this is an important issue to resolve in future.*

5) On page 5 they cite the 2012 Kops paper for identifying the LxxIxE motif. This is not a correct citation as the Kops lab argued for similarity to SgoI. I would recommend citing either Kruse 2013 or Hertz 2016 here. Further down they cite Wang 2016 papers for showing that phosphorylation of LxxIxE motif by Plk1 enhances PP2A-B56 binding. Here the correct citations would be the Kops 2012 paper and Kruse 2013.

- *The reference on page 5 was cited in relation to the acronym “KARD”, which was originally coined by the Kops lab (“LxxIxE (also known as KARD (Suijkerbuijk et al., 2012)”). We appreciate it could be taken to mean the LxxIxE motif specifically and so we have also now also added the Hertz paper here.*
- *We have added the Kruse reference in relation to the pS676 PLK1 site, but the Suijkerbuijk 2012 paper from the Kops lab specifically address pT680 and not pS676 which we refer to specifically in the text. We have also kept the Wang references here since they provide important structural explanations for why this phosphorylation increases B56 affinity.*

6) The authors focus very much on SAC silencing. It could be a possibility that Plk1 recruits PP2A-B56 to BuBR1 to have dynamic phosphorylation of MELT motifs and other Plk1/Mps1 phosphorylation sites which is important for alignment. Mentioning this in the

discussion could be useful because the way the discussion is written now one cannot help think why this system is so conserved if you do not need the PPPs.

- *This is a good point and a discussion of relevance to microtubule attachment has now been included.*

Reviewer #3 (Comments to the Authors (Required)):

One point that needs further clarification is why authors use CENPC kinetochore levels to normalise phosphorylation of KNL1 at MELT repeats, instead of using KNL1 kinetochore levels (e.g. in BUBR1delPP2A cells), which would be a more appropriate control. This is particularly relevant in light of the fact that in several experiments (e.g. Fig. S3B,D related to Fig. 2A,C) kinetochore levels of KNL1 are elevated in BUBR1delPP2A relative to WT cells. To what extent is the fluorescence intensity of pMELT-KNL1 accounted for by increased levels of KNL1 in BUBR1delPP2A cells?

- *Cenp-C was used as internal control in all of these assays because it gives a strong consistent kinetochore stain, where the KNL1 antibodies are weaker and subject to noise. Therefore, it is not now possible to plot individual cells in relation to total KNL1. You can see an example of the variation in total KNL1 intensity when comparing KNL1 levels in Δ PP2A cells 0 min timepoint from S3B vs S3D, which is identical. It does appear, however, that there is a trend towards increased KNL1 in Δ PP2A cells, which is interesting and likely to at least contribute to the elevated MELT phosphorylation in this condition. Although it is important to state that this would not be able to account for the inability to dephosphorylate MELTs following MPS1 inhibition. We would also like to point out though that by plotting the phospho-KNL1 and total-KNL1 graphs separately (both in relation to Cenp-C) and referring the reader to both, they are then able to assess and reflect on the pKNL1 and KNL1 differences separately. This would not be possible if they were normalised to each other and Cenp-C was not included.*

There are a few typos in the text and figures. Some examples (not an exhaustive list):
inhibitor instead of inhibitor (Fig. 3D-E, S9F)
calycuin instead of calyculin (Fig. 4E)

In the legend of Fig. S3 I believe that authors mean "restore" rather than "rescue"

- *Thanks, these issues have been corrected and the manuscript has been carefully checked for others.*